# On the Optimization Trajectory of DeepWalk Embeddings

**Christopher Harker** [1]  **Aditya Bhaskara** [1]

## Abstract

The DeepWalk algorithm has been widely used for learning node embeddings in graphs. Combined with the idea of *negative sampling*, the DeepWalk algorithm has been shown to be implementable at scale, easily handling graphs with millions of nodes. However, theoretical guarantees on the resulting embeddings are much less understood. Recent results have studied the minimizers of the objective and have shown interesting guarantees for certain graph classes. However, the optimization *trajectory*, i.e., what happens when we start at a random initialization and run gradient descent, remains poorly understood. This is especially true for the implementation of DeepWalk using Skip-gram with negative sampling (SGNS), since the variance of the stochastic updates turns out to be very large. In this work, we make progress on this question. We show that for "small norm" initialization, under a spectral gap assumption on the graph, the DeepWalk embeddings align with the column space of a fixed low-rank matrix. For graphs generated from Stochastic Block Models with certain separation conditions, our results imply that the DeepWalk embeddings recover cluster structure. To the best of our knowledge, our results give the first analysis of the optimization trajectory of DeepWalk with negative sampling on non-trivial graph classes.

## 1. Introduction

Embeddings play a pivotal role in modern machine learning. Language models begin their inference and generation process by taking in embeddings of textual tokens. Reasoning, finding similarities, and other semantic analyses of text can often be related to the geometric structure of the associ-

ated embedding vectors. The Word2vec algorithm (Mikolov et al., 2013a), as well as more modern methods like BERT (Devlin et al., 2018) and ELMo (Peters et al., 2018), have been heavily influential in natural language processing. In the world of graphs, embeddings have been studied extensively, both for algorithmic purposes (e.g., (Alon, 1986; Sinclair & Jerrum, 1989; McSherry, 2001; Ng et al., 2001; Von Luxburg, 2007)), as well as in data mining applications such as node clustering (Belkin & Niyogi, 2001; Ng et al., 2001; Rohe et al., 2011; Von Luxburg, 2007), link prediction (Backstrom & Leskovec, 2011; Grover & Leskovec, 2016; Tang et al., 2015), etc. The DeepWalk (Perozzi et al., 2014) and Node2vec (Grover & Leskovec, 2016) algorithms are particularly interesting because they bring techniques from the NLP world (e.g., Word2vec) to graph analysis. By treating graph vertices as "words" and random walks on graphs as "sentences", these methods produce embeddings of graphs that capture graph properties, leading to their adoption for applications like community detection and node classification (Grover & Leskovec, 2016; Zhang & Tang, 2024). In a broader context, these algorithms are part of a generalized class of random walk based embedding methods that map network topologies to low-dimensional spaces by utilizing the graph's diffusion dynamics (Huang et al., 2021). For example, the works of Delvenne et al. (2010) and Schaub et al. (2019) leverage the autocovariance matrix of random-walk processes to construct dynamic node embeddings that can be used to detect community structures and measure node similarity.

However, in contrast to spectral embeddings or other convex programming methods, DeepWalk and Node2vec calculate embeddings by optimizing a nonconvex objective, and much less is known by way of theoretical guarantees on the resulting embeddings. The work of Levy & Goldberg (2014) and Qiu et al. (2018) showed that when the dimension of the embedding is large, specifically $\geq n$, the embedding vectors can be characterized *exactly* using the factorization of a certain mutual information matrix. When the embedding dimension is small (as is often the case in practice), Levy & Goldberg (2014) argue that the embedding vectors tend to *approximately* factorize the matrix (see Table 1 in Levy & Goldberg (2014)). However, they do not prove that the embedding vectors *align* with the cluster structure in when applied to graphs, particularly for low-dimensional embed-

---

[1]Kahlert School of Computing, University of Utah, Salt Lake City, UT, USA. Correspondence to: Christopher Harker <chris.harker@utah.edu>, Aditya Bhaskara <bhaskaraaditya@gmail.com>.

*Proceedings of the $43^{rd}$ International Conference on Machine Learning*, Seoul, South Korea. PMLR 306, 2026. Copyright 2026 by the author(s).

dings. More recent work has obtained such results. Kojaku et al. (2024) showed that assuming embeddings can be characterized in terms of the mutual information matrix (which holds for high-dimensional embeddings), the embedding vectors enable cluster recovery. For low-dimensional embeddings, Barot et al. (2021), Davison et al. (2024), Zhang & Tang (2024), and Harker & Bhaskara (2023) proved that for well-studied graph classes such as stochastic block models (SBM), the optimal solution to the nonconvex optimization procedure used in algorithms like DeepWalk and Node2Vec is guaranteed to align with the hidden clusters of the SBM. However, these works focus on the *global optima* of the DeepWalk and Node2vec objectives. For convex optimization, it is well known that gradient descent must converge to the global optimum. However for nonconvex optimization, it is well known that gradient descent may converge only to local optima or get "stuck" at saddle points. In practice, DeepWalk embeddings are obtained by starting with a random initial embedding and performing gradient descent. The question we ask is thus: *can we analyze the dynamics of gradient descent with random initialization for the DeepWalk objective, for nontrivial graph classes?*[1]

Analyzing gradient descent trajectories for nonconvex objectives has been a rich area of study, where several challenges still remain. There has been progress on several specific problems (e.g., (Bhojanapalli et al., 2016; Ge et al., 2017; 2016; Du et al., 2019; Arora et al., 2018)); see also the thesis by Du (2019) and references therein. In the context of graph embeddings, Harker & Bhaskara (2024) recently showed that for the *softmax-based* DeepWalk objective on stochastic block models (SBM), gradient descent with a sufficiently small initialization finds an embedding that approximately recovers the cluster structure of the SBM. Their main observation (see also Stöger & Soltanolkotabi (2021)), is that if parameters are initialized randomly within a small ball around the origin, gradient descent updates behave nearly like linear updates as long as the number of iterations is small enough, and so spectral methods can be leveraged. However, their work only applies to DeepWalk with the softmax objective, which is seldom used in practice for computational reasons, as we explain below. The approach fails when we consider negative sampling, a stochastic procedure (similar in spirit to SGD) that has proved crucial for scaling the DeepWalk algorithm to larger graphs.

Another recent work by Karkada et al. (2026) studies the training dynamics of Word2Vec by deriving a quartic Taylor approximation of the objectiive around the origin and solving its gradient-flow training dynamics. The gradient-flow solutions reveal that the model learns a discrete sequence

---

[1] We focus on DeepWalk in this paper. The Node2Vec objective, while similar, uses biased random walks (Grover & Leskovec, 2016). While our methods might extend, the co-occurrence matrix turns out to be more technical, and analyzing it remains open.

of rank-incrementing orthogonal linear subspaces. Nevertheless, they analyze an approximation of Word2Vec's objective rather than the true objective, and they do so under the conditions that the step size is tending towards zero and that the embeddings are symmetric. In reality, the algorithm relies on discrete stochastic updates with finite learning rates and the embeddings are asymmetric.

## 1.1. Our Results

Given a graph $G = (\mathcal{V}, \mathcal{E})$, the DeepWalk algorithm maintains and updates embedding vectors $\mathbf{x}_i, \mathbf{y}_i \in \mathbb{R}^d$ for each vertex $i \in \mathcal{V}$. By taking random walks and using a sliding window over the vertices in the walk (see Section 2 for details), the algorithm generates a sequence of vertex pairs $(i_1, j_1), (i_2, j_2), \ldots, (i_t, j_t), \ldots$. At each step, a loss function $\ell_t$ is considered (that depends on $i_t, j_t$), and the embedding vectors are updated using the gradient of $\ell_t$. The original work of Perozzi et al. (2014) used the loss function $\log \Pr(j_t|i_t)$, where $\Pr()$ is defined using a softmax. However, this turns out to be very expensive to work with. To alleviate the problem, Perozzi et al. (2014) used the hierarchical Softmax. However, a better approach turns out to be *negative sampling*, developed in the earlier Skip-gram context by Mikolov et al. (2013b). The key idea is to replace the computationally expensive denominator in softmax with a term that can be computed by taking a small number of samples. Modern implementations of DeepWalk typically use Skip-gram with negative sampling (SGNS). DeepWalk is appealing in applications because the step of generating walks is highly parallelizable, and gradient updates can computed quite efficiently using SGNS. (They can be parallelized to a considerable extent because of the sparsity of the gradient updates (e.g., (Zhu et al., 2019)).

Our main results are on analyzing the embeddings learned via SGNS for graphs drawn from a $K$-community stochastic block model (SBM). The question here is whether the learned embedding vectors reveal the (hidden) communities in the network. For the *symmetric* SBM, where we have $K$ groups of vertices of size $n/K$ each (see Section 2), we show two main results:

**Emergence of Low-Rank Structure.** Starting with a sufficiently small (inverse polynomial) random initialization, we prove that the $2n \times d$ matrix whose rows correspond to the $d$-dimensional embeddings of the vertices (we have two embeddings per vertex; see Section 2) will have all of its columns lying in a special $2K$-dimensional space (essentially the top eigenspace of the adjacency matrix of the SBM), with high probability. See Theorem 4.1 for the formal statement. The algebraic structure of this subspace is closely related to the autocovariance matrix of a random walk on the graph. The works of Delvenne et al. (2010), Schaub et al. (2019), and Huang et al. (2021) provide a

broader context for why random-walk-based embeddings perform well on downstream tasks by linking them to the dynamical properties of the graph. Our work complements these by showing that DeepWalk (with negative sampling) actually tracks these dynamical properties even under high-variance stochastic updates when the radius remains small.

**Cluster Recovery.** We show that after $MT$ iterations with $M, T$ both small polynomials of $n$, with high probability the embedding vectors are "well clustered", and the cluster structure aligns with that of the SBM. See Theorem 5.1 for the formal statement.

Our recovery guarantees for SBMs are similar to those in the recent work of Davison et al. (2024). However, their results focus on proving structural properties of *optimal embeddings*, i.e., the (global) minimizers of the (full) DeepWalk objective (see Equation (2)), while our goal is to understand stochastic updates and how the embeddings evolve over time (in other words, the SGD trajectory).

Typically, community recovery in the SBM (e.g., via spectral methods) requires the embedding dimension to be greater than the number of communities. Interestingly, our results hold for *all* embedding dimensions, even one-dimensional embeddings.

**Techniques.** Our results rely on a careful analysis of the stochastic SGNS updates. We derive a closed form for the updates and also the *expected* updates at each step. Then, we show that when the embedding vectors have a small norm, the updates admit a good linear approximation. This step is reminiscent of prior analyses, but the key challenge for us is that the stochastic updates have a factor $\approx n$ larger variance than the expected updates. Because of this, lengths of the embedding vectors can change significantly at each step. We show that increases in length are *spread out*, thus with high probability, the maximum length can be bounded well. Our analysis also draws from the elegant line of work on analyzing stochastic PCA (e.g., (Balsubramani et al., 2013; Jain et al., 2016; Shamir, 2015)). In contrast to these works, we have extra error terms arising due to the nonlinear sigmoid function, and we need to handle a larger variance. To control the error, we use a novel *conditional* martingale inequality, which may be of independent interest.

While the work of Harker & Bhaskara (2024) analyzes the trajectories of the embeddings throughout learning, they do so by studying the gradient descent dynamics of the original softmax-based DeepWalk objective, without negative sampling. In addition, Karkada et al. (2026) analyze an approximation to the SGNS objective assuming symmetric embeddings; by contrast, we analyze the true SGNS objective assuming asymmetric embeddings. Indeed, it is our inclusion of the nonlinear sigmoid function and asymmetric embeddings that makes our analysis challenging, requiring

new techniques. In further contrast to Karkada et al. (2026), while we require the learning rate to be small — even decreasing as a function of the number of nodes in the graph — we provide an upper bound on the learning rate under which our analysis still holds.

## 2. Notation and Preliminaries

For a column vector $\boldsymbol{x} = (x_1, \ldots, x_n) \in \mathbb{R}^n$, we denote the $\ell_2$ norm as $\|\boldsymbol{x}\| = [\sum_{i=1}^n x_i^2]^{1/2}$. We let $\mathbf{1} \in \mathbb{R}^n$ denote the all-ones vector and $\boldsymbol{e}_i \in \mathbb{R}^n$ the basis vector whose $i^{th}$ entry is equal to one and whose all other entries are equal to zero. We denote the dot product between two vectors $\boldsymbol{x}, \boldsymbol{y} \in \mathbb{R}^n$ as $\langle \boldsymbol{x}, \boldsymbol{y} \rangle$. The operator $diag(\boldsymbol{x}) \in \mathbb{R}^{n \times n}$ denotes the diagonal matrix whose $i^{th}$ diagonal entry is $x_i$. We also write $|\boldsymbol{x}| = \sum_{i=1}^n x_i$.

For any matrix $\boldsymbol{A} \in \mathbb{R}^{n \times n}$, let $\boldsymbol{A}_{i:}$ denote its $i^{th}$ row and $\boldsymbol{A}_{:j}$ denote its $j^{th}$ column. We denote the the sum of its entries as $|\boldsymbol{A}| = \sum_{i=1}^n \sum_{j=1}^n A_{ij}$. The spectral norm is denoted by $\|\boldsymbol{A}\| = \max_{\|\boldsymbol{x}\|=1} \|\boldsymbol{A}\boldsymbol{x}\|$ and the Frobenious norm by $\|\boldsymbol{A}\|_F = [\sum_{i,j=1}^n A_{ij}^2]^{1/2}$. Also, the denote the max row norm as $\|\boldsymbol{A}\|_{2,\infty} = \max_i \|\boldsymbol{A}_{i:}\|$. We define the degree operator $\boldsymbol{D} : \mathbb{R}^{n \times n} \to \mathbb{R}^{n \times n}$ as $\boldsymbol{D}(\boldsymbol{A}) = diag(\boldsymbol{A}\mathbf{1})$. At times, we use $\boldsymbol{D}_{\boldsymbol{A}}$ in place of $\boldsymbol{D}(\boldsymbol{A})$ when it is notationally convenient. The all-ones matrix (of dimensions $n \times n$ unless specified otherwise) will be denoted by $\boldsymbol{J}$. Given another matrix $\boldsymbol{B} \in \mathbb{R}^{n \times n}$, the element-wise multiplication of $\boldsymbol{A}$ and $\boldsymbol{B}$ is denoted as $\boldsymbol{A} \odot \boldsymbol{B}$.

Given a graph $\mathcal{G} = (\mathcal{V}, \mathcal{E})$ with $|\mathcal{V}| = n$ nodes and $|\mathcal{E}| = m$ edges, we denote the adjacency matrix of the graph as $\boldsymbol{A}$. For a node $i$, its degree is $d_i = \sum_{j=1}^n A_{ij}$. The transition matrix of a random walk on the graph is $\boldsymbol{P} = \boldsymbol{D}_{\boldsymbol{A}}^{-1} \boldsymbol{A}$, while the stationary distribution of a random walk on the graph is denoted by the probability vector $\boldsymbol{\pi}$, where $\pi_i = d_i/2m$. Lastly, given two positive integers $i$ and $j$, the Kronecker delta $\delta_{ij}$ is defined to be 1 if $i = j$ and 0 otherwise.

**The Stochastic Block Model.** The stochastic block model (SBM) (Holland et al., 1983) is a popular generative model for graphs with *community structure*, and it has served as a classic benchmark for community recovery algorithms (Abbe, 2018). We focus on the *symmetric $K$-block SBM*. Here, the vertices of the graph are partitioned into $K$ blocks or *clusters* $\Omega_1, \Omega_2, \ldots, \Omega_k$, of size $n/K$ each. The SBM has two additional parameters $p, q \in (0, 1)$, with $p > q$. Throughout our analysis, we assume that $n^{\rho-1} < q < p$ for some $\rho \in (0, 1)$. Two vertices in the same cluster have an edge with probability $p$ and vertices in different clusters have an edge with probability $q$ (independently). We *require a gap* between $p, q$ for recovery to be possible. In our paper, we assume $p > 1.1q$; any constant $> 1$ works for the analysis. (See Appendix A.1 for more details.)

## 2.1. The DeepWalk Algorithm

Given a graph $\mathcal{G} = (\mathcal{V}, \mathcal{E})$, the DeepWalk algorithm is an elegant iterative procedure that maintains and updates embedding vectors for the vertices $\mathcal{V}$. For each $i \in \mathcal{V}$, the algorithm maintains two $d$-dimensional embedding vectors, $\boldsymbol{x}_i$ and $\boldsymbol{y}_i$, that are called the *node* and *context* embedding.[2] We will denote by $\boldsymbol{X}$ and $\boldsymbol{Y}$ the $n \times d$ matrices whose rows denote the node and context embedding vectors of the $n$ nodes, respectively.

The DeepWalk algorithm consists of two procedures: the first one generates pairs of vertices $(i_t, j_t)$, for $t = 1, 2, \ldots$ using by random walks (see Appendix A.2 for details), and the second procedure, which can be viewed as operating in parallel with the first procedure, updates the embedding vectors using $(i_t, j_t)$, iteratively. From the perspective of a theoretical analysis, the first procedure is equivalent to sampling $(i_t, j_t)$ from an appropriately defined "limiting co-occurrence matrix" that we denote by $\boldsymbol{C}$. Corresponding to this pair, a (nonconvex) loss function $\ell_t$ is constructed as defined below, and the embedding $\boldsymbol{X}, \boldsymbol{Y}$ is updated using the gradients of $\ell_t$. Thus, DeepWalk can be viewed as online gradient descent with the nonconvex losses $\ell_t$.

Following prior theoretical works (Barot et al., 2021; Harker & Bhaskara, 2023; 2024; Qiu et al., 2018; Zhang & Tang, 2024), we have the following closed form for the *limiting* co-occurrence matrix $\boldsymbol{C}$, defined as the number of random walks tends to infinity (see Appendix A.2).

**Definition 2.1.** Given a graph $\mathcal{G} = (\mathcal{V}, \mathcal{E})$, for the random walk parameters $L$ (walk length), $S$ (context window), the limiting co-occurrence matrix is

$$\boldsymbol{C} = 2 \sum_{s=1}^{S} \frac{(L-s)}{2m} \boldsymbol{D_A} \boldsymbol{P}^s,$$

where $\boldsymbol{P} = \boldsymbol{D_A}^{-1} \boldsymbol{A}$ is the transition matrix of a random walk on the graph and $m$ is the number edges.

For matrix $\boldsymbol{C}$ as above, the random walk based sampling is equivalent to sampling $(i, j)$ with probability $\frac{C_{ij}}{|\boldsymbol{C}|}$, where $|\boldsymbol{C}|$ is simply the sum of all the entries (which by definition are non-negative); see Corollary A.3. We also assume that samples at different time steps are independent.

*Online update of embeddings.* As in Skip-gram for word embeddings, the idea is that nodes that have high "co-occurrence" are similar to one another. Thus, given a pair of vertices $(i, j)$, we consider the loss function:

$$\ell_{(i,j)}(\boldsymbol{X}, \boldsymbol{Y}) := \\ - \log \sigma(\langle \boldsymbol{x}_i, \boldsymbol{y}_j \rangle) - s_n \mathbb{E}_{l \sim P_n}[\log \sigma(-\langle \boldsymbol{x}_i, \boldsymbol{y}_l \rangle)], \quad (1)$$

---

[2]Note that while two vectors are common in implementations (following SkipGram), the original paper (Perozzi et al., 2014) used $\boldsymbol{x}_i = \boldsymbol{y}_i$. Our analysis assumes two embeddings per vertex.

where $\sigma(x) = (1 + e^{-x})^{-1}$ is the sigmoid function. The first term of the loss function encourages the embeddings of $i, j$ to be *close*. To avoid "collapse" (all vectors become equal), the second *contrastive* term is used. It encourages the embedding of node $i$ to be different from that of node $l$, drawn from a *negative sample distribution* $P_n$. The term $s_n$ is called the negative sampling parameter, i.e., the relative weight of the negative sampling (repulsive) term to the first (attractive) term. Traditionally (e.g., in Mikolov et al. (2013b)), $s_n$ is also called the *number* of negative samples and the negative sampling term in the objective is replaced by $\sum_{t=1}^{s_n} \mathbb{E}_{l_t \sim P_n}[\log \sigma(-\langle \boldsymbol{x}_i, \boldsymbol{y}_{l_t} \rangle)]$, and the expectation is computed using exactly one sampled index $l_t$. It is easy to see that these two formulations are the same (in expectation). The update rule for the embeddings $\boldsymbol{X}, \boldsymbol{Y}$ is now: initialize randomly, and iteratively move along the gradient of $\ell_{(i_t, j_t)}$, for $t = 1, 2, \ldots$.

We note that given Equation (1), the *full* or global loss function that DeepWalk with SGNS optimizes is:

$$\mathcal{L}(\boldsymbol{X}, \boldsymbol{Y}; \boldsymbol{C}) = \sum_{i,j} \frac{C_{ij}}{|\boldsymbol{C}|} \cdot \ell_{(i,j)}(\boldsymbol{X}, \boldsymbol{Y}). \qquad (2)$$

We might hope is that the iterative update procedure converges to the global minimum of Equation (2) and has a cluster structure. However, since the objectives are nonconvex, this need not hold. Our main contribution is to prove that the online update still reveals the cluster structure in graphs drawn from a stochastic block model.

## 3. Iterative Updates and Algorithm

In this section, we will derive simplified expressions for the DeepWalk SGNS update rule, that will allow us to analyze how embeddings evolve. We will also describe the algorithm and provide a high level overview of the analysis for graphs drawn from the SBM.

First, let us describe the negative sampling distribution $P_n$ from Equation (1). Prior work has used the uniform distribution, degree raised to a power, etc. For our analysis of symmetric SBM, all of these distributions are equal (up to lower order terms) to the uniform distribution. For concreteness, we will use the following choice of $P_n$ (following Barot et al. (2021); Zhang & Tang (2024); Harker & Bhaskara (2023)), that uses the co-occurrence matrix $\boldsymbol{C}$:

$$P_n(k) := \frac{\sum_{i=1}^{n} C_{ik}}{\sum_{i,j=1}^{n} C_{ij}} = \frac{|\boldsymbol{C}_{:k}|}{|\boldsymbol{C}|}. \qquad (3)$$

For a pair of randomly sampled nodes $(i, j)$ sampled with

probability $C_{ij}/|C|$, the local objective function is thus

$$\ell_{(i,j)}(X, Y; C) = - \log \sigma(\langle x_i, y_j \rangle)$$
$$- s_n \sum_{k=1}^n \frac{|C_{:k}|}{|C|} \log \sigma(-\langle x_i, y_k \rangle). \quad (4)$$

Then for a pair of nodes $(i_t, j_t)$ randomly sampled on iteration $t$, the corresponding stochastic update equations for all nodes $i' \in [n]$ with a learning rate $\eta > 0$ are

$$x_{i'}^{(t+1)} = x_{i'}^{(t)} - \eta \nabla_{x_{i'}} \ell_{(i_t, j_t)}(X^{(t)}, Y^{(t)}; C),$$
$$y_{i'}^{(t+1)} = y_{i'}^{(t)} - \eta \nabla_{y_{i'}} \ell_{(i_t, j_t)}(X^{(t)}, Y^{(t)}; C). \quad (5)$$

The update equations can be expressed in matrix form. Define the matrix $W \in \mathbb{R}^{2n \times d}$ that stacks the embedding and node matrices $X$ and $Y$:

$$W = \begin{bmatrix} X \\ Y \end{bmatrix}. \quad (6)$$

Also, for a pair of nodes $(i, j)$ we define the $n \times n$ matrix

$$\widetilde{H}_{(i,j)}(X, Y; C) = e_i e_i^\top \sigma(-XY^\top) e_j e_j^\top$$
$$- \frac{s_n}{|C|} e_i e_i^\top ((\mathbf{1}\mathbf{1}^\top C) \odot \sigma(XY^\top)), \quad (7)$$

where the sigmoid function $\sigma(\cdot)$ is applied element-wise. For ease of notation, we drop the explicit dependency of $X, Y$, and $C$, and instead write $\widetilde{H}_{(i,j)}$. Next, we define block matrix $H_{(i,j)} \in \mathbb{R}^{2n \times 2n}$ as

$$H_{(i,j)} = \begin{bmatrix} 0_{n \times n} & \widetilde{H}_{(i,j)} \\ \widetilde{H}_{(i,j)}^\top & 0_{n \times n} \end{bmatrix}. \quad (8)$$

Then the update equation at iteration $t$ can be expressed as

$$W^{(t+1)} = W^{(t)} + \eta H_{(i_t, j_t)} W^{(t)} \quad (9)$$

(see Appendix A.3). The update equation expressed in Equation (9) is noticeably similar to a stochastic power iteration. However, there is a key difference — the entries of the matrix $H_{(i_t, j_t)}$ are nonlinear functions of the entries of $X$ and $Y$. Therefore, the entries of $H_{(i_t, j_t)}$ change each iteration, making this update challenging to analyze.

To analyze this nonlinear update, we show that the matrix $H_{(i,j)}$ can be "linearized" by expressing it as the sum of a linear term and a nonlinear term that depends on $X$ and $Y$.

**Lemma 3.1.** *Given a co-occurrence matrix $C$ and node and context embedding matrices $X$ and $Y$, suppose $H_{(i,j)}$ is defined as in Equation (8). Let's define matrices $\widetilde{L}_{(i,j)} \in$*

$\mathbb{R}^{n \times n}$ *and $\widetilde{E}_{(i,j)} \in \mathbb{R}^{n \times n}$ as*

$$\widetilde{L}_{(i,j)} = \frac{1}{2} e_i (e_j^\top - \frac{s_n}{|C|} \mathbf{1}^\top C),$$
$$\widetilde{E}_{(i,j)} = e_i e_i^\top (\frac{1}{2} J - \sigma(XY^\top)) e_j e_j^\top$$
$$+ \frac{s_n}{|C|} e_i e_i^\top ((\mathbf{1}\mathbf{1}^\top C) \odot (\frac{1}{2} J - \sigma(XY^\top))),$$

*where $s_n$ is the number of negative samples. If we let $L_{(i,j)}$ and $E_{(i,j)}$ be the block matrices*

$$L_{(i,j)} = \begin{bmatrix} 0_{n \times n} & \widetilde{L}_{(i,j)} \\ \widetilde{L}_{(i,j)}^\top & 0_{n \times n} \end{bmatrix},$$
$$E_{(i,j)} = \begin{bmatrix} 0_{n \times n} & \widetilde{E}_{(i,j)} \\ \widetilde{E}_{(i,j)}^\top & 0_{n \times n} \end{bmatrix},$$

*then $H_{(i,j)} = L_{(i,j)} + E_{(i,j)}$.*

The proof results from observation that $\sigma(XY^\top) = \frac{1}{2} J - \frac{1}{2} J + \sigma(XY^\top)$ and some simple algebra. It can be found in detail in Appendix B.1.

From Lemma 3.1, we see that the update equation can be expressed as the sum of linear and nonlinear terms.

$$W^{(t+1)} = \underbrace{(I + \eta L_{(i_t, j_t)}) W^{(t)}}_{\text{linear}} + \eta \underbrace{E_{(i_t, j_t)} W^{(t)}}_{\text{nonlinear}}. \quad (10)$$

*Remark* 3.2. The linear update matrix $L_{(i_t, j_t)}$ has a clear connection to the diffusion dynamics of the graph. Let $L = \mathbb{E}[L_{(i_t, j_t)}]$ be the expectation of the linear portion. When the number of negative samples $s_n = 1$, the off-diagonal blocks of $L$ simplify to a scalar multiple of the matrix $C - \frac{1}{|C|}(C^\top \mathbf{1}\mathbf{1}^\top C)$. Recalling from Definition 2.1 that $C$ is a weigthed-sum of transition matrices $\frac{1}{2m} D_A P^s$, we can see that $L$ is a weighted-sum of autocovariance matrices across time scales $s \in \{1, \ldots, S\}$.

**Analysis intuition.** From the split above, we observe that when the size of the node and context embeddings $X^{(t)}$ and $Y^{(t)}$ are small, the matrix $E_{(i_t, j_t)} W^{(t)}$ is also quite small, and thus the update equation in Equation (10) can be approximated by the linear update

$$W^{(t+1)} \approx (I + \eta L_{(i_t, j_t)}) W^{(t)}. \quad (11)$$

If this approximation holds, we can leverage methods for analyzing stochastic PCA (in which one encounters a similar update) to conclude that in expectation,

$$W^{(t)} \approx (I + \eta L)^t W^{(0)}, \quad (12)$$

where $L = \mathbb{E}[L_{(i_t, j_t)}]$. In other words, the update mimics power iteration on the expectation of the linear portion.

---

**Algorithm 1** DeepWalk via SGNS

---

**Input:** Graph $\mathcal{G}$, initialization radius $\tau_0$, negative sampling parameter $s_n$, learning rate $\eta$; parameters $M, T$ as specified later.

**Initialize:** $\boldsymbol{W}^{(0)} \in \mathbb{R}^{2n \times d}$ with $W_{ij}^{(0)} \sim \mathcal{N}\left(0, \frac{\tau_0^2}{d \log n}\right)$, $\forall i \in [2n]$ and $j \in [d]$.

**for** $t = 0, 1, 2, ..., MT$ **do**

    Sample $(i_t, j_t)$ using random walks as described.

    $\boldsymbol{W}^{(t+1)} = \boldsymbol{W}^{(t)} + \eta \boldsymbol{H}_{(i_t, j_t)} \boldsymbol{W}^{(t)}$.

**end for**

**Return** $\boldsymbol{W}^{(MT)}$.

---

However, the key challenge is that the stochastic updates are very "sparse", and they have a much higher variance (about an $n$ factor more) than the expected update, which is well-spread. This requires us to choose a small learning rate $\eta$ and perform $\approx n/\eta$ iterations. We prove that while vector lengths can change substantially in each step – around $1 + \eta$ in certain directions – the increase over $n/\eta$ steps can still be bounded by $O(1)$.

Algorithm 1 formally presents DeepWalk using Skip-gram with negative sampling. As noted before, sampling $(i, j)$ using random walks is equivalent to sampling $(i, j)$ with probability $\frac{C_{ij}}{|\boldsymbol{C}|}$. We use the latter in our analysis. $d$ is the target embedding dimension, which is typically a constant. We will *analyze the algorithm in epochs*, where we have $M$ epochs of length $T$ each.

The remainder of the paper is organized as follows. In Section 4, we analyze the evolution of the embeddings $\boldsymbol{W}^{(t)}$. Recall that the main update can be decomposed as in Equation (10). For convenience, we will write

$$\boldsymbol{L} = \mathbb{E}\left[\boldsymbol{L}_{(i,j)}\right], \quad \boldsymbol{M}_t = \mathbb{E}\left[\boldsymbol{E}_{(i,j)} \mid \boldsymbol{W}^{(t)}\right]. \quad (13)$$

Also, denote the eigenvalues and eigenvectors of $\boldsymbol{L}$ by $\{\sigma_i, \boldsymbol{v}_i\}_{1 \leq i \leq 2n}$ respectively. One important observation will be that when the co-occurrence matrix $\boldsymbol{C}$ is obtained from random walks from a graph sampled from a $K$-block SBM, the matrix $\boldsymbol{L}$ is approximately low rank (more precisely, the spectral gap $\Delta_{2K} := \sigma_{2K} - \sigma_{2K+1}$ will be large). In Section 4, we show that the majority of the mass of $\boldsymbol{W}$ lies in the space spanned by the top $2K$ eigenvectors of $\boldsymbol{L}$. We then use this to prove that the embedding vectors capture the hidden community structure, approximately (Section 5).

## 4. Convergence to a Low-Rank Matrix

Our first result shows that when Algorithm 1 terminates on graphs drawn from the SBM, most of the mass of $\boldsymbol{W}$ lies in the space spanned by the top $2K$ eigenvectors of $\boldsymbol{L}$, under the condition that the embeddings have a sufficiently small initialization. This result is stated formally, along with the

required parameter conditions, in Theorem 4.1.

**Theorem 4.1.** *Suppose that* $\mathcal{G} \sim SBM(n, K, p, q)$ *with* $p > 1.1q \geq n^{\rho-1}$ *and* $\rho \in (0, 1)$, *and that* $\boldsymbol{C}$ *is the co-occurrence matrix of* $\mathcal{G}$ *as defined in Definition 2.1. Fix the confidence parameter* $\delta \in (0, 1)$ *and define* $b = \frac{n \Delta_{2K}}{6(1+s_n)^2}$. *Set the learning rate and the initial embedding radius to*

$$\tau_0^2 \leq \frac{a_0 b}{24 n^{5/b + 3/2}} \text{ and } \eta \leq \frac{a_0^2 b^2}{36 n^2 \ln(4/\delta)},$$

*where* $a_0 = 1/n$. *Suppose that* $\boldsymbol{W}^{(0)}$ *is initialized so that the following hold:*

*(a)* $\frac{\|\boldsymbol{V} \boldsymbol{V}^\top \boldsymbol{W}^{(0)}\|_F^2}{\|\boldsymbol{W}^{(0)}\|_F^2} \geq a_0$, *where the columns of* $\boldsymbol{V}$ *are* $\{\boldsymbol{v}_i\}_{i \leq 2K}$.

*(b)* $\|\boldsymbol{W}^{(0)}\|_{2,\infty}^2 \leq \tau_0^2$.

*Then if Algorithm 1 runs with* $M = O(\log n)$ *epochs of length* $T = \frac{1}{6(1+s_n)^2} \cdot \frac{n}{\eta}$, *then* $\exists$ *a constant* $C$ *such that*

$$\frac{\|\boldsymbol{V} \boldsymbol{V}^\top \boldsymbol{W}^{(MT)}\|_F^2}{\|\boldsymbol{W}^{(MT)}\|_F^2} \geq 1 - \frac{C}{n^{3/2}}$$

*with probability at least* $1 - MT\delta$.

We now provide a rough sketch of the proof of Theorem 4.1. The analysis proceeds as follows:

1. We define a potential function $\psi_j^t$ that measures the fraction of the embeddings' mass that is orthogonal to the top $2K$-dimensional singular vectors of the expectation of the linear portion $\boldsymbol{L} = \mathbb{E}\left[\boldsymbol{L}_{(i,j)}\right]$.

2. We analyze the expected drop in the potential function in a single iteration (Lemma 4.2).

3. We show that potential drops by a sufficient amount over the course of an epoch with high probability (Lemma 4.4). We discuss the challenges in analyzing an epoch and provide a technical lemma that we utilize to address those challenges (Lemma 4.3).

4. We analyze the entire algorithm run, thereby proving Theorem 4.1.

We focus on the potential function defined as follows. Let $\boldsymbol{V}$ be the $2n \times 2K$ matrix whose columns are $\{\boldsymbol{v}_i\}_{i \leq 2K}$.

$$\Psi_t := 1 - \frac{\|\boldsymbol{V} \boldsymbol{V}^\top \boldsymbol{W}^{(t)}\|_F^2}{\|\boldsymbol{W}^{(t)}\|_F^2}, \quad (14)$$

Note that this measures the fraction of mass of the columns of $\boldsymbol{W}^{(t)}$ that are orthogonal to the span of the top $2K$ singular vectors. By definition, $\Psi_t \in [0, 1]$. Throughout our

analysis, however, we analyze the progress of a single column $\boldsymbol{W}_{:j}$ of the matrix $\boldsymbol{W}$ by tracking the progress of the potential

$$\psi_j^t := 1 - \frac{\|\boldsymbol{V}\boldsymbol{V}^\top \boldsymbol{W}_{:j}^{(t)}\|^2}{\|\boldsymbol{W}_{:j}^{(t)}\|^2}. \tag{15}$$

This measures the fraction of mass of the column $\boldsymbol{W}_{:j}^{(t)}$ that is orthogonal to the span of the top $2K$ singular vectors. Note that showing convergence of $\psi_j^t$ and then using a union bound over all columns implies the convergence of $\Psi_t$ since

$$\Psi_t = 1 - \frac{\|\boldsymbol{V}\boldsymbol{V}^\top \boldsymbol{W}^{(t)}\|_F^2}{\|\boldsymbol{W}^{(t)}\|_F^2} = \frac{\sum_j \|(\boldsymbol{I} - \boldsymbol{V}\boldsymbol{V}^\top)\boldsymbol{W}_{:j}^{(t)}\|^2}{\sum_j \|\boldsymbol{W}_{:j}^{(t)}\|^2}$$

$$\leq \max_{j \in [d]} \psi_j^t.$$

(We used the fact that $\frac{\sum_i a_i}{\sum_i b_i} \leq \max_i \frac{a_i}{b_i}$.)

Let $\boldsymbol{w}$ be any column of $\boldsymbol{W}$. Our first lemma analyzes the expected drop in the potential

$$\psi_t = 1 - \frac{\|\boldsymbol{V}\boldsymbol{V}^\top \boldsymbol{w}^{(t)}\|^2}{\|\boldsymbol{w}^{(t)}\|^2},$$

up to appropriate error terms in a single iteration.

**Lemma 4.2.** *Let $\boldsymbol{w}^{(t)} \in \mathbb{R}^{2n}$ be a column of $\boldsymbol{W}^{(t)}$ that satisfies*

$$\frac{\|\boldsymbol{V}\boldsymbol{V}^\top \boldsymbol{w}^{(t)}\|^2}{\|\boldsymbol{w}^{(t)}\|^2} = \phi$$

*for some parameter $\phi \in (0,1)$. Consider the update $\boldsymbol{w}^{(t+1)} = (\boldsymbol{I} + \eta \boldsymbol{H}_{(i_t, j_t)})\boldsymbol{w}^{(t)}$, which is the update from Equation (9) restricted to one column. Then, assuming that $\sqrt{2}\eta \leq 1/4$, we have*

$$\mathbb{E}\left[ \frac{\|\boldsymbol{V}\boldsymbol{V}^\top \boldsymbol{w}^{(t+1)}\|^2}{\|\boldsymbol{w}^{(t+1)}\|^2} \mid \boldsymbol{W}^{(t)} \right] \geq \phi + 2\eta \Delta_{2K} \phi (1 - \phi)$$
$$- \frac{4\eta(1 + s_n)\sqrt{r\phi}}{n} \cdot \|\boldsymbol{W}^{(t)}\|_{2,\infty}^2 - 12\eta^2,$$

*where $\Delta_{2K} := \sigma_{2K}(\boldsymbol{L}) - \sigma_{2K+1}(\boldsymbol{L})$ as before, and $r$ is an appropriate stable rank term (see Lemma C.1).*

The next step of the analysis of an epoch is to show that $\psi_t$ decreases by a sufficient amount over the course of an epoch with high probability. In Lemma 4.2, we see that the error term depends on the maximum *row norm* of the embedding matrix, i.e., the maximum length of an embedding vector over the vertices. We also can see that progress can be slow if $\psi_t$ gets close to one (even with some probability), or if $\|\boldsymbol{W}^{(t)}\|_{2,\infty}^2$ grows too rapidly. Therefore, to assist us in analyzing an epoch, we show two useful lemmas: first, we tightly bound the norm of $\|\boldsymbol{W}^{(t)}\|_{2,\infty}^2$ w.h.p., and then show that $\psi_t$ remains bounded away from one for every iteration

in the epoch with high probability (see Lemmas C.4 and C.5 in Appendix C).

When proving these lemmas, we encounter a technical challenge: first, we need to ensure that $\|\boldsymbol{W}^{(t)}\|_{2,\infty}^2$ stays small enough. However, the failure probability of this event is non-zero, making it impossible to use a standard Hoeffding bound. Therefore, we need to derive a *conditional version* that is tailored to our setting. The following technical lemma upper bounds the moment generating function of $\psi_t - \psi_0$ under the condition that $\|\boldsymbol{W}^{(t)}\|_{2,\infty}^2$ is small.

**Lemma 4.3.** *Let $\tau = \|\boldsymbol{W}^{(0)}\|_{2,\infty}$ and $\mathcal{E}_1, \mathcal{E}_2, \ldots, \mathcal{E}_T$ be a sequence events such that $\mathcal{E}_t$ is the event that $\max_{j \leq t} \|\boldsymbol{W}^{(j)}\|_{2,\infty}^2 \leq c\tau^2$ for some constant $c$. For any small $\delta > 0$, if $\eta$ and $T$ are chosen so that*

$$T \cdot \frac{6\eta(1 + s_n)^2}{n} + 4\eta(1 + s_n)^2 \sqrt{2T \ln\left(\frac{1}{\delta}\right)} \leq \ln(c),$$

*then for any $\lambda > 0$ and any $t \in [T]$,*

$$\mathbb{E}[\exp(\lambda(\psi_t - \psi_0)) \mid \mathcal{E}_T]$$
$$\leq 2\exp\left( t\left( \frac{4c\lambda\eta(1 + s_n)r^{1/2}\tau^2}{n} + 12\lambda\eta^2 + 36\lambda^2\eta^2 \right) \right).$$

The final step of the analysis of an epoch is to show that $\psi_t$ decreases by a sufficient amount over the course of an epoch with high probability. The following lemma shows that if $\psi_0 < 1 - a$ and the parameters $\tau$, $\eta$, and $T$ are chosen appropriately, then at the end of the epoch $\psi_T < 1 - (1 + \Omega(1))a$ with high probability.

**Lemma 4.4.** *Let $\tau = \|\boldsymbol{W}^{(0)}\|_{2,\infty}$ and suppose that $\psi_0 < 1 - a$ for some $a \in (0, 1/2)$. Fix a small $\delta > 0$ and define the constant[3]*

$$b = \frac{n\Delta_{2k}}{6(1 + s_n)^2}.$$

*Also, suppose that*

$$\tau^2 \leq \frac{ab}{24r^{1/2}n}, \quad \eta \leq \frac{a^2b^2}{36n^2 \ln(4/\delta)}, \quad T = \frac{n}{6(1 + s_n)^2\eta}.$$

*Then with prob. at least $1 - (T + 1)\delta$, $\psi_T < \psi_0 - \frac{ab}{3}$.*

Finally, we analyze the entire run of the algorithm, thereby proving Theorem 4.1. We do this in two parts: first, we run the algorithm for $M = O(\ln n)$ epochs of length $T$ applying Lemma 4.4 repeatedly, at which point the value of $1 - \psi_t$ will have increased to a constant value. We then perform one final epoch, upon which the final value of the potential $\psi_t$ will be small. Note that when analyzing the entire algorithm run, we set the initial embedding radius $\tau_0^2$

---

[3]It is important that $b$ is actually a constant; this holds because of the right scaling for $\boldsymbol{L}$.

to be a factor of $n^{5/b}$ smaller than the value of $\tau^2$ used in Lemma 4.4, where $b$ is a constant. This is to account for the amount that $\|\boldsymbol{W}^{(t)}\|_{2,\infty}^2$ could grow over $M$ epochs.

For the full details of the analysis, please see Appendix C.

## 5. Community Recovery in Block Models

Theorem 4.1 shows that the columns of $\boldsymbol{W}$ converge to vectors that lie in the span of the top $2K$ eigenvectors of $\mathbb{E}[\boldsymbol{L}_{(i,j)}]$. Since this space is well known to be aligned with the "cluster structure" of the SBM, we can show that any column $\boldsymbol{w}$ of $\boldsymbol{W}$ can recover a $1 - o(1)$ fraction of each community. This is our second main result. For describing our results, given a vector $\boldsymbol{x} \in \mathbb{R}^n$, define $\boldsymbol{\mu}(\boldsymbol{x})$ to be the vector of "intra-cluster averages" formed from $\boldsymbol{x}$. When $\boldsymbol{x}$ is clear from the context, we will simply write $\boldsymbol{\mu}$. For any cluster $k$ and vertex $j \in \Omega_k$,

$$\mu_j = \frac{\sum_{j' \in \Omega_k} x_{j'}}{|\Omega_k|}. \tag{16}$$

In other words, we replace $x_j$ with the "average $x$ within its cluster". Since we do not know the clustering, we cannot compute $\boldsymbol{\mu}(\boldsymbol{x})$, it is used only for analysis. We will also abuse notation slightly and write, for a cluster $i \in [K]$,

$$\mu_{\Omega_i} := \frac{\sum_{j \in \Omega_i} x_j}{|\Omega_i|}. \tag{17}$$

**Theorem 5.1.** *Suppose that $\mathcal{G} \sim SBM(n, K, p, q)$ where $p > 1.1q \geq n^{\rho-1}$ and $\rho \in (0, 1)$ and let $\boldsymbol{w}$ be a column of the embedding produced by Algorithm 1 after $M = O(\ln n)$ epochs of length $T = \frac{n}{10\eta\Delta_{2K}}$, and $s_n = 1$. Let $\boldsymbol{x} \in \mathbb{R}^n$ be the node embedding formed by the first $n$ entries of $\boldsymbol{w}$. Then the following hold with probability $\geq 0.95$ (over the initialization and the randomness in the algorithm):*

1. *For distinct clusters $i, j \in [K]$ and $i \neq j$,*

$$|\mu_{\Omega_i} - \mu_{\Omega_j}| \geq \frac{1}{60K^2\sqrt{n}}\|\boldsymbol{w}\|.$$

2. *Average distance between $x_j$ and its corresponding "cluster center" $\mu_j$ (see Equation (17)) is small:*

$$\frac{1}{n}\sum_j |x_j - \mu_j|^2 \leq \frac{c\|\boldsymbol{w}\|^2}{n^{1+\rho}}.$$

The theorem implies that the average distance between $x_j$ and the intra-cluster average $\mu_j$ is around $\frac{\|w\|}{\sqrt{n}} \cdot n^{-\rho/2}$, considerably smaller than the separation between $\mu$ values for different clusters, assuming $K^2 \ll n^{\rho/2}$. This separation structure easily implies that for $\rho > 0$ and $K$ constant, each cluster can be recovered to $(1 - o(1))$ error. Such a statement was also proved in Davison et al. (2024) (Theorem 3);

for completeness, we give a simple proof that holds even in 1D, in Appendix F.3.

Part (1) of Theorem 5.1 is proved (see Lemma E.4) by first showing sufficient "initial separation" between cluster means, using the randomness in initialization, and then showing that the separation persists. This is done by using concentration inequalities for vector-valued martingales to prove that $\boldsymbol{z}^{(t)} = \boldsymbol{V}\boldsymbol{V}^\top\boldsymbol{w}^{(t)}$ remains close to the "linearized" update $(\boldsymbol{I} + \eta\boldsymbol{L})^T\boldsymbol{z}^{(0)}$. This turns out to imply the desired separation between the cluster means.

Part (2) of Theorem 5.1 is a consequence of Lemma E.1 in the Appendix. The idea is to first argue using Theorem 4.1 that $\boldsymbol{w}$ aligns with the top eigenspace of $\boldsymbol{L}$. Because of well-known results, one can show that this space aligns with the ground-truth cluster structure. This then implies the required statement about $\boldsymbol{x}$.

We prove Theorem 5.1 using two lemmas. The first one leverages the low-rank structure shown earlier to obtain part (2) of the theorem. The second lemma proves part (1), on separation between clusters.

In Appendix Section E, we utilize Lemmas Theorem E.1 and Theorem E.4 to complete the proof of Theorem Theorem 5.1.

## 6. Experiments

In this section, we use experiments to explore two aspects of our theory: (i) the dependence of the recovery guarantees on the negative sampling parameter $s_n$, and (ii) the constraints on the length of the initialization vectors. Below, we focus primarily on (i). For (ii), we observe that empirically, convergence occurs for a much larger range of initializations (Appendix G.2). It is an interesting open question to extend our results to larger initializations; in this case the linear approximations fail to hold, so we need novel ideas.

Let us thus discuss the role of $s_n$. Note that for any (constant) $s_n$, Theorem 4.1 shows that each column of the embedding (denoted vector $\boldsymbol{w}$ in the analysis) converges to a vector that lies in the span of the top $2K$ singular vectors of $\boldsymbol{L} = \mathbb{E}\left[\boldsymbol{L}_{(i,j)}\right]$ (see Equation (13)). However, the clustering guarantees of Theorem 5.1 only hold when $s_n = 1$. The reason is as follows: the top $2K$ singular values contain the "uninformative" all-ones eigenvector, with eigenvalue $\lambda_1 = \frac{1}{2n}(1 - s_n)$, and the other "informative" eigenvectors (of the form $\boldsymbol{1}_{\Omega_i} - \boldsymbol{1}_{\Omega_j}$ for different clusters $i, j$) with eigenvalue $\sim 1/n$ (see the proof of Lemma C.1 in Appendix D). When $s_n = 1$, $\lambda_1 = 0$, so the embedding mostly falls into the *informative* subspace (denoted $\Pi$ in what follows). Due to this, the embeddings clearly reveal the cluster structure. But for larger $s_n$, the all-ones vector term dominates, due to which we no longer see a sufficient separation of clusters

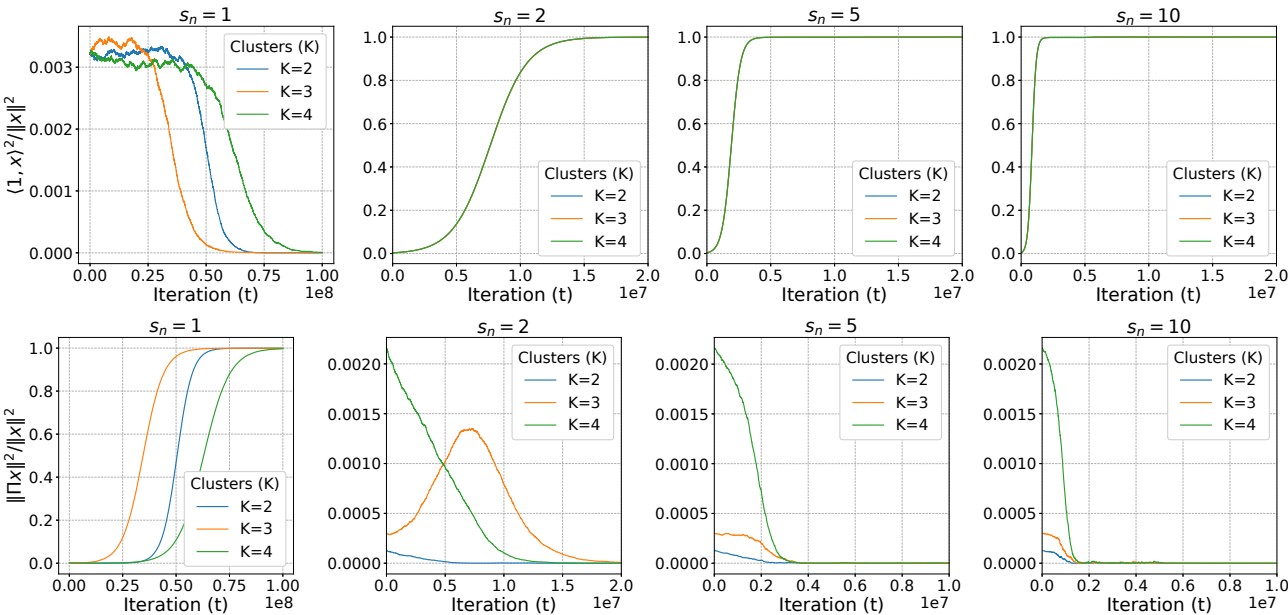

*Figure 1.* Convergence and divergence of a one-dimensional node embedding $x$ throughout training via SGNS to (top) the all-ones vector $u_1 = 1$, and (bottom) the space spanned by the ground-truth vectors $u_2 = 1_{\Omega_1} - 1_{\Omega_2}$, $u_3 = 1_{\Omega_2} - 1_{\Omega_3}$ and $u_4 = 1_{\Omega_3} - 1_{\Omega_4}$ for an increasing value of the negative sampling parameter. **Note:** The curves for $K = 2, 3, 4$ nearly overlap for the top right figures.

(especially compared to the *norm* of $w$).

We observe this quite clearly in experiments on the SBM, for a range of parameters $n, K, s_n$. Figure 1 shows results from an SBM graph with $n = 1200$ nodes and $K = 2, 3, 4$ communities using parameters $p = 0.6$ and $q = 0.1$, for a one-dimensional embedding obtained using Algorithm 1. The figure shows the fraction of the mass of $x$ that lies on $\Pi$, as well as cosine between $x$ and the all-ones vector, throughout training for increasing values of $s_n$. As predicted, the $x$-embedding aligns with the informative space $\Pi$ when $s_n = 1$, and aligns with $u_1$ as $s_n$ increases. In Appendix G, we show experiments using different $n$ values. Further, we show the resulting $x$-embeddings with their labeled ground-truth communities, showing that DeepWalk with SGNS and $s_n = 1$ can recover communities even with one-dimensional embeddings! The experiments suggest that smaller values of $s_n$ may be more conducive for community recovery, and supports prior works that suggest that an embedding's performance on downstream tasks may be negatively impacted if the number of negative samples is too large (e.g., see Saunshi et al. (2019)). Other aspects of how $s_n$ affects convergence are also discussed in Appendix G.

## 7. Conclusion

In this work, we provide a rigorous theoretical analysis of the DeepWalk algorithm with negative sampling. We view the algorithm as online gradient descent with non-convex objectives at each step, and study the trajectory of the embedding vectors, starting with random small-norm initialization.

For graphs drawn from a $K$-block stochastic block model (SBM) with sufficient separation between edge probabilities, we showed that the obtained embeddings (a) have $(1 - o(1))$ mass along the $2K$-dimensional space spanned by the top eigenvectors of the graph, and (b) reveal the cluster structure of the SBM with $1 - o(1)$ error. Our analysis leaves open two interesting questions: (i) obtaining convergence guarantees for larger norm initializations (that empirically appear to exhibit convergence), and (ii) showing convergence when the gap $p - q$ (intra- versus inter- cluster edge probability) of the SBM is smaller, potentially matching the recovery guarantees for spectral methods.

## Acknowledgements

The authors are supported by supported by the National Science Foundation under Grant Nos. CCF-2008688 and CCF-2047288. We also thank the ICML reviewers for their comments and suggestions on improving the presentation.

## Impact Statement

This paper presents work whose goal is to advance the field of Machine Learning. There are many potential societal consequences of our work, none which we feel must be specifically highlighted here. The work is primarily theoretical in nature, focusing on the optimization trajectory of a well-studied nonconvex optimization problem.

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

# A. Additional Background

## A.1. Stochastic block models

The stochastic block model is an extension of the classic Erdős-Rényi model for graphs that produces graphs with a *community* structure. It is defined using parameters $n, K \in \mathbb{N}$ such that $K|n$ along with a $K \times K$ matrix $\boldsymbol{B}$ with entries in $[0, 1]$. Given these parameters, a graph on $n$ vertices from the $K$-block stochastic block model (SBM) is generated as follows. The vertices are indexed by $[n]$ and the blocks or *communities* are indexed by $[K]$.

First, each node in the graph is assigned to a community in $[K]$ as described below. $\Omega_k$ will denote the set of vertices belonging to community $k$. The matrix $\boldsymbol{Z} \in \{0, 1\}^{n \times K}$ will be used to denote community assignments, i.e., $\boldsymbol{Z}_{ik} = 1$ if node $i \in \Omega_k$ and $\boldsymbol{Z}_{ik} = 0$ otherwise. To assign communities to vertices, we choose $\Omega_1, \Omega_2, \ldots, \Omega_K$ to be a *uniformly random partition* of $[n]$, such that $|\Omega_k| = n/K$ for all $k \in [K]$.

Second, edges are placed between nodes pairs using the following random process. Let $\boldsymbol{B} \in \mathbb{R}^{K \times K}$ be a matrix whose entries $B_{kk'}$ denote the probability of an edge being placed between a node in community $k$ and a node in community $k'$. In the *symmetric SBM*, the diagonal entries $B_{kk} = p$ and the off-diagonal entries are assigned a value $B_{kk'} = q$. Throughout our analysis, we assume that $n^{\rho-1} < q < p$ for some $\rho \in (0, 1)$.

We denote the adjacency matrix of a matrix drawn from a $K$-block stochastic block model with parameters $p$ and $q$ by $\boldsymbol{A} \sim SBM(n, K, p, q)$. The entries of $\boldsymbol{A}$ are then independently distributed Bernoulli random variables. Let $\omega(i)$ denote the community assignment of node $i$. Then $A_{ij} \sim Ber(p)$ if $\omega(i) = \omega(j)$ and $A_{ij} \sim Ber(q)$ if $\omega(i) \neq \omega(j)$. We denote $\mathbb{E}[\boldsymbol{A}]$ by $\overline{\boldsymbol{A}}$. Indeed, $\overline{\boldsymbol{A}} = \boldsymbol{Z}\boldsymbol{B}\boldsymbol{Z}^\top$, the matrix whose entries are the expected values of their corresponding entries in $\boldsymbol{A}$.

## A.2. The DeepWalk algorithm

The DeepWalk algorithm is implemented as an iterative *online* algorithm, as described in Section 2. However for the purpose of analysis, we can view the algorithm as first constructing a co-occurrence matrix $\boldsymbol{C}$ and then sampling pairs of indices from it and then optimizing an objective function.

**Constructing Co-occurrences**    Given a graph $\mathcal{G} = (\mathcal{V}, \mathcal{E})$, the co-occurrence between two nodes $v_i, v_j \in \mathcal{V}$ is defined as in previous work (Barot et al., 2021):

*Definition* A.1. Given a graph $\mathcal{G} = (\mathcal{V}, \mathcal{E})$, suppose that $R$ random walks of length $L$ are performed on the graph. Let $w^{(r)}$ denote a path of length $L$ generated by the $r^{th}$ random walk, and let $w_l^{(r)}$ be the $l^{th}$ step of the $r^{th}$ random walk. Given a context window of size $S$, the co-occurrence between two nodes $v_i, v_j \in \mathcal{V}$ is the total number of times node $v_j$ appears within $S$ steps before or after node $v_i$ in the $R$ random walks, i.e.,

$$C'_{ij} = \sum_{r=1}^{R} \sum_{s=1}^{S} \sum_{l=1}^{L-s} \mathbb{1}\{w_l^{(r)} = i, w_{l+s}^{(r)} = j\} + \mathbb{1}\{w_l^{(r)} = j, w_{l+s}^{(r)} = i\}.$$

Furthermore, the matrix $(\boldsymbol{C}')_{ij} = C'_{ij}$ whose entries are the co-occurrences $C'_{ij}$ is called the random walk co-occurrence matrix, or simply the co-occurrence matrix of graph $\mathcal{G}$.

We can find closed forms for the "limiting" co-occurrence matrix by taking either the number of random walks $R \to \infty$ (fixing $L, S$) (Barot et al., 2021; Zhang & Tang, 2024; Harker & Bhaskara, 2023) or let the length of the random walk $L \to \infty$ (Qiu et al., 2018), or even the window size $S \to \infty$ (Chanpuriya & Musco, 2020). In our analysis we are in the former regime, where we fix the walk length and window size.

*Lemma* A.2. Given a graph $\mathcal{G} = (\mathcal{V}, \mathcal{E})$ with adjacency matrix $\boldsymbol{A}$, let $\boldsymbol{C}'$ be its co-occurrence matrix defined as in Definition A.1. Then almost surely

$$\frac{(\boldsymbol{C}')_{ij}}{R} \overset{a.s.}{\underset{R \to \infty}{\to}} 2 \sum_{s=1}^{S} (L - s) \pi_i (\boldsymbol{P}^s)_{ij},$$

where $\boldsymbol{P} = \boldsymbol{D}_{\boldsymbol{A}}^{-1} \boldsymbol{A}$ is the transition matrix of the graph and $\pi_i = \frac{d_i}{2m}$ is the stationary distribution of a random walk on the graph.

*Proof.* By Definition A.1, we have

$$
\frac{C'_{ij}}{R} = \sum_{r=1}^{R}\sum_{s=1}^{S}\sum_{l=1}^{L-s} \left[ \frac{\mathbb{1}\{w_l^{(r)}=i, w_{l+s}^{(r)}=j\}}{R} + \frac{\mathbb{1}\{w_l^{(r)}=j, w_{l+s}^{(r)}=i\}}{R} \right]
$$
$$
= \sum_{s=1}^{S}\sum_{l=1}^{L-s} \left[ \sum_{r=1}^{R}\left(\frac{\mathbb{1}\{w_l^{(r)}=i, w_{l+s}^{(r)}=j\}}{R}\right) + \sum_{r=1}^{R}\left(\frac{\mathbb{1}\{w_l^{(r)}=j, w_{l+s}^{(r)}=i\}}{R}\right) \right].
$$

By the strong law of large numbers, we have almost surely

$$
\sum_{r=1}^{R}\left(\frac{\mathbb{1}\{w_l^{(r)}=i, w_{l+s}^{(r)}=j\}}{R}\right) \underset{R\to\infty}{\overset{a.s.}{\to}} \Pr[w_l=i, w_{l+s}=j].
$$

It follows that

$$
\frac{C'_{ij}}{R} \underset{R\to\infty}{\overset{a.s.}{\to}} \sum_{s=1}^{S}\sum_{l=1}^{L-s} \left( \Pr[w_l=i, w_{l+s}=j] + \Pr[w_l=j, w_{l+s}=i] \right)
$$
$$
= \sum_{s=1}^{S}(L-s)\left( \Pr[w_{s+1}=j \mid w_1=i]\Pr[w_1=i] + \Pr[w_{s+1}=i \mid w_1=j]\Pr[w_1=j] \right)
$$
$$
= \sum_{s=1}^{S}(L-s)\left( (\boldsymbol{P}^s)_{ij}\frac{d_i}{2m} + (\boldsymbol{P}^s)_{ji}\frac{d_j}{2m} \right)
$$
$$
= 2\sum_{s=1}^{S}(L-s)\pi_i(\boldsymbol{P}^s)_{ij},
$$

where the last step is due to the fact that the stationary distribution $\boldsymbol{\pi}$ satisfies the detailed balance condition, i.e., $\pi_i \boldsymbol{P}_{ij} = \pi_j \boldsymbol{P}_{ji}$. This concludes the proof. $\qquad\square$

Using Lemma A.2, we obtain the following equivalence that we use in our analysis.

*Corollary* A.3. *Given a graph $\mathcal{G} = (\mathcal{V}, \mathcal{E})$ and integer parameters $L \geq S$, the following two processes are equivalent:*

*(i) Define $\boldsymbol{C}$ as*

$$
\boldsymbol{C} = 2\sum_{s=1}^{S}\frac{(L-s)}{2m}\boldsymbol{D_A}\boldsymbol{P}^s,
$$

*where $\boldsymbol{P} = \boldsymbol{D_A}^{-1}\boldsymbol{A}$ is the transition matrix of the graph and $m$ is the number edges; sample $(i,j)$ with probability $\frac{C_{ij}}{|C|}$.*

*(ii) Perform a random walk of length $L$ starting with a node $u$ sampled from the stationary distribution $\boldsymbol{\pi}$ on $\mathcal{G}$, choose two vertices at random from a random sliding window of length $S$.*

**Computing Embeddings** The DeepWalk algorithm takes a co-occurrence matrix $\boldsymbol{C}$ as input and computes two $d$-dimensional embeddings $\boldsymbol{x}_i, \boldsymbol{y}_i \in \mathbb{R}^d$ for each node $v_i \in \mathcal{V}$. The vectors $\boldsymbol{x}_i$ and $\boldsymbol{y}_i$ are referred to as the node and context embedding of node $v_i$, respectively. They are computed by an iterative minimization of the DeepWalk objective. The optimization steps are discussed in sufficient depth in the main body, Section 2.

### A.3. Derivation of Gradients

Recall from Equation (4), the objective function for a pair of randomly sampled nodes $(i, j)$ evaluated at $\boldsymbol{X}$ and $\boldsymbol{Y}$, and given an input co-occurrence matrix $\boldsymbol{C}$ is

$$
\ell_{(i,j)}(\boldsymbol{X}, \boldsymbol{Y}; \boldsymbol{C}) = -\log\sigma(\langle\boldsymbol{x}_i, \boldsymbol{y}_j\rangle) - s_n\sum_{k=1}^{n}\frac{|C_{:k}|}{|C|}\log\sigma(-\langle\boldsymbol{x}_i, \boldsymbol{y}_k\rangle). \tag{18}
$$

The partial derivatives with respect to the node and context embeddings $\boldsymbol{x}_{i'}$ and $\boldsymbol{y}_{j'}$ are

$$\nabla_{\boldsymbol{x}_{i'}}\ell_{(i,j)}(\boldsymbol{X},\boldsymbol{Y};\boldsymbol{C}) = \delta_{i'i}(-\sigma(-\langle\boldsymbol{x}_{i'},\boldsymbol{y}_j\rangle)\boldsymbol{y}_j + s_n\sum_{k=1}^{n}\frac{|\boldsymbol{C}_{:k}|}{|\boldsymbol{C}|}\sigma(\langle\boldsymbol{x}_{i'},\boldsymbol{y}_k\rangle)\boldsymbol{y}_k),$$

$$\nabla_{\boldsymbol{y}_{j'}}\ell_{(i,j)}(\boldsymbol{X},\boldsymbol{Y};\boldsymbol{C}) = \delta_{j'j}(-\sigma(-\langle\boldsymbol{x}_i,\boldsymbol{y}_{j'}\rangle)\boldsymbol{x}_i + s_n\frac{|\boldsymbol{C}_{:j'}|}{|\boldsymbol{C}|}\sigma(\langle\boldsymbol{x}_i,\boldsymbol{y}_{j'}\rangle)\boldsymbol{x}_i. \tag{19}$$

We can write the gradient in matrix form.

$$\nabla_{\boldsymbol{X}}\ell_{(i,j)}(\boldsymbol{X},\boldsymbol{Y};\boldsymbol{C}) = -\boldsymbol{e}_i\boldsymbol{e}_i^\top\sigma(-\boldsymbol{X}\boldsymbol{Y}^\top)\boldsymbol{e}_j\boldsymbol{e}_j^\top\boldsymbol{Y} + \frac{s_n}{|\boldsymbol{C}|}\boldsymbol{e}_i\boldsymbol{e}_i^\top((\boldsymbol{1}\boldsymbol{1}^\top\boldsymbol{C})\odot\sigma(\boldsymbol{X}\boldsymbol{Y}^\top))\boldsymbol{Y},$$

$$\nabla_{\boldsymbol{Y}}\ell_{(i,j)}(\boldsymbol{X},\boldsymbol{Y};\boldsymbol{C}) = -\boldsymbol{e}_j\boldsymbol{e}_j^\top\sigma(-\boldsymbol{Y}\boldsymbol{X}^\top)\boldsymbol{e}_i\boldsymbol{e}_i^\top\boldsymbol{X} + \frac{s_n}{|\boldsymbol{C}|}((\boldsymbol{1}\boldsymbol{1}^\top\boldsymbol{C})\odot\sigma(\boldsymbol{X}\boldsymbol{Y}^\top))^\top\boldsymbol{e}_i\boldsymbol{e}_i^\top\boldsymbol{X}. \tag{20}$$

We can further simplify the expression for the gradient. For a sampled $(i,j)$, we define the operator $\widetilde{\boldsymbol{H}}_{(i,j)}(\boldsymbol{X},\boldsymbol{Y};\boldsymbol{C})$ as

$$\widetilde{\boldsymbol{H}}_{(i,j)}(\boldsymbol{X},\boldsymbol{Y};\boldsymbol{C}) = \boldsymbol{e}_i\boldsymbol{e}_i^\top\sigma(-\boldsymbol{X}\boldsymbol{Y}^\top)\boldsymbol{e}_j\boldsymbol{e}_j^\top - \frac{s_n}{|\boldsymbol{C}|}\boldsymbol{e}_i\boldsymbol{e}_i^\top((\boldsymbol{1}\boldsymbol{1}^\top\boldsymbol{C})\odot\sigma(\boldsymbol{X}\boldsymbol{Y}^\top)).$$

Therefore, the negative gradient can be expressed as

$$-\nabla\ell_{(i,j)}(\boldsymbol{X},\boldsymbol{Y};\boldsymbol{C}) = \begin{bmatrix} -\nabla_{\boldsymbol{X}}\ell_{(i,j)} \\ -\nabla_{\boldsymbol{Y}}\ell_{(i,j)} \end{bmatrix} = \begin{bmatrix} \boldsymbol{0} & \widetilde{\boldsymbol{H}}_{(i,j)}(\boldsymbol{X},\boldsymbol{Y};\boldsymbol{C}) \\ \widetilde{\boldsymbol{H}}_{(i,j)}(\boldsymbol{X},\boldsymbol{Y};\boldsymbol{C})^\top & \boldsymbol{0} \end{bmatrix} \begin{bmatrix} \boldsymbol{X} \\ \boldsymbol{Y} \end{bmatrix}. \tag{21}$$

## B. Proofs from Section 3

### B.1. Proof of Lemma 3.1

The proof is the result of a simple observation that $\sigma(\boldsymbol{X}\boldsymbol{Y}^\top) = \frac{1}{2}\boldsymbol{J} - \frac{1}{2}\boldsymbol{J} + \sigma(\boldsymbol{X}\boldsymbol{Y}^\top)$ and some simple algebra.

The result follows from some simple calculations. By Equation (7),

$$\widetilde{\boldsymbol{H}}_{(i,j)}(\boldsymbol{X},\boldsymbol{Y};\boldsymbol{C}) = \boldsymbol{e}_i\boldsymbol{e}_i^\top\sigma(-\boldsymbol{X}\boldsymbol{Y}^\top)\boldsymbol{e}_j\boldsymbol{e}_j^\top - \frac{s_n}{|\boldsymbol{C}|}\boldsymbol{e}_i\boldsymbol{e}_i^\top((\boldsymbol{1}\boldsymbol{1}^\top\boldsymbol{C})\odot\sigma(\boldsymbol{X}\boldsymbol{Y}^\top))$$

$$= \boldsymbol{e}_i\boldsymbol{e}_i^\top(\frac{1}{2}\boldsymbol{J} - \frac{1}{2}\boldsymbol{J} + \sigma(-\boldsymbol{X}\boldsymbol{Y}^\top))\boldsymbol{e}_j\boldsymbol{e}_j^\top$$
$$\qquad - \frac{s_n}{|\boldsymbol{C}|}\boldsymbol{e}_i\boldsymbol{e}_i^\top((\boldsymbol{1}\boldsymbol{1}^\top\boldsymbol{C})\odot(\frac{1}{2}\boldsymbol{J} - \frac{1}{2}\boldsymbol{J} + \sigma(\boldsymbol{X}\boldsymbol{Y}^\top)))$$

$$= \frac{1}{2}\boldsymbol{e}_i\boldsymbol{e}_j^\top + \boldsymbol{e}_i\boldsymbol{e}_i^\top(\frac{1}{2}\boldsymbol{J} - \sigma(\boldsymbol{X}\boldsymbol{Y}^\top))\boldsymbol{e}_j\boldsymbol{e}_j^\top - \frac{s_n}{2|\boldsymbol{C}|}\boldsymbol{e}_i\boldsymbol{e}_i^\top(\boldsymbol{1}\boldsymbol{1}^\top\boldsymbol{C})$$
$$\qquad + \frac{s_n}{|\boldsymbol{C}|}\boldsymbol{e}_i\boldsymbol{e}_i^\top((\boldsymbol{1}\boldsymbol{1}^\top\boldsymbol{C})\odot(\frac{1}{2}\boldsymbol{J} - \sigma(\boldsymbol{X}\boldsymbol{Y}^\top)))$$

$$= \frac{1}{2}\boldsymbol{e}_i\boldsymbol{e}_j^\top - \frac{s_n}{2|\boldsymbol{C}|}\boldsymbol{e}_i\boldsymbol{e}_i^\top(\boldsymbol{1}\boldsymbol{1}^\top\boldsymbol{C}) + \boldsymbol{e}_i\boldsymbol{e}_i^\top(\frac{1}{2}\boldsymbol{J} - \sigma(\boldsymbol{X}\boldsymbol{Y}^\top))\boldsymbol{e}_j\boldsymbol{e}_j^\top$$
$$\qquad + \frac{s_n}{|\boldsymbol{C}|}\boldsymbol{e}_i\boldsymbol{e}_i^\top((\boldsymbol{1}\boldsymbol{1}^\top\boldsymbol{C})\odot(\frac{1}{2}\boldsymbol{J} - \sigma(\boldsymbol{X}\boldsymbol{Y}^\top)))$$

$$= \frac{1}{2}\boldsymbol{e}_i(\boldsymbol{e}_j^\top - \frac{s_n}{|\boldsymbol{C}|}\boldsymbol{1}^\top\boldsymbol{C}) + \boldsymbol{e}_i\boldsymbol{e}_i^\top(\frac{1}{2}\boldsymbol{J} - \sigma(\boldsymbol{X}\boldsymbol{Y}^\top))\boldsymbol{e}_j\boldsymbol{e}_j^\top$$
$$\qquad + \frac{s_n}{|\boldsymbol{C}|}\boldsymbol{e}_i\boldsymbol{e}_i^\top((\boldsymbol{1}\boldsymbol{1}^\top\boldsymbol{C})\odot(\frac{1}{2}\boldsymbol{J} - \sigma(\boldsymbol{X}\boldsymbol{Y}^\top)))$$

$$= \widetilde{\boldsymbol{L}}_{(i,j)} + \widetilde{\boldsymbol{E}}_{(i,j)}.$$

This concludes the proof.

## C. Analysis: Convergence to a Low Rank Matrix

Our goal in this section is to prove that for each of the columns of $W^{(t)}$, most of the mass ends up on the span of the top $2K$ singular vectors of $L$. Formally, we focus on the potential function defined as follows. Let $V$ be the $2n \times 2K$ matrix whose columns are $\{v_i\}_{i \leq 2K}$.

$$\Psi_t := 1 - \frac{\|VV^\top W^{(t)}\|_F^2}{\|W^{(t)}\|_F^2}, \tag{22}$$

Note that this measures the fraction of mass of the columns of $W^{(t)}$ that are orthogonal to the span of the top $2K$ singular vectors. By definition, $\Psi_t \in [0, 1]$. Throughout our analysis, however, we analyze the progress of a single column $W_{:j}$ of the matrix $W$ by tracking the progress of the potential

$$\psi_j^t := 1 - \frac{\|VV^\top W_{:j}^{(t)}\|^2}{\|W_{:j}^{(t)}\|^2}. \tag{23}$$

This measures the fraction of mass of the column $W_{:j}^{(t)}$ that is orthogonal to the span of the top $2K$ singular vectors. Note that showing convergence of $\psi_j^t$ and then using a union bound over all columns implies the convergence of $\Psi_t$ since

$$\Psi_t = 1 - \frac{\|VV^\top W^{(t)}\|_F^2}{\|W^{(t)}\|_F^2} = \frac{\sum_j \|(I - VV^\top)W_{:j}^{(t)}\|^2}{\sum_j \|W_{:j}^{(t)}\|^2} \leq \max_{j \in [d]} \psi_j^t.$$

(We used the fact that $\frac{\sum_i a_i}{\sum_i b_i} \leq \max_i \frac{a_i}{b_i}$).

Throughout the argument here, we assume that the negative sampling parameter $s_n$ is a constant (does not grow with the graph size). The following lemma shows that some important properties hold with high probability for the DeepWalk updates described above, on a $K$-block SBM. The proof can be found in Appendix D.1.

*Lemma C.1. Suppose that $A \sim SBM(n, K, p, q)$ and the co-occurrence matrix $C$ is defined as in Definition 2.1. Suppose that the parameters $p, q$ satisfy $p > 1.1q > n^{\rho-1}$. Furthermore, suppose that $\overline{C}$ is the expected co-occurrence matrix defined as in Definition 2.1, using the graph's expected adjacency matrix $\overline{A}$. I.e.,*

$$\overline{C} = 2 \sum_{s=1}^{S} \frac{(L-s)}{n\overline{d}} D_{\overline{A}} \overline{P}^s,$$

*where $\overline{d}$ is the expected degree of the graph and $\overline{P} = D_{\overline{A}}^{-1} \overline{A}$. Then with probability at least $1 - n^{-5}$,*

*(a) The ratio $\|C\|_\infty / |C| \leq 2/n$.*

*(b) Let $r$ be the stable rank[4] of the matrix $\frac{1}{|C|}\left(C + \frac{s_n}{|C|}C^\top \mathbf{1}\mathbf{1}^\top C\right)$. Then for any embedding matrix $W$, we have*
$$\left\|\mathbb{E}[E_{(i,j)} \mid W]\right\| \leq (1 + s_n)r^{1/2}n^{-1} \cdot \|W\|_{2,\infty}^2.$$

*(c) For any node-context pair $(i, j)$, we have $\|H_{(i,j)}\| \leq \sqrt{2}$.*

*(d) Let $\overline{L}$ be the block matrix*
$$\overline{L} = \begin{bmatrix} 0 & L' \\ L' & 0 \end{bmatrix},$$
*where $L' = \frac{1}{2|C|}\left(\overline{C} - \frac{s_n}{|C|}\overline{C}^\top \mathbf{1}\mathbf{1}^\top \overline{C}\right)$. Then $\|L - \overline{L}\| = O\left(\frac{\log^{1/2} n}{n^{1+\rho/2}}\right)$.*

*(e) The spectral gap of $L$ is $\Delta_{2K} = \Theta(1/n)$.*

We would like to remark that while our analysis focuses on graphs drawn from stochastic block models, our analysis holds for any graph that satisfies the properties (a-c) and (e) in Lemma C.1. (Note that for general graphs, $\overline{L}$ is not well-defined.)

The remainder the analysis is broken up into three parts: first, we analyze the expected drop in the potential in a single iteration. Second, we analyze the drop in the potential over the course of an epoch. Finally, we analyze the entire algorithm run, proving Theorem 4.1.

---

[4]Recall that the stable rank of a matrix $A$ is defined as $\|A\|_F^2 / \|A\|_2^2$.

## C.1. Initialization

Here, we make a comment on the initialization assumptions in Theorem 4.1. In Algorithm 1 we initialize our $2n \times d$ embedding matrix $\boldsymbol{W}^{(0)}$ such that each entry $W_{ij}^{(0)}$ is drawn from a normal distribution with small variance, i.e., $\mathcal{N}(0, \tau_0^2/(d \log n))$. Consider a single column $\boldsymbol{v}$ of $\boldsymbol{V}$ and assume the w.l.o.g that $\boldsymbol{v} = \boldsymbol{e}_1$. Then $\langle \boldsymbol{v}, \hat{\boldsymbol{w}}^{(0)} \rangle = \frac{\boldsymbol{w}_1^2}{\sum_i \boldsymbol{w}_i^2}$. Standard concentration inequalities show that with high probability the numerator is $\Theta(1)$ and the denominator is $\Theta(2n)$. If we perform a union bound over the $2K$ columns of $\boldsymbol{V}$ and the $d$ columns of $\boldsymbol{W}^{(0)}$, assumption (a) holds with high constant probability. Similarly, $\|\boldsymbol{W}^{(0)}\|_{2,\infty}^2$ is a $\chi^2$ random variable with $d$-degrees of freedom. We can use standard concentration bounds of sub-exponential random variables to show that assumption (b) in Theorem 4.1 holds with high probability.

## C.2. Analysis of an iteration

Let $\boldsymbol{w}$ be any column of $\boldsymbol{W}$. Our first lemma analysis the expected drop in the potential

$$\psi_t = 1 - \frac{\|\boldsymbol{V}\boldsymbol{V}^\top \boldsymbol{w}^{(t)}\|^2}{\|\boldsymbol{w}^{(t)}\|^2},$$

up to appropriate error terms. The proof is deferred to Appendix D.2.1.

*Lemma C.2. Let $\boldsymbol{w}^{(t)} \in \mathbb{R}^{2n}$ be a column of $\boldsymbol{W}^{(t)}$ that satisfies*

$$\frac{\|\boldsymbol{V}\boldsymbol{V}^\top \boldsymbol{w}^{(t)}\|^2}{\|\boldsymbol{w}^{(t)}\|^2} = \phi$$

*for some parameter $\phi \in (0, 1)$. Consider the update $\boldsymbol{w}^{(t+1)} = (\boldsymbol{I} + \eta \boldsymbol{H}_{(i_t, j_t)})\boldsymbol{w}^{(t)}$, which is the update from Equation (9) restricted to one column. Then, assuming that $\sqrt{2}\eta \leq 1/4$, we have*

$$\mathbb{E}\left[\frac{\|\boldsymbol{V}\boldsymbol{V}^\top \boldsymbol{w}^{(t+1)}\|^2}{\|\boldsymbol{w}^{(t+1)}\|^2} \mid \boldsymbol{W}^{(t)}\right] \geq \phi + 2\eta\Delta_{2k}\phi(1 - \phi) - \frac{4\eta(1 + s_n)\sqrt{r\phi}}{n} \cdot \|\boldsymbol{W}^{(t)}\|_{2,\infty}^2 - 12\eta^2,$$

*where $\Delta_{2k} := \sigma_{2k}(\boldsymbol{L}) - \sigma_{2k+1}(\boldsymbol{L})$ is the gap between the $2k^{th}$ and $(2k+1)^{th}$ singular value of $\boldsymbol{L}$.*

In Lemma C.2, we see that the error term depends on the maximum *row norm* of the embedding matrix, i.e., the maximum length of an embedding vector over the vertices. The following useful lemma bounds how much the norm $\|\boldsymbol{W}^{(t)}\|_{2,\infty}^2$ grows, as well as its expectation, in a single iteration. The proof can be found in Appendix D.2.2.

*Lemma C.3. For any iteration $t$,*

$$\mathbb{E}[\|\boldsymbol{W}^{(t+1)}\|_{2,\infty}^2 \mid \boldsymbol{W}^{(t)}] \leq \left(1 + \frac{6\eta(1 + s_n)^2}{n}\right) \|\boldsymbol{W}^{(t)}\|_{2,\infty}^2,$$

*and*

$$\|\boldsymbol{W}^{(t+1)}\|_{2,\infty}^2 \leq \left(1 + 3\eta(1 + s_n)^2\right) \|\boldsymbol{W}^{(t)}\|_{2,\infty}^2.$$

## C.3. Analysis of an Epoch

Our goal will be to prove that in each epoch, if the potential $\psi$ starts at $1 - a$ at the start of the epoch, then at the end, it drops to $1 - (1 + \Omega(1))a$ with high probability. However, from Lemma C.2 we can see that progress can be slow if $\psi_t$ gets close to one (even with some probability), or if $\|\boldsymbol{W}^{(t)}\|_{2,\infty}^2$ grows too rapidly. Therefore, we show two useful lemmas, first proving a tight bound on the norm of the embedding vectors, and then we show that $\psi_t$ remains bounded away from one for all $t \in [T]$ with high probability. We will denote by $t = 0$ the start of the current epoch.

*Lemma C.4. Let $\tau = \|\boldsymbol{W}^{(0)}\|_{2,\infty}$, and let $\delta \in (0, 1)$ be a confidence parameter.*

*(a) At any time $t \geq 1$, we have that with probability at least $1 - \delta$,*

$$\|\boldsymbol{W}^{(t)}\|_{2,\infty}^2 \leq \tau^2 \cdot \exp\left(t \cdot \frac{6\eta(1 + s_n)^2}{n} + 4\eta(1 + s_n)^2 \sqrt{2t \ln\left(\frac{1}{\delta}\right)}\right).$$

*(b) Suppose that the epoch length $T$ and learning rate $\eta$ are chosen so that*

$$T \cdot \frac{6\eta(1 + s_n)^2}{n} + 4\eta(1 + s_n)^2 \sqrt{2T \ln\left(\frac{1}{\delta}\right)} \leq \ln(c)$$

*for some constant c. Then with probability at least $1 - \delta$, it holds that*

$$\max_{t \in [T]} \|\boldsymbol{W}^{(t)}\|_{2,\infty}^2 \leq c\tau^2.$$

Lemma C.4 is proved in Section D.3.1. For our argument, we can set $c = 3$, and $T$ and $\eta$ as described in the algorithm.

The following lemma shows that $\psi_t$ remains bounded away from one for all $t \in [T]$ with high probability.

*Lemma C.5. Let $\tau = \|\boldsymbol{W}^{(0)}\|_{2,\infty}$ and suppose that $\psi_0 < 1 - a$ for some $a \in (0, 1/2)$. For any small $\delta > 0$, if the epoch length $T$ and learning rate $\eta$ are chosen so that the following conditions hold:*

*(a)*

$$T \cdot \frac{6\eta(1 + s_n)^2}{n} + 4\eta(1 + s_n)^2 \sqrt{2T \ln\left(\frac{1}{\delta}\right)} \leq \ln(c),$$

*(b)*

$$T \cdot \frac{4c\eta(1 + s_n)r^{\frac{1}{2}}\tau^2}{n} + T \cdot 12\eta^2 + 12\eta\sqrt{T \ln\left(\frac{4}{\delta}\right)} \leq \frac{a}{2},$$

*where c is a constant. Then with probability at least $1 - T\delta$, it holds that*

$$\psi_t \leq 1 - \frac{a}{2}.$$

*for all $t \in [T]$.*

When proving this lemma (see Appendix D.3.2), we encounter a technical challenge: first, we need to ensure that $\|\boldsymbol{W}^{(t)}\|_{2,\infty}^2$ stays is small enough. This is ensured by Lemma C.4. However, the failure probability is still non-zero making it impossible to use a standard Hoeffding bounds. Therefore, we need to derive a *conditional version* that is tailored to our setting. The following technical lemma upper bounds the moment generating function of $\psi_t - \psi_0$ for all $t \in [T]$ under the condition that $\|\boldsymbol{W}^{(t)}\|_{2,\infty}^2$ has a small norm. The proof is deferred to Appendix D.3.3

*Lemma C.6. Let $\tau = \|\boldsymbol{W}^{(0)}\|_{2,\infty}$ and $\mathcal{E}_1, \mathcal{E}_2, \ldots, \mathcal{E}_T$ be a sequence events such that $\mathcal{E}_t$ is the event that $\max_{j \leq t} \|\boldsymbol{W}^{(j)}\|_{2,\infty}^2 \leq c\tau^2$ for some constant c. For any small $\delta > 0$, if $\eta$ and $T$ are chosen so that*

$$T \cdot \frac{6\eta(1 + s_n)^2}{n} + 4\eta(1 + s_n)^2 \sqrt{2T \ln\left(\frac{1}{\delta}\right)} \leq \ln(c),$$

*then for any $\lambda > 0$ and any $t \in [T]$,*

$$\mathbb{E}[\exp\left(\lambda(\psi_t - \psi_0)\right) \mid \mathcal{E}_T] \leq 2\exp\left(t\left(\frac{4c\lambda\eta(1 + s_n)r^{1/2}\tau^2}{n} + 12\lambda\eta^2 + 36\lambda^2\eta^2\right)\right).$$

The final step is to show that $\psi_t$ decreases by a sufficient amount over the course of an epoch with high probability. The following lemma shows that if $\psi_0 < 1 - a$ and the parameters $\tau$, $\eta$, and $T$ are chosen appropriately, then at the end of the epoch $\psi_T < 1 - (1 + \Omega(1))a$ with high probability. The proof has been deferred to Appendix D.3.4.

*Lemma C.7. Let $\tau = \|\boldsymbol{W}^{(0)}\|_{2,\infty}$ and suppose that $\psi_0 < 1 - a$ for some $a \in (0, 1/2)$. Fix a small $\delta > 0$ and define the constant*

$$b = \frac{n\Delta_{2k}}{6(1 + s_n)^2}.$$

*Also, suppose that*

$$\tau^2 \leq \frac{ab}{24r^{1/2}n}, \; \eta \leq \frac{a^2b^2}{36n^2 \ln{(4/\delta)}}, \; and \; T = \frac{1}{6(1+s_n)^2} \cdot \frac{n}{\eta}.$$

*Then with probability at least $1 - (T+1)\delta$, it holds that*

$$\psi_T < \psi_0 - \frac{ab}{3}.$$

### C.4. Proof of Theorem 4.1

We now analyze the entire run of the algorithm, thereby proving Theorem 4.1. We do this in two parts: first, we run the algorithm for $M = O(\ln n)$ epochs of length $T$, at which point the value of $1 - \psi_t$ has increased to a constant value. We then perform one final epoch, upon which the final value of the potential $\psi_t$ will be small. Note that we set $\tau_0^2$ to be a factor of $n^{5/b}$ smaller than the value of $\tau^2$ used in Lemma C.7, where $b$ is a constant. This is to account for the amount that $\|\boldsymbol{W}^{(t)}\|_{2,\infty}^2$ could grow over $M$ epochs. Let $\psi_0, \psi_1, ..., \psi_M$ denote the potential at the end of the $M^{th}$ epoch.

*Proof of Theorem 4.1.* The proof is a result of applying Lemma C.7 repeatedly, with updated values of $\tau^2$ and $a$ as the max row norm of the embedding $\boldsymbol{W}$ grows and the potential drops. As before, we analyze the progress of a single column $\boldsymbol{w}$ of $\boldsymbol{W}$.

We begin with the first epoch. Given our choice of parameters $\tau_0^2$, $\eta$, and $T$, a direct application of Lemma C.7 gives

$$\psi_1 \leq \psi_0 - \frac{a_0 b}{3} \leq 1 - a_0\left(1 + \frac{b}{3}\right)$$

with probability at least $1 - (T+1)\delta$.

At the beginning of the second epoch, we update the value of $a_1 = a_0(1 + b/3)$. Also, recall from Lemma C.4 that given our choice of $\eta$ and $T$, at the start of the second epoch $\tau_1^2 \leq 3\tau_0^2$ with high probability. Therefore, with high probability

$$\tau_1^2 \leq \frac{3a_0 b}{24r^{1/2}n^{5/b+1}} \leq \frac{a_0 b}{24r^{1/2}n} \leq \frac{a_1 b}{24r^{1/2}n},$$

and

$$\eta \leq \frac{a_0^2 b^2}{36n^2 \ln{(4/\delta)}} \leq \frac{a_1^2 b^2}{36n^2 \ln{(4/\delta)}}.$$

since $a_1 > a_0$. Therefore, the conditions of Lemma C.7 are satisfied and we can apply Lemma C.7 again with $\tau_1^2$ and $a_1$, giving

$$\psi_2 \leq \psi_1 - \frac{a_1 b}{3} \leq 1 - a_0\left(1 + \frac{b}{3}\right)^2$$

with probability at least $1 - 2(T+1)\delta$.

On the $M^{th}$ epoch, we have $a_M = a_0(1 + b/3)^M$ and embedding radius and learning rate

$$\tau_M^2 \leq \frac{3^M a_0 b}{24r^{1/2}n^{5/b+1}} \; and \; \eta \leq \frac{a_0^2 b^2}{36n^2 \ln{(4/\delta)}}.$$

Importantly, $3^M = O(n^{5/b})$ as long as the number of epochs $M \leq \frac{\ln{(n/2)}}{\ln{(1+b/3)}}$ and the conditions of Lemma C.7 are still satisfied. Therefore, after $M = \frac{\ln{(n/2)}}{\ln{(1+b/3)}}$ epochs,

$$\psi_M \leq 1 - a_0\left(1 + \frac{b}{3}\right)^M = 1 - a_0\left(1 + \frac{b}{3}\right)^{\frac{\ln{(n/2)}}{\ln{(1+b/3)}}} = \frac{1}{2}$$

with probability at least $1 - M(T+1)\delta$.

So far we have shown that after $M = O(\ln n)$ epochs the potential is $\leq 1/2$. We run the algorithm for one more epoch of size $T = \frac{n}{\eta}$. At the end of the $(M+1)^{th}$ epoch, we have with probability at least $1 - (M+1)(T+1)\delta$ that

$$\psi_{M+1} \leq (1 - T\eta\Delta_{2K})\psi_M + \frac{4e^{6(1+s_n)^2}T\eta(1+s_n)r^{1/2}\tau_M^2}{n} + 12T\eta^2 + 12\eta\sqrt{T\ln\left(\frac{4}{\delta}\right)}.$$

Given the chosen values of $T$, $\eta$, and $\tau_M^2$,

$$\psi_{M+1} \leq 4e^{6(1+s_n)^2}(1+s_n)\frac{a_0 b}{6n} + \frac{a_0^2 b^2}{3n\ln(4/\delta)} + \frac{2a_0 b}{n^{1/2}} \leq \frac{3b}{n^{3/2}}.$$

In the second inequality, we use the fact that $a_0 = 1/n$ by assumption and the assumptions that $s_n$ and $b$ are constants. The result follows from the fact that

$$\Psi_{M+1} \leq \max_{j\in[d]} \psi_j^{M+1},$$

assuming that the embedding dimension $d$ is constant. This concludes the proof. $\qquad\square$

## D. Proofs from Appendix C

### D.1. Proof of Lemma C.1

Before we begin our proof, we define the "expected" co-occurrence matrix, constructed using the expected adjacency matrix $\overline{A}$.

*Definition* D.1. Consider the expected adjacency matrix $\overline{A}$ of a graph $\mathcal{G} = (\mathcal{V}, \mathcal{E})$ drawn from a stochastic block model with $K$-blocks. The expected limiting co-occurrence matrix is defined as

$$\overline{C} = 2\sum_{s=1}^{S} \frac{(L-s)}{n\overline{d}} D_{\overline{A}}\overline{P}^s,$$

where $\overline{P} = D_{\overline{A}}^{-1}\overline{A}$ is the expected transition matrix, $\overline{d} = \frac{n}{K}p + \frac{n(K-1)}{K}q$ is the expected degree of $\mathcal{G}$, $L$ is the length of the random walk, and $S$ is the size of the context window.

Also, we make a couple simple observations.

*Observation* D.2. Let $\kappa = 2\sum_{s=1}^{S}(L-s)$. For any co-occurrence matrix $C$ defined as in Definition 2.1, the following hold:

(a) $C\mathbf{1} = \kappa\boldsymbol{\pi}$, where $\boldsymbol{\pi}$ is the stationary distribution vector of the graph.

(b) $|C| = \mathbf{1}^\top C\mathbf{1} = \kappa$.

(c) $\|C\| \leq \kappa d_{max}/(2m)$, where $d_{max}$ is the maximum degree of the graph $G$.

We now proceed to prove Lemma C.1.

*Proof of (a).* From Observation D.2, we have

$$\frac{\|C\|_\infty}{|C|} = \frac{\|C\mathbf{1}\|_\infty}{|C|} = \|\boldsymbol{\pi}\|_\infty = \frac{d_{max}}{2m}.$$

Let $\overline{C}$ be the expected limiting co-occurrence matrix defined as in Definition D.1. Then

$$\frac{\|\overline{C}\|_\infty}{|\overline{C}|} = \frac{1}{n}.$$

From standard chernoff/hoffding bounds, we have with probability $1 - n^{-5}$ that

$$\left| d_i - \overline{d} \right| = O\left( \left( \overline{d} \right)^{-1/2} \ln^{1/2} n \right), \forall i \in [n],$$ (24)

$$\left| m - \frac{n\overline{d}}{2} \right| = O\left( (n\overline{d})^{-1/2} \ln^{1/2} n \right).$$ (25)

Therefore, we have

$$\left| \frac{\|C\|_\infty}{|C|} - \frac{\|\overline{C}\|_\infty}{|\overline{C}|} \right| = \left| \frac{d_{max}}{2m} - \frac{1}{n} \right| = \left| \frac{nd_{max} - 2m}{2nm} \right| = O\left( (n\overline{d})^{-3/2} \log^{1/2} n \right).$$

which implies that $\|C\|_\infty / |C| \le 2/n$.

*Proof of (b).* First, from the definition of $E_{(i,j)}$,

$$\mathbb{E}[E_{(i,j)} \mid W] = \frac{1}{\kappa} \left( C + \frac{s_n}{\kappa} C^\top \mathbf{1}\mathbf{1}^\top C \right) \odot \left( \frac{1}{2} J - \sigma(XY^\top) \right).$$

It follows that

$$\left\| \mathbb{E}[E_{(i,j)} \mid W] \right\| \le \left\| \frac{1}{\kappa} \left( C + \frac{s_n}{\kappa} C^\top \mathbf{1}\mathbf{1}^\top C \right) \odot \left( \frac{1}{2} J - \sigma(XY^\top) \right) \right\|_F$$

$$\le \cdot \max_{ij} \left| \frac{1}{2} - \sigma(\langle x_i, y_j \rangle) \right| \cdot \left\| \frac{1}{\kappa} \left( C + \frac{s_n}{\kappa} \kappa^2 \pi\pi^\top \right) \right\|_F$$

$$\le \frac{\sqrt{r}}{4} \|W\|_{2,\infty}^2 \left\| \frac{1}{\kappa} \left( C + \frac{s_n}{\kappa} \kappa^2 \pi\pi^\top \right) \right\|$$

$$\le \frac{\sqrt{r}}{4} \|W\|_{2,\infty}^2 \left( \frac{d_{max}}{2m} + s_n \|\pi\|^2 \right).$$

We've already seen that $\frac{d_{max}}{2m} \le \frac{2}{n}$ with high probability. So

$$\|\pi\|^2 \le n \left( \frac{d_{max}}{2m} \right)^2 \le \frac{4}{n}.$$

Therefore,

$$\left\| \mathbb{E}[E_{(i,j)} \mid W] \right\| \le \frac{\sqrt{r}}{4} \|W\|_{2,\infty}^2 \left( \frac{2}{n} + s_n \frac{4}{n} \right)$$

$$\le \frac{\sqrt{r}}{4} \|W\|_{2,\infty}^2 \frac{4}{n} (1 + s_n)$$

$$= \sqrt{r} \|W\|_{2,\infty}^2 (1 + s_n) \cdot \frac{1}{n}.$$

*Proof of (c).* Recall that

$$H_{(i,j)} = \begin{bmatrix} \mathbf{0} & \widetilde{H}_{(i,j)} \\ \widetilde{H}_{(i,j)}^\top & \mathbf{0} \end{bmatrix},$$

where

$$\widetilde{H}_{(i,j)} = e_i e_i^\top \sigma(-XY^\top) e_j e_j^\top - \frac{s_n}{|C|} e_i e_i^\top \left( (\mathbf{1}\mathbf{1}^\top C) \odot \sigma(XY^\top) \right).$$

Since $\|H_{(i,j)}\|^2 = \|\widetilde{H}_{(i,j)}\|^2$, and

$$\widetilde{H}_{(i,j)} \widetilde{H}_{(i,j)}^\top = \left( \sigma^2(-\langle x_i^{(t)}, y_j^{(t)} \rangle) - 2s_n \sigma(-\langle x_i^{(t)}, y_j^{(t)} \rangle) \sigma(\langle x_i^{(t)}, y_j^{(t)} \rangle) \pi_j \right) e_i e_i^\top$$

$$+ s_n^2 \sum_{j=1}^n \pi_j^2 \sigma^2(\langle x_i^{(t)}, y_j^{(t)} \rangle) e_i e_i^\top,$$

which is a matrix with only one non-zero entry, we have,

$$\|\widetilde{\boldsymbol{H}}_{(i,j)}\|^2 = \sigma^2(-\langle \boldsymbol{x}_i^{(t)}, \boldsymbol{y}_j^{(t)} \rangle) - 2s_n\sigma(-\langle \boldsymbol{x}_i^{(t)}, \boldsymbol{y}_j^{(t)} \rangle)\sigma(\langle \boldsymbol{x}_i^{(t)}, \boldsymbol{y}_j^{(t)} \rangle)\pi_j$$
$$+ s_n^2 \sum_{j=1}^n \pi_j^2 \sigma^2(\langle \boldsymbol{x}_i^{(t)}, \boldsymbol{y}_j^{(t)} \rangle)$$
$$\leq 1 + \frac{1}{2}s_n\frac{d_{max}}{2m} + s_n^2 n \cdot \left(\frac{d_{max}}{2m}\right)^2.$$

We've already seen that $d_{max}/2m \leq 2/n$ with probability $1 - n^{-5}$. So

$$\|\widetilde{\boldsymbol{H}}_{(i,j)}\|^2 \leq 1 + \frac{1}{2}s_n\frac{2}{n} + s_n^2\frac{4}{n} \leq 1 + \frac{5s_n^2}{n} \leq 2.$$

Therefore,

$$\|\boldsymbol{H}_{(i,j)}\| = \|\widetilde{\boldsymbol{H}}_{(i,j)}\| \leq \sqrt{2}.$$

*Proof of (d).* Let $\overline{\boldsymbol{C}}$ be defined as in Definition D.1 and define

$$\overline{\boldsymbol{L}} = \begin{bmatrix} \boldsymbol{0} & \boldsymbol{L}' \\ \boldsymbol{L}' & \boldsymbol{0} \end{bmatrix},$$

where $\boldsymbol{L}' = \frac{1}{2\kappa}\left(\overline{\boldsymbol{C}} - \frac{s_n}{\kappa}\overline{\boldsymbol{C}}^\top \mathbf{1}\mathbf{1}^\top \overline{\boldsymbol{C}}\right)$. Recall that $\boldsymbol{L}$ has a similar form:

$$\boldsymbol{L} = \begin{bmatrix} \boldsymbol{0} & \widetilde{\boldsymbol{L}} \\ \widetilde{\boldsymbol{L}} & \boldsymbol{0} \end{bmatrix},$$

where $\widetilde{\boldsymbol{L}} = \frac{1}{2\kappa}\left(\boldsymbol{C} - \frac{s_n}{\kappa}\boldsymbol{C}^\top \mathbf{1}\mathbf{1}^\top \boldsymbol{C}\right)$.

Now, we focus on bounding $\|\boldsymbol{L} - \overline{\boldsymbol{L}}\|$. Since these are both off-diagonal block matrices of the form $\begin{bmatrix} \boldsymbol{0} & \boldsymbol{X} \\ \boldsymbol{X} & \boldsymbol{0} \end{bmatrix}$, the norm $\|\boldsymbol{L} - \overline{\boldsymbol{L}}\| = \|\widetilde{\boldsymbol{L}} - \boldsymbol{L}'\|$. Therefore, we focus on bounding $\|\widetilde{\boldsymbol{L}} - \boldsymbol{L}'\|$. By definition and the triangle inequality, we have

$$\|\widetilde{\boldsymbol{L}} - \boldsymbol{L}'\| = \left\|\frac{1}{2\kappa}(\boldsymbol{C} - \overline{\boldsymbol{C}}) + \frac{s_n}{2}(\overline{\boldsymbol{\pi}}\overline{\boldsymbol{\pi}}^\top - \boldsymbol{\pi}\boldsymbol{\pi}^\top)\right\|$$
$$\leq \frac{1}{2\kappa}\|\boldsymbol{C} - \overline{\boldsymbol{C}}\| + \frac{s_n}{2}\|\overline{\boldsymbol{\pi}}\overline{\boldsymbol{\pi}}^\top - \boldsymbol{\pi}\boldsymbol{\pi}^\top\|.$$

From (Harker & Bhaskara, 2024) (see Lemma 3.1), we know that

$$\|\boldsymbol{C} - \overline{\boldsymbol{C}}\| = \kappa \cdot O\left(\frac{\ln^{1/2}(n)}{n\overline{d}^{1/2}}\right) = \kappa \cdot O\left(\frac{\ln^{1/2}(n)}{n^{1+\rho/2}}\right).$$

where $\overline{d} = \frac{n}{K}p + \frac{n(K-1)}{K}q$ is the expected degree and $p > q > n^{\rho-1}$ for $\rho \in (0,1)$ and $\rho$ is not too small. Also

$$\|\overline{\boldsymbol{\pi}}\overline{\boldsymbol{\pi}}^\top - \boldsymbol{\pi}\boldsymbol{\pi}^\top\| = \|\overline{\boldsymbol{\pi}}\overline{\boldsymbol{\pi}}^\top - \overline{\boldsymbol{\pi}}\boldsymbol{\pi}^\top + \overline{\boldsymbol{\pi}}\boldsymbol{\pi}^\top - \boldsymbol{\pi}\boldsymbol{\pi}^\top\|$$
$$\leq \|\overline{\boldsymbol{\pi}} - \boldsymbol{\pi}\|\left(\frac{1}{n} + \|\boldsymbol{\pi}\|\right).$$

We bound $\|\overline{\boldsymbol{\pi}} - \boldsymbol{\pi}\|$. Recall from earlier that

$$\left|\frac{d_i}{2m} - \frac{1}{n}\right| = O\left((n\overline{d})^{-3/2}\ln^{1/2}n\right).$$

Then

$$\|\overline{\boldsymbol{\pi}} - \boldsymbol{\pi}\| = \left[ \sum_{i=1}^{n} \left( \frac{d_i}{2m} - \frac{1}{n} \right)^2 \right]^{1/2} = O\left( \frac{\ln^{1/2} n}{n^{1+3\rho/2}} \right),$$

and

$$\|\boldsymbol{\pi}\| = \left[ \sum_{i=1}^{n} \left( \frac{d_i}{2m} \right)^2 \right]^{1/2} \le 2n^{-1/2}.$$

Therefore,

$$\|\overline{\boldsymbol{\pi}}\overline{\boldsymbol{\pi}}^\top - \boldsymbol{\pi}\boldsymbol{\pi}^\top\| = O\left( \frac{\ln^{1/2} n}{n^{2+3\rho/2}} \right) + O\left( \frac{\ln^{1/2} n}{n^{3/2+3\rho/2}} \right)$$

$$= O\left( \frac{\ln^{1/2} n}{n^{3/2+3\rho/2}} \right).$$

Putting it all together, we have

$$\|\widetilde{\boldsymbol{L}} - \boldsymbol{L}'\| \le \frac{1}{2\kappa} \|\boldsymbol{C} - \overline{\boldsymbol{C}}\| + \frac{s_n}{2} \|\overline{\boldsymbol{\pi}}\overline{\boldsymbol{\pi}}^\top - \boldsymbol{\pi}\boldsymbol{\pi}^\top\|$$

$$= O\left( \frac{\ln^{1/2} n}{n^{1+\rho/2}} \right) + O\left( \frac{\ln^{1/2} n}{n^{3/2+3\rho/2}} \right)$$

$$= O\left( \frac{\ln^{1/2} n}{n^{1+\rho/2}} \right).$$

*Proof of (e).* Our goal is to bound $|\sigma_{2K}(\boldsymbol{L}) - \sigma_{2K+1}(\boldsymbol{L})|$. Notice that

$$|\sigma_{2K}(\boldsymbol{L}) - \sigma_{2K+1}(\boldsymbol{L})| \le \left|\sigma_{2K}(\boldsymbol{L}) - \sigma_{2K}(\overline{\boldsymbol{L}})\right| + \left|\sigma_{2K+1}(\boldsymbol{L}) - \sigma_{2K+1}(\overline{\boldsymbol{L}})\right|$$

$$+ \left|\sigma_{2K}(\overline{\boldsymbol{L}}) - \sigma_{2K+1}(\overline{(}\boldsymbol{L})\right|,$$

$$\le 2\|\boldsymbol{L} - \overline{\boldsymbol{L}}\| + \left|\sigma_{2K}(\overline{\boldsymbol{L}}) - \sigma_{2K+1}(\overline{\boldsymbol{L}})\right|,$$

where in the last step, we're applying Weyl's theorem. Similarly,

$$|\sigma_{2K}(\boldsymbol{L}) - \sigma_{2K+1}(\boldsymbol{L})| \ge \left|\sigma_{2K}(\overline{\boldsymbol{L}}) - \sigma_{2K+1}(\overline{\boldsymbol{L}})\right| - 2\|\boldsymbol{L} - \overline{\boldsymbol{L}}\|.$$

We know the bound on $\|\boldsymbol{L} - \overline{\boldsymbol{L}}\|$ from part (d), so the last step is to determine $|\sigma_{2K}(\overline{\boldsymbol{L}}) - \sigma_{2K+1}(\overline{\boldsymbol{L}})|$. Notice that since $\overline{\boldsymbol{C}}$ is a block matrix with values $a$ in the diagonal blocks and $b$ elsewhere, then $\boldsymbol{L}'$ is also a block matrix with values $r = \frac{a}{2\kappa} - \frac{s_n}{2n^2}$ in the diagonal blocks and $s = \frac{b}{2\kappa} - \frac{s_n}{2n^2}$ in the off-diagonal blocks. The eigenvalues of $\boldsymbol{L}'$ are

$$\lambda_i = \begin{cases} \frac{n}{K}r + \frac{n(K-1)}{K}s, & \text{one copy} \\ \frac{n}{K}(r - s), & K - 1 \text{ copies}. \end{cases}$$

The remaining eigenvalues are zero. Recalling that $\kappa = n\left( \frac{n}{K}a + \frac{n(K-1)}{K}b \right)$, we have

$$\frac{n}{K}r + \frac{n(K-1)}{K}s = \frac{1}{2n}(1 - s_n),$$

and,

$$\frac{n}{K}(r - s) = \frac{1}{2n}\left( \frac{a - b}{a + (K-1)b} \right).$$

From (Harker & Bhaskara, 2024) (see Lemma A.1), we have that

$$\frac{1}{2n}\left(\frac{a-b}{a+(K-1)b}\right) = \frac{1}{n\kappa}\left(\sum_{s=1}^{S}(L-s)\left(\frac{p-q}{p+(K-1)q}\right)^s\right).$$

Since $\overline{L}$ is an off-diagonal block matrix, its eigenvalues come in $\pm$ pairs of the singular values of $L'$, which are the magnitudes of the eigenvalues of $L'$. And if $s_n > 1$, then

$$\left|\sigma_{2K}(\overline{L}) - \sigma_{2K+1}(\overline{L})\right| = \left|\frac{1}{n\kappa}\left(\sum_{s=1}^{S}(L-s)\left(\frac{p-q}{p+(K-1)q}\right)^s\right) - 0\right| = \Theta\left(\frac{1}{n}\right).$$

Note that the last step uses the assumption that $p > 1.1q$. Therefore, it follows that

$$\left|\sigma_{2K}(L) - \sigma_{2K+1}(L)\right| = O\left(\frac{\ln^{1/2}n}{n^{1+\rho/2}}\right) + O\left(\frac{1}{n}\right) = O\left(\frac{1}{n}\right),$$

and

$$\left|\sigma_{2K}(L) - \sigma_{2K+1}(L)\right| = \Omega\left(\frac{1}{n}\right) - O\left(\frac{\ln^{1/2}n}{n^{1+\rho/2}}\right) = \Omega\left(\frac{1}{n}\right).$$

This proves the statement that $\left|\sigma_{2K}(L) - \sigma_{2K+1}(L)\right| = \Theta\left(1/n\right)$ with high probability.

### D.2. Proofs from Section C.2

#### D.2.1. PROOF OF LEMMA C.2

For convenience, let us write $H_t = H_{(i_t, j_t)}$. Let us define the ratio of interest $\phi'$ as

$$\phi' := \frac{\|VV^\top w^{(t+1)}\|^2}{\|w^{(t+1)}\|^2} = \frac{\|VV^\top(w^{(t)} + \eta H_t w^{(t)})\|^2}{\|(w^{(t)} + \eta H_t w^{(t)}\|^2}.$$

We first simplify the numerator and denominator separately. We have

$$\|VV^\top(w^{(t)} + \eta H_t w^{(t)})\|^2 \geq \|VV^\top w^{(t)}\|^2 + 2\eta\langle VV^\top w^{(t)}, VV^\top H_t w^{(t)}\rangle$$
$$= \|VV^\top w^{(t)}\|^2 + 2\eta\langle VV^\top w^{(t)}, H_t w^{(t)}\rangle,$$

where we used the fact that $VV^\top$ is a projection matrix. Next, writing $\widehat{w}^{(t)} = w^{(t)}/\|w^{(t)}\|$, we can simplify the denominator above as:

$$\|(w^{(t)} + \eta H_t w^{(t)})\|^2 = \|w^{(t)}\|^2\left[1 + 2\eta\langle\widehat{w}^{(t)}, H_t\widehat{w}^{(t)}\rangle + \eta^2\|H_t\widehat{w}^{(t)}\|^2\right]$$
$$\leq \|w^{(t)}\|^2\left[1 + 2\eta\langle\widehat{w}^{(t)}, H_t\widehat{w}^{(t)}\rangle + 2\eta^2\right].$$

Next, using our assumption that $\sqrt{2}\eta < 1/4$ together with $\frac{1}{1+x} \geq (1-x)$ for all $|x| \leq 1$, we have

$$\phi' \geq \left(\|VV^\top\widehat{w}^{(t)}\|^2 + 2\eta\langle VV^\top\widehat{w}^{(t)}, H_t\widehat{w}^{(t)}\rangle\right)\left(1 - 2\eta\langle\widehat{w}^{(t)}, H_t\widehat{w}^{(t)}\rangle - 2\eta^2\right).$$

Now by definition, $\|VV^\top\widehat{w}^{(t)}\|^2 = \phi$. Thus, the first term in the parentheses is $\leq \phi + 2\sqrt{2}\eta < 2$, by our assumption $\sqrt{2}\eta \leq 1/4$ and $\phi \in (0, 1)$. This implies that

$$\phi' \geq \left(\phi + 2\eta\langle VV^\top\widehat{w}^{(t)}, H_t\widehat{w}^{(t)}\rangle\right)\left(1 - 2\eta\langle\widehat{w}^{(t)}, H_t\widehat{w}^{(t)}\rangle\right) - 4\eta^2.$$

We also have that $\langle VV^\top\widehat{w}^{(t)}, H_t\widehat{w}^{(t)}\rangle\langle\widehat{w}^{(t)}, H_t\widehat{w}^{(t)}\rangle \leq 2$, by definition. Plugging this in, we obtain

$$\phi' \geq \phi + 2\eta\langle VV^\top\widehat{w}^{(t)}, H_t\widehat{w}^{(t)}\rangle - 2\eta\phi\langle\widehat{w}^{(t)}, H_t\widehat{w}^{(t)}\rangle - 12\eta^2. \tag{26}$$

Thus the key term that we need to bound becomes:

$$X := \langle \boldsymbol{V}\boldsymbol{V}^\top \widehat{\boldsymbol{w}}^{(t)}, \boldsymbol{H}_t \widehat{\boldsymbol{w}}^{(t)} \rangle - \phi \langle \widehat{\boldsymbol{w}}^{(t)}, \boldsymbol{H}_t \widehat{\boldsymbol{w}}^{(t)} \rangle = (\widehat{\boldsymbol{w}}^{(t)})^\top (\boldsymbol{V}\boldsymbol{V}^T - \phi \boldsymbol{I}) \boldsymbol{H}_t \widehat{\boldsymbol{w}}^{(t)}.$$

We can compute the expectation of this term (conditional on $\boldsymbol{W}^{(t)}$) as follows:

$$\begin{aligned}
\mathbb{E}[X] &= (\widehat{\boldsymbol{w}}^{(t)})^\top (\boldsymbol{V}\boldsymbol{V}^T - \phi \boldsymbol{I})(\boldsymbol{L} + \boldsymbol{M}_t) \widehat{\boldsymbol{w}}^{(t)} \\
&= (\widehat{\boldsymbol{w}}^{(t)})^\top (\boldsymbol{V}\boldsymbol{V}^T - \phi \boldsymbol{I}) \boldsymbol{L} \widehat{\boldsymbol{w}}^{(t)} + (\widehat{\boldsymbol{w}}^{(t)})^\top (\boldsymbol{V}\boldsymbol{V}^T - \phi \boldsymbol{I}) \boldsymbol{M}_t \widehat{\boldsymbol{w}}^{(t)},
\end{aligned}$$

where $\boldsymbol{L} = \mathbb{E}[\boldsymbol{L}_{(i,j)}]$ and $\boldsymbol{M}_t = \mathbb{E}[\boldsymbol{E}_{(i,j)} \mid \boldsymbol{W}^{(t)}]$. We consider the two terms separately. The first term is the most relevant one for the bound, and we can evaluate it in terms of the spectrum of $\boldsymbol{L}$. Suppose $\sigma_i, \boldsymbol{v}_i$ are the singular values and vectors of $\boldsymbol{L}$ respectively, where the singular values are ordered in decreasing order. Then

$$(\boldsymbol{V}\boldsymbol{V}^T - \phi \boldsymbol{I})\boldsymbol{L} = (1 - \phi) \sum_{i \le k} \sigma_i \boldsymbol{v}_i \boldsymbol{v}_i^\top - \phi \sum_{i > k} \sigma_i \boldsymbol{v}_i \boldsymbol{v}_i^\top.$$

Thus, if we were to write $\widehat{\boldsymbol{w}}^{(t)} = \sum_i \alpha_i \boldsymbol{v}_i$, we have

$$\begin{aligned}
(\widehat{\boldsymbol{w}}^{(t)})^\top (\boldsymbol{V}\boldsymbol{V}^T - \phi \boldsymbol{I}) \boldsymbol{L} \widehat{\boldsymbol{w}}^{(t)} &= (1 - \phi) \sum_{i \le k} \sigma_i \alpha_i^2 - \phi \sum_{i > k} \sigma_i \alpha_i^2 \\
&\ge (1 - \phi) \sigma_k \sum_{i \le k} \alpha_i^2 - \phi \sigma_{k+1} \sum_{i > k} \alpha_i^2.
\end{aligned}$$

By definition, we have $\phi = \sum_{i \le k} \alpha_i^2$ and thus the term above is precisely $\phi(1 - \phi)(\sigma_k - \sigma_{k+1})$. The final error term can be bounded as:

$$|(\widehat{\boldsymbol{w}}^{(t)})^\top (\boldsymbol{V}\boldsymbol{V}^T - \phi \boldsymbol{I}) \boldsymbol{M}_t \widehat{\boldsymbol{w}}^{(t)}| \le \|(\boldsymbol{V}\boldsymbol{V}^\top - \phi \boldsymbol{I}) \widehat{\boldsymbol{w}}^{(t)}\| \cdot \|\boldsymbol{M}_t \widehat{\boldsymbol{w}}^{(t)}\|.$$

The first term on the RHS, by triangle inequality, is $\le \sqrt{\phi} + \phi < 2\sqrt{\phi}$, and the second term is $\le \frac{(1+s_n)\sqrt{r}}{n} \|\boldsymbol{W}^{(t)}\|_{2,\infty}^2$, from Lemma C.1. Plugging this in, we get

$$\mathbb{E}[X] \ge \phi(1 - \phi)(\sigma_k - \sigma_{k+1}) - \frac{2(1 + s_n)\sqrt{r\phi}}{n} \cdot \|\boldsymbol{W}^{(t)}\|_{2,\infty}^2.$$

Plugging this back into Equation (26), the lemma follows.

### D.2.2. PROOF OF LEMMA C.3

Our goal is to obtain upper bounds for both $\mathbb{E}\left[\|\boldsymbol{W}^{(t+1)}\|_{2,\infty}^2 \mid \boldsymbol{W}^{(t)}\right]$ and $\|\boldsymbol{W}^{(t+1)}\|_{2,\infty}^2$. Let $\boldsymbol{w}_m^{(t+1)}$ be the row of $\boldsymbol{W}^{(t+1)}$ with the largest norm. Since $\boldsymbol{W}$ is defined as the matrix with the node embedding matrix $\boldsymbol{X}$ and the context embedding matrix $\boldsymbol{Y}$ stacked on top of each other, the row with the largest norm $\boldsymbol{w}_m^{(t+1)}$ can either belong to the set $\{\boldsymbol{x}_i^{(t+1)}\}_{i=1}^n$ or to the set $\{\boldsymbol{y}_j^{(t+1)}\}_{j=1}^n$. We must consider both cases because the node embeddings $\boldsymbol{x}_i$ have different gradients than the context embeddings $\boldsymbol{y}_i$.

First, suppose that $\boldsymbol{w}_m^{(t+1)} \in \{\boldsymbol{x}_i\}_{i=1}^n$ and let $\boldsymbol{x}_m^{(t+1)} = \boldsymbol{w}_m^{(t+1)}$ for simplicity. Then

$$\boldsymbol{x}_m^{(t+1)} = \boldsymbol{x}_m^{(t)} - \eta \nabla_{\boldsymbol{x}_m} \ell_{(i,j)}(\boldsymbol{W}^{(t)}).$$

Also, we have

$$\|\boldsymbol{x}_m^{(t+1)}\|^2 = \|\boldsymbol{x}_m^{(t)}\|^2 + \eta^2 \|\nabla_{\boldsymbol{x}_m} \ell_{(i,j)}(\boldsymbol{W}^{(t)})\|^2 - 2\eta \langle \boldsymbol{x}_m^{(t)}, \nabla_{\boldsymbol{x}_m} \ell_{(i,j)}(\boldsymbol{W}^{(t)}) \rangle,$$

where

$$\nabla_{\boldsymbol{x}_m} \ell_{(i,j)}(\boldsymbol{W}^{(t)}) = \delta_{mi} \left( -\sigma(-\langle \boldsymbol{x}_m^{(t)}, \boldsymbol{y}_j^{(t)} \rangle) \boldsymbol{y}_j^{(t)} + s_n \sum_{k=1}^n \frac{|\boldsymbol{C}_{:k}|}{|\boldsymbol{C}|} \sigma(\langle \boldsymbol{x}_m^{(t)}, \boldsymbol{y}_k^{(t)} \rangle) \boldsymbol{y}_k^{(t)} \right).$$

We bound each term separately. Notice that the gradient is zero unless the source node $i$ of the random sample $(i,j)$ is equal to the row $m$. First, we bound $-\langle \boldsymbol{x}_m^{(t)}, \nabla_{\boldsymbol{x}_m}\ell_{(i,j)}(\boldsymbol{W}^{(t)})\rangle$:

$$
\begin{aligned}
-\langle \boldsymbol{x}_m^{(t)}, \nabla_{\boldsymbol{x}_m}\ell_{(i,j)}\rangle &= -\left\langle \boldsymbol{x}_m^{(t)}, \delta_{mi}\left(-\sigma(-\langle\boldsymbol{x}_m^{(t)},\boldsymbol{y}_j^{(t)}\rangle)\boldsymbol{y}_j^{(t)} + s_n\sum_{k=1}^{n}\frac{|\boldsymbol{C}_{:k}|}{|\boldsymbol{C}|}\sigma(\langle\boldsymbol{x}_m^{(t)},\boldsymbol{y}_k^{(t)}\rangle)\boldsymbol{y}_k^{(t)}\right)\right\rangle \\
&= \delta_{mi}\sigma(-\langle\boldsymbol{x}_m^{(t)},\boldsymbol{y}_j^{(t)}\rangle)\langle\boldsymbol{x}_m^{(t)},\boldsymbol{y}_j^{(t)}\rangle - \delta_{mi}s_n\sum_{k=1}^{n}\frac{|\boldsymbol{C}_{:k}|}{|\boldsymbol{C}|}\sigma(\langle\boldsymbol{x}_m^{(t)},\boldsymbol{y}_k^{(t)}\rangle)\langle\boldsymbol{x}_m^{(t)},\boldsymbol{y}_k^{(t)}\rangle \\
&\leq \delta_{mi}\|\boldsymbol{x}_m^{(t)}\|\cdot\|\boldsymbol{y}_j^{(t)}\| + \delta_{mi}\cdot s_n\sum_{k=1}^{n}\frac{|\boldsymbol{C}_{:k}|}{|\boldsymbol{C}|}\|\boldsymbol{x}_m^{(t)}\|\cdot\|\boldsymbol{w}_k^{(t)}\| \\
&\leq \delta_{mi}\|\boldsymbol{W}^{(t)}\|_{2,\infty}^2(1+s_n).
\end{aligned}
$$

Next, we bound $\|\nabla_{\boldsymbol{x}_m}\ell_{(i,j)}(\boldsymbol{W}^{(t)})\|^2$:

$$
\begin{aligned}
\|\nabla_{\boldsymbol{x}_m}\ell_{(i,j)}(\boldsymbol{W}^{(t)})\|^2 &= \langle\nabla_{\boldsymbol{x}_m}\ell_{(i,j)}(\boldsymbol{W}^{(t)}), \nabla_{\boldsymbol{x}_m}\ell_{(i,j)}(\boldsymbol{W}^{(t)})\rangle \\
&= \left\langle \delta_{mi}\left(-\sigma(-\langle\boldsymbol{x}_m^{(t)},\boldsymbol{y}_j^{(t)}\rangle)\boldsymbol{y}_j^{(t)} + s_n\sum_{k=1}^{n}\frac{|\boldsymbol{C}_{:k}|}{|\boldsymbol{C}|}\sigma(\langle\boldsymbol{x}_m^{(t)},\boldsymbol{y}_k^{(t)}\rangle)\boldsymbol{y}_k^{(t)}\right), \right.\\
&\qquad\left. \delta_{mi}\left(-\sigma(-\langle\boldsymbol{x}_m^{(t)},\boldsymbol{y}_j^{(t)}\rangle)\boldsymbol{y}_j^{(t)} + s_n\sum_{k=1}^{n}\frac{|\boldsymbol{C}_{:k}|}{|\boldsymbol{C}|}\sigma(\langle\boldsymbol{x}_m^{(t)},\boldsymbol{y}_k^{(t)}\rangle)\boldsymbol{y}_k^{(t)}\right)\right\rangle \\
&\leq \delta_{mi}\|\boldsymbol{y}_j^{(t)}\|^2 + \delta_{mi}\cdot 2s_n\sum_{k=1}^{n}\frac{|\boldsymbol{C}_{:k}|}{|\boldsymbol{C}|}\|\boldsymbol{y}_j^{(t)}\|\cdot\|\boldsymbol{y}_k^{(t)}\| \\
&\quad + \delta_{mi}\cdot s_n^2\sum_{k=1}^{n}\sum_{k'=1}^{n}\frac{|\boldsymbol{C}_{:k}||\boldsymbol{C}_{:k'}|}{|\boldsymbol{C}|^2}\|\boldsymbol{y}_k^{(t)}\|\cdot\|\boldsymbol{y}_{k'}^{(t)}\| \\
&= \delta_{mi}\|\boldsymbol{W}^{(t)}\|_{2,\infty}^2(1+s_n)^2.
\end{aligned}
$$

Therefore, we have

$$
\begin{aligned}
\|\boldsymbol{x}_m^{(t+1)}\|^2 &= \|\boldsymbol{x}_m^{(t)}\|^2 + \eta^2\|\nabla_{\boldsymbol{x}_m}\ell_{(i,j)}(\boldsymbol{W}^{(t)})\|^2 - 2\eta\langle\boldsymbol{x}_m^{(t)}, \nabla_{\boldsymbol{x}_m}\ell_{(i,j)}(\boldsymbol{W}^{(t)})\rangle \\
&\leq \|\boldsymbol{W}^{(t)}\|_{2,\infty}^2 + \eta^2\delta_{mi}\|\boldsymbol{W}^{(t)}\|_{2,\infty}^2(1+s_n)^2 + 2\eta\delta_{mi}\|\boldsymbol{W}^{(t)}\|_{2,\infty}^2(1+s_n) \\
&= \|\boldsymbol{W}^{(t)}\|_{2,\infty}^2(1+\eta^2\delta_{mi}(1+s_n)^2 + 2\eta\delta_{mi}(1+s_n)) \\
&\leq \|\boldsymbol{W}^{(t)}\|_{2,\infty}^2(1+\delta_{mi}\cdot 3\eta(1+s_n)^2).
\end{aligned}
$$

Taking the expectation conditional on $\boldsymbol{W}^{(t)}$, we have

$$
\begin{aligned}
\mathbb{E}[\|\boldsymbol{W}^{(t+1)}\|_{2,\infty}^2 \mid \boldsymbol{W}^{(t)}] &\leq \sum_{i=1}^{n}\sum_{j=1}^{n}\frac{C_{ij}}{|\boldsymbol{C}|}\cdot\|\boldsymbol{W}^{(t)}\|_{2,\infty}^2(1+\delta_{mi}\cdot 3\eta(1+s_n)^2) \\
&= \|\boldsymbol{W}^{(t)}\|_{2,\infty}^2\left(1+3\eta(1+s_n)^2\frac{|\boldsymbol{C}_{m:}|}{|\boldsymbol{C}|}\right).
\end{aligned}
$$

Similarly, assuming the worst-case when $\delta_{mi}=1$, when $\boldsymbol{w}_m^{(t+1)}\in\{\boldsymbol{x}_i\}_{i=1}^{n}$ we have

$$
\|\boldsymbol{W}^{(t+1)}\|_{2,\infty}^2 \leq \left(1+3\eta(1+s_n)^2\right)\|\boldsymbol{W}^{(t)}\|_{2,\infty}^2.
$$

Now, we consider the second case when $\boldsymbol{w}_m^{(t+1)}\in\{\boldsymbol{y}_j\}_{j=1}^{n}$. Again, let $\boldsymbol{y}_m^{(t+1)}=\boldsymbol{w}_m^{(t+1)}$ for simplicity. Then

$$
\boldsymbol{w}_m^{(t+1)} = \boldsymbol{y}_m^{(t)} - \eta\nabla_{\boldsymbol{y}_m}\ell_{(i,j)}(\boldsymbol{W}^{(t)}).
$$

Also, we have

$$\|\boldsymbol{y}_m^{(t+1)}\|^2 = \|\boldsymbol{y}_m^{(t)}\|^2 + \eta^2\|\nabla_{\boldsymbol{y}_m}\ell_{(i,j)}(\boldsymbol{W}^{(t)})\|^2 - 2\eta\langle\boldsymbol{y}_m^{(t)}, \nabla_{\boldsymbol{y}_m}\ell_{(i,j)}(\boldsymbol{W}^{(t)})\rangle,$$

where

$$\nabla_{\boldsymbol{y}_m}\ell_{(i,j)}(\boldsymbol{W}^{(t)}) = \delta_{mj}(-\sigma(-\langle\boldsymbol{x}_i^{(t)}, \boldsymbol{y}_m^{(t)}\rangle)\boldsymbol{x}_i^{(t)}) + s_n\frac{|\boldsymbol{C}_{:m}|}{|\boldsymbol{C}|}\sigma(\langle\boldsymbol{x}_i^{(t)}, \boldsymbol{y}_m^{(t)}\rangle)\boldsymbol{x}_i^{(t)}.$$

Notice that unlike the updates for $\boldsymbol{x}_i$, the values of $\boldsymbol{y}_j$ are always updated, albeit with an additional term if the context node $j$ in the random sample $(i,j)$ is equal to $m$. We first bound $-\langle\boldsymbol{y}_m^{(t)}, \nabla_{\boldsymbol{y}_m}\ell_{(i,j)}(\boldsymbol{W}^{(t)})\rangle$:

$$-\langle\boldsymbol{y}_m^{(t)}, \nabla_{\boldsymbol{y}_m}\ell_{(i,j)}(\boldsymbol{W}^{(t)})\rangle = -\left\langle\boldsymbol{y}_m^{(t)}, \delta_{mj}(-\sigma(-\langle\boldsymbol{x}_i^{(t)}, \boldsymbol{y}_m^{(t)}\rangle)\boldsymbol{x}_i^{(t)})\right\rangle$$

$$-\left\langle\boldsymbol{y}_m^{(t)}, s_n\frac{|\boldsymbol{C}_{:m}|}{|\boldsymbol{C}|}\sigma(\langle\boldsymbol{x}_i^{(t)}, \boldsymbol{y}_m^{(t)}\rangle)\boldsymbol{x}_i^{(t)}\right\rangle$$

$$\leq \delta_{mj}\|\boldsymbol{W}^{(t)}\|_{2,\infty}^2 + s_n\frac{|\boldsymbol{C}_{:m}|}{|\boldsymbol{C}|}\|\boldsymbol{W}^{(t)}\|_{2,\infty}^2.$$

Next, we bound $\|\nabla_{\boldsymbol{y}_m}\ell_{(i,j)}(\boldsymbol{W}^{(t)})\|^2$

$$\|\nabla_{\boldsymbol{y}_m}\ell_{(i,j)}(\boldsymbol{W}^{(t)})\|^2 = \left\langle\delta_{mj}(-\sigma(-\langle\boldsymbol{x}_i^{(t)}, \boldsymbol{y}_m^{(t)}\rangle)\boldsymbol{x}_i^{(t)}) + s_n\frac{|\boldsymbol{C}_{:m}|}{|\boldsymbol{C}|}\sigma(\langle\boldsymbol{x}_i^{(t)}, \boldsymbol{y}_m^{(t)}\rangle)\boldsymbol{x}_i^{(t)},\right.$$

$$\left.\delta_{mj}(-\sigma(-\langle\boldsymbol{x}_i^{(t)}, \boldsymbol{y}_m^{(t)}\rangle)\boldsymbol{x}_i^{(t)}) + s_n\frac{|\boldsymbol{C}_{:m}|}{|\boldsymbol{C}|}\sigma(\langle\boldsymbol{x}_i^{(t)}, \boldsymbol{y}_m^{(t)}\rangle)\boldsymbol{x}_i^{(t)}\right\rangle$$

$$\leq \delta_{mj}\|\boldsymbol{W}^{(t)}\|_{2,\infty}^2 + \delta_{mj}\cdot 2s_n\frac{|\boldsymbol{C}_{:m}|}{|\boldsymbol{C}|}\|\boldsymbol{W}^{(t)}\|_{2,\infty}^2 + s_n^2\frac{|\boldsymbol{C}_{:m}|^2}{|\boldsymbol{C}|^2}\|\boldsymbol{W}^{(t)}\|_{2,\infty}^2.$$

Therefore, we have

$$\|\boldsymbol{y}_m^{(t+1)}\|^2 = \|\boldsymbol{y}_m^{(t)}\|^2 + \eta^2\|\nabla_{\boldsymbol{y}_m}\ell_{(i,j)}(\boldsymbol{W}^{(t)})\|^2 - 2\eta\langle\boldsymbol{y}_m^{(t)}, \nabla_{\boldsymbol{y}_m}\ell_{(i,j)}(\boldsymbol{W}^{(t)})\rangle$$

$$\leq \|\boldsymbol{W}^{(t)}\|_{2,\infty}^2 + \eta^2\|\boldsymbol{W}^{(t)}\|_{2,\infty}^2\left(\delta_{mj} + \delta_{mj}\cdot 2s_n\frac{|\boldsymbol{C}_{:m}|}{|\boldsymbol{C}|} + s_n^2\frac{|\boldsymbol{C}_{:m}|^2}{|\boldsymbol{C}|^2}\right)$$

$$+ 2\eta\|\boldsymbol{W}^{(t)}\|_{2,\infty}^2\left(\delta_{mj} + s_n\frac{|\boldsymbol{C}_{:m}|}{|\boldsymbol{C}|}\right)$$

$$= \|\boldsymbol{W}^{(t)}\|_{2,\infty}^2\left(1 + 2\eta s_n\frac{|\boldsymbol{C}_{:m}|}{|\boldsymbol{C}|} + \eta^2 s_n^2\frac{|\boldsymbol{C}_{:m}|^2}{|\boldsymbol{C}|^2}\right)$$

$$+ \delta_{mj}\|\boldsymbol{W}^{(t)}\|_{2,\infty}^2\left(\eta^2 + 2\eta^2 s_n\frac{|\boldsymbol{C}_{:m}|}{|\boldsymbol{C}|} + 2\eta\right).$$

Taking the expectation conditional on $\boldsymbol{W}^{(t)}$, we have

$$\mathbb{E}[\|\boldsymbol{W}^{(t+1)}\|_{2,\infty}^2 \mid \boldsymbol{W}^{(t)}] \leq \|\boldsymbol{W}^{(t)}\|_{2,\infty}^2\left(1 + 2\eta s_n\frac{|\boldsymbol{C}_{:m}|}{|\boldsymbol{C}|} + \eta^2 s_n^2\frac{|\boldsymbol{C}_{:m}|^2}{|\boldsymbol{C}|^2}\right)$$

$$+ \sum_{i=1}^n\sum_{j=1}^n\frac{C_{ij}}{|\boldsymbol{C}|}\delta_{mj}\|\boldsymbol{W}^{(t)}\|_{2,\infty}^2\left(\eta^2 + 2\eta^2 s_n\frac{|\boldsymbol{C}_{:m}|}{|\boldsymbol{C}|} + 2\eta\right)$$

$$= \|\boldsymbol{W}^{(t)}\|_{2,\infty}^2\left(1 + 3\eta(1 + s_n)^2\frac{|\boldsymbol{C}_{:m}|}{|\boldsymbol{C}|}\right).$$

Again assuming the worst case $\delta_{mj} = 1$, when $\boldsymbol{w}_m^{(t+1)} \in \{\boldsymbol{y}_j\}_{j=1}^n$ we have

$$\|\boldsymbol{W}^{(t+1)}\|_{2,\infty}^2 \leq \|\boldsymbol{W}^{(t)}\|_{2,\infty}^2\left(1 + 3\eta(1 + s_n)^2\right).$$

From Lemma C.1, we have that $\|C\|_\infty / |C| \le 2/n$. It follows that

$$\mathbb{E}[\|W^{(t+1)}\|_{2,\infty}^2 \mid W^{(t)}] \le \|W^{(t)}\|_{2,\infty}^2 \left(1 + \frac{6\eta(1+s_n)^2}{n}\right).$$

We have also shown that

$$\|W^{(t+1)}\|_{2,\infty}^2 \le \|W^{(t)}\|_{2,\infty}^2 (1 + 3\eta(1+s_n)^2).$$

This concludes the proof.

### D.3. Proofs From Section C.3

D.3.1. PROOF OF LEMMA C.4

For any $t$, from Lemma C.3 we have that

$$\mathbb{E}[\|W^{(t)}\|_{2,\infty}^2 \mid W^{(t)}] \le \left(1 + \frac{6\eta(1+s_n)^2}{n}\right) \|W^{(t)}\|_{2,\infty}^2.$$

Define $Y_t := \ln(\|W^{(t)}\|_{2,\infty}^2)$. Then

$$\begin{aligned}
\mathbb{E}[Y_t \mid W^{(t-1)}] &\le \ln\left(\mathbb{E}[\|W^{(t)}\|_{2,\infty}^2 \mid W^{(t-1)}]\right) \\
&\le \ln\left(\left(1 + \frac{6\eta(1+s_n)^2}{n}\right)\|W^{(t-1)}\|_{2,\infty}^2\right) \\
&\le \ln\left(\|W^{(t-1)}\|_{2,\infty}^2\right) + \frac{6\eta(1+s_n)^2}{n}.
\end{aligned}$$

In the first inequality, we used the concavity of the logarithm, and in the last inequality, we used $ln(1+x) < x$ for all $x > 0$.

With this information, we define $M_t := Y_t - t\left(\frac{6\eta(1+s_n)^2}{n}\right) - Y_0$. It is easy to see that this is a supermartingale with respect to $W^{(t)}$:

$$\begin{aligned}
\mathbb{E}[M_t \mid W^{(t-1)}] &= \mathbb{E}\left[Y_t - t\left(\frac{6\eta(1+s_n)^2}{n}\right) - Y_0 \mid W^{(t-1)}\right] \\
&\le Y_{t-1} + \frac{6\eta(1+s_n)^2}{n} - t\left(\frac{6\eta(1+s_n)^2}{n}\right) - Y_0 \\
&= Y_{t-1} + (t-1)\left(\frac{6\eta(1+s_n)^2}{n}\right) - Y_0 \\
&= M_{t-1}.
\end{aligned}$$

Now, we bound the difference $|M_t - M_{t-1}|$:

$$\begin{aligned}
|M_t - M_{t-1}| &= |Y_t - Y_{t-1} - \frac{6\eta(1+s_n)^2}{n}| \\
&= \left|\ln\left(\frac{\|W^{(t)}\|_{2,\infty}^2}{\|W^{(t-1)}\|_{2,\infty}^2}\right) - \frac{6\eta(1+s_n)^2}{n}\right|.
\end{aligned}$$

From Lemma C.3, we have that

$$\|W^{(t)}\|_{2,\infty}^2 \le \left(1 + 3\eta(1+s_n)^2\right)\|W^{(t-1)}\|_{2,\infty}^2.$$

It follows that

$$\begin{aligned}
|M_t - M_{t-1}| &\le \left|\ln\left(1 + 3\eta(1+s_n)^2\right)\right| + \frac{6\eta(1+s_n)^2}{n} \\
&\le 3\eta(1+s_n)^2 + \frac{6\eta(1+s_n)^2}{n} \\
&\le 4\eta(1+s_n)^2.
\end{aligned}$$

Property (a) now follows by Azuma's inequality. For any $t$, we have

$$\Pr\left[Y_t \geq Y_0 + t \cdot \frac{6\eta(1+s_n)^2}{n} + \sqrt{32t\eta^2(1+s_n)^4 \ln\left(\frac{1}{\delta}\right)}\right] \leq \delta.$$

Property (b) is also easy to see by applying Azuma's maximal inequality. We have

$$\Pr\left[\max_{t\in[T]} Y_t > Y_0 + T \cdot \frac{6\eta(1+s_n)^2}{n} + \sqrt{32T\eta^2(1+s_n)^4 \ln\left(\frac{1}{\delta}\right)}\right] \leq \delta.$$

By assumption, $T$ and $\eta$ have been chosen so that

$$T \cdot \frac{6\eta(1+s_n)^2}{n} + \sqrt{32T\eta^2(1+s_n)^4 \ln\left(\frac{1}{\delta}\right)} \leq \ln(c),$$

for some constant $c$. Therefore, with probability at least $1 - \delta$

$$Y_t \leq Y_0 + \ln(c)$$

for all $t \in [T]$. It immediately follows that

$$\|\boldsymbol{W}^{(t)}\|_{2,\infty}^2 \leq c\|\boldsymbol{W}^{(0)}\|_{2,\infty}^2$$

for all $t \in [T]$ with probability at least $1 - \delta$. This concludes the proof.

### D.3.2. PROOF OF LEMMA C.5

By condition (a) assumed in the lemma statement, we can use Markov's conditional inequality and Lemma C.6 to obtain for any $t \in [T]$ and parameter $s > 0$,

$$\Pr[\psi_t - \psi_0 \geq s \mid \mathcal{E}_T] \leq 2\exp\left(t\left(\frac{4c\lambda\eta(1+s_n)r^{\frac{1}{2}}\tau^2}{n} + 12\lambda\eta^2 + 36\lambda^2\eta^2\right) - \lambda s\right).$$

Setting $s = 4ct\eta(1+s_n)r^{\frac{1}{2}}\tau^2/n + 12t\eta^2 + s'$, we have

$$\Pr\left[\psi_t - \psi_0 \geq \frac{4ct\eta(1+s_n)r^{\frac{1}{2}}\tau^2}{n} + 12t\eta^2 + s' \mid \mathcal{E}_T\right] \leq 2\exp\left(36t\lambda^2\eta^2 - \lambda s'\right).$$

If $\lambda = s'/72t\eta^2$, this simplifies to

$$\Pr\left[\psi_t - \psi_0 \geq \frac{4ct\eta(1+s_n)r^{\frac{1}{2}}\tau^2}{n} + 12t\eta^2 + s' \mid \mathcal{E}_T\right] \leq 2\exp\left(-\frac{(s')^2}{4(36)t\eta^2}\right).$$

Therefore, with probability at least $1 - \delta_1$, we have for any $t \in [T]$ that

$$\Pr\left[\psi_t \geq \psi_0 + \frac{4ct\eta(1+s_n)r^{\frac{1}{2}}\tau^2}{n} + 12t\eta^2 + 12\eta\sqrt{t\ln\left(\frac{2}{\delta_1}\right)} \mid \mathcal{E}_T\right] \leq \delta_1.$$

The law of total probability says that given two events $A$ and $\mathcal{E}_T$

$$\Pr[A] = \Pr[A \mid \mathcal{E}_T]\Pr[\mathcal{E}_T] + \Pr[A \mid \mathcal{E}_T^c]\Pr[\mathcal{E}_T^c] \leq \Pr[A \mid \mathcal{E}_T] + \Pr[\mathcal{E}_T^c].$$

From Lemma C.4 we have $\Pr[\mathcal{E}_T^c] = \delta_2$. Therefore,

$$\Pr\left[\psi_t \geq \psi_0 + \frac{4ct\eta(1+s_n)r^{\frac{1}{2}}\tau^2}{n} + 12t\eta^2 + 12\eta\sqrt{t\ln\left(\frac{2}{\delta_1}\right)}\right] \leq \delta_1 + \delta_2.$$

Taking a union bound over all $t \in [T]$ and setting $\delta_1 = \delta_2 = \delta/2$, we have that

$$\Pr\left[\psi_t \leq \psi_0 + \frac{4ct\eta(1+s_n)r^{\frac{1}{2}}\tau^2}{n} + 12t\eta^2 + 12\eta\sqrt{t\ln\left(\frac{4}{\delta}\right)}\right] \leq 1 - T\delta$$

for all $t \in [T]$. By assumption, $\psi_0 < 1 - a$ and $\eta$ and $T$ are chosen so that

$$T \cdot \frac{4c\eta(1+s_n)r^{\frac{1}{2}}\tau^2}{n} + T \cdot 12\eta^2 + 12\eta\sqrt{T\ln\left(\frac{4}{\delta}\right)} \leq \frac{a}{2}.$$

Therefore,

$$\psi_t \leq 1 - \frac{a}{2}.$$

for all $t \in [T]$ with probability $1 - T\delta$. This concludes the proof.

### D.3.3. PROOF OF LEMMA C.6

Define the martingale difference $X_t = \psi_t - \psi_{t-1}$. Then

$$\mathbb{E}[\exp\left(\lambda(\psi_t - \psi_0)\right) \mid \mathcal{E}_T] = \mathbb{E}[\exp\left(\lambda(X_1 + \ldots + X_t)\right) \mid \mathcal{E}_T].$$

Let $A$ be the event that $\exp\left(\lambda(X_1 + \ldots + X_t)\right) = z$ for some potential value $z \in \mathbb{R}$. Then

$$\Pr[A \mid \mathcal{E}_T] = \frac{\Pr[A \cap \mathcal{E}_T]}{\Pr[\mathcal{E}_T]}.$$

By construction $\mathcal{E}_T \subseteq \mathcal{E}_{T-1} \subseteq \ldots \subseteq \mathcal{E}_1$, which implies that $A \cap \mathcal{E}_T \subseteq A \cap \mathcal{E}_{t-1}$ and therefore $\Pr[A \cap \mathcal{E}_T] \leq \Pr[A \cap \mathcal{E}_{t-1}]$. Then

$$\Pr[A \mid \mathcal{E}_T] \leq \frac{\Pr[A \cap \mathcal{E}_T]}{\Pr[\mathcal{E}_T]} \leq \frac{\Pr[A \cap \mathcal{E}_{t-1}]}{\Pr[\mathcal{E}_T]} = \frac{\Pr[A \cap \mathcal{E}_{t-1}]}{\Pr[\mathcal{E}_{t-1}]}\frac{\Pr[\mathcal{E}_{t-1}]}{\mathcal{E}_T} \leq \frac{\Pr[A \cap \mathcal{E}_{t-1}]}{\Pr[\mathcal{E}_{t-1}]}\frac{1}{\Pr[\mathcal{E}_T]}.$$

This implies that

$$\mathbb{E}[\exp\left(\lambda(X_1 + \ldots + X_t)\right) \mid \mathcal{E}_T] \leq \frac{1}{\Pr[\mathcal{E}_T]} \cdot \mathbb{E}[\exp\left(\lambda(X_1 + \ldots + X_t)\right) \mid \mathcal{E}_{t-1}].$$

Recall that $\mathcal{E}_T$ is the event that $\max_{j \leq T} \|\boldsymbol{W}^{(j)}\|_{2,\infty}^2 \leq c\tau^2$. From Lemma C.4, we know that for any small $\delta > 0$, if $\eta$ and $T$ are chosen appropriately as we have assumed they have in the lemma statement, $\|\boldsymbol{W}^{(t)}\|_{2,\infty}^2 \leq c\tau^2$ for all $t \in [T]$ with probability $1 - \delta$. Also, $X_t$ is conditionally independent of $X_1, X_2, \ldots X_{t-1}$. Therefore,

$$\mathbb{E}[\exp\left(\lambda(X_1 + \ldots + X_t)\right) \mid \mathcal{E}_T] \leq \frac{\mathbb{E}[\exp\left(\lambda X_t\right) \mid \mathcal{E}_{t-1}]}{1 - \delta} \cdot \mathbb{E}[\exp\left(\lambda(X_1 + \ldots + X_{t-1})\right) \mid \mathcal{E}_{t-1}].$$

We bound $\mathbb{E}[\exp\left(\lambda X_t\right) \mid \mathcal{E}_{t-1}]$ using Hoeffding's lemma:

$$\mathbb{E}[\exp\left(\lambda X_t\right) \mid \mathcal{E}_{t-1}] \leq \exp\left(\frac{4c\lambda\eta(1+s_n)r^{\frac{1}{2}}\tau^2}{n} + 12\lambda\eta^2 + 36\lambda^2\eta^2\right).$$

Let $f(\lambda) = \frac{4c\lambda\eta(1+s_n)r^{\frac{1}{2}}\tau^2}{n} + 12\lambda\eta^2 + 36\lambda^2\eta^2$. Thus, we have

$$\mathbb{E}\left[\exp\left(\lambda\left(\sum_{i=1}^{t}X_i\right)\right) \mid \mathcal{E}_T\right] \leq \frac{\exp\left(f(\lambda)\right)}{1 - \delta} \cdot \mathbb{E}\left[\exp\left(\lambda\left(\sum_{i=1}^{t-1}X_i\right)\right) \mid \mathcal{E}_{t-1}\right].$$

Repeating this line of reasoning another $t - 1$ times gives

$$\mathbb{E}\left[\exp\left(\lambda\left(\sum_{i=1}^{t}X_i\right)\right) \mid \mathcal{E}_T\right] \leq \frac{1}{(1 - \delta)^t}\exp\left(tf(\lambda)\right).$$

Since $\delta$ is assumed to be small, then $1/(1-\delta)^t < 2$. Then

$$\mathbb{E}\left[\exp\left(\lambda\left(\sum_{i=1}^{t} X_i\right)\right) \mid \mathcal{E}_T\right] \leq 2\exp\left(tf(\lambda)\right).$$

This concludes the proof.

### D.3.4. PROOF OF LEMMA C.7

Given the choice of parameters

$$T = \frac{1}{6(1+s_n)^2} \cdot \frac{n}{\eta} \text{ and } \eta \leq \frac{a^2 b^2}{36 n^2 \ln\left(4/\delta\right)},$$

Lemma C.4 implies that $\|\mathbf{W}^{(t)}\|_{2,\infty}^2 \leq 3\tau^2$ for all $t \in [T]$ with probability $1-\delta$. With this observation, we define a sequence of events $\mathcal{F}_T \subseteq \mathcal{F}_{t-1} \subseteq \ldots \subseteq \mathcal{F}_1$ where $\mathcal{F}_t$ is the event that $\max_{j\leq t} \psi_j \leq 1 - a/2$ and $\max_{j\leq t}\|\mathbf{W}^{(j)}\|_{2,\infty}^2 \leq 3\tau^2$. Following the same line of reasoning as in the proof of Lemma C.6 to bound the moment generating function, we have for any $\lambda > 0$ that

$$\mathbb{E}[\exp\left(\lambda(\psi_T - \psi_0) \mid \mathcal{F}_T\right]$$
$$\leq 2 \cdot \exp\left(T\left(-\frac{\lambda\eta\Delta_{2k}a}{2} + \frac{4\cdot3\cdot\lambda\eta(1+s_n)r^{\frac{1}{2}}\tau^2}{n} + 12\lambda\eta^2 + 36\lambda^2\eta^2\right)\right).$$

Using the conditional Markov inequality, for $s > 0$ we have

$$\Pr[\psi_T - \psi_0 \geq s \mid \mathcal{F}_T]$$
$$\leq 2 \cdot \exp\left(T\left(-\frac{\lambda\eta\Delta_{2k}a}{2} + \frac{4\cdot3\cdot\lambda\eta(1+s_n)r^{\frac{1}{2}}\tau^2}{n} + 12\lambda\eta^2 + 36\lambda^2\eta^2\right) - \lambda s\right).$$

Setting $s = -\frac{T\eta\Delta_{2k}a}{2} + \frac{4\cdot3\cdot T\eta(1+s_n)r^{\frac{1}{2}}\tau^2}{n} + 12T\eta^2 + s'$ and $\lambda = \frac{s'}{72T\eta^2}$ gives

$$\Pr\left[\psi_T \geq \psi_0 - \frac{T\eta\Delta_{2k}a}{2} + \frac{4\cdot3\cdot T\eta(1+s_n)r^{\frac{1}{2}}\tau^2}{n} + 12T\eta^2 + 12\eta\sqrt{T\ln\left(\frac{4}{\delta}\right)} \mid \mathcal{F}_T\right] \leq \frac{\delta}{2},$$

for a small confidence parameter $\delta \in (0,1)$, further implying the unconditional probability

$$\Pr\left[\psi_T \geq \psi_0 - \frac{T\eta\Delta_{2k}a}{2} + \frac{4\cdot3\cdot T\eta(1+s_n)r^{\frac{1}{2}}\tau^2}{n} + 12T\eta^2 + 12\eta\sqrt{T\ln\left(\frac{4}{\delta}\right)}\right] \leq \frac{\delta}{2} + T\delta,$$

where we used Lemma C.5, which says that $\Pr[\mathcal{F}_T^c] \leq T\delta$. Therefore,

$$\Pr\left[\psi_T \leq \psi_0 - \frac{T\eta\Delta_{2k}a}{2} + \frac{12T\eta(1+s_n)r^{\frac{1}{2}}\tau^2}{n} + 12T\eta^2 + 12\eta\sqrt{T\ln\left(\frac{4}{\delta}\right)}\right] \leq 1 - (T+1)\delta.$$

Given our chosen values of $T$, $\eta$ and $\tau^2$ and the fact that $\Delta_{2k} = \Theta(1/n)$ (see Lemma C.1), we have

$$\frac{12T\eta(1+s_n)r^{\frac{1}{2}}\tau^2}{n} + 12T\eta^2 + 12\eta\sqrt{T\ln\left(\frac{4}{\delta}\right)} \leq \frac{ab}{6}.$$

It then follows that

$$\psi_T < \psi_0 + \frac{ab}{6} - \frac{T\eta\Delta_{2k}a}{2} = \psi_0 - \frac{ab}{3}.$$

with probability at least $1 - (T+1)\delta$. This concludes the proof.

# E. Community Recovery in the Stochastic Block Model

In the preceding sections, we showed that the columns of $W$ converge to a vector that lies in the span of the top $2k$ eigenvectors of $\mathbb{E}[L_{(i,j)}]$. In this section, we show that every column of $W$ has a well-defined cluster structure. To this end, we first show that the embeddings belonging to the same community are close to each other when the algorithm ends, forming clusters. Second, we show that the cluster means are separated. This implies Theorem 5.1.

## E.1. Well-clustered Embeddings

We now prove a Lemma which states that the resulting embeddings are well-clustered around the cluster means. Formally,

*Lemma E.1. Suppose that $\mathcal{G}, \rho, M, T$ be as defined in Theorem 5.1, and let $w, x$ be the resulting embeddings. Then with probability $\geq 0.95$, we have*

$$\|x - \mu\|^2 \leq \frac{c\|w\|^2}{n^\rho},$$

*where $\mu$ is the vector of intra-cluster averages (Equation (17)), and $c$ is an absolute constant.*

*Proof.* First, we define some notation. Let $\Omega_k$ be the set of indices that belong to cluster $k \in [K]$ and let $\mathbf{1}_{\Omega_k}$ the vector where $(\mathbf{1}_{\Omega_k})_i = 1$ if $i \in \Omega_k$ and 0 otherwise. For any vector $v \in \mathbb{R}^n$, we let $v^+ = \begin{bmatrix} v \\ v \end{bmatrix}$ and $v^- = \begin{bmatrix} v \\ -v \end{bmatrix}$.

We define the $2n \times 2n$ matrix $\overline{L}$ as

$$\overline{L} = \begin{bmatrix} 0 & L' \\ L' & 0 \end{bmatrix},$$

where $L' = \frac{1}{2\kappa} \left( \overline{C} - \frac{s_n}{\kappa} \overline{C}^\top \mathbf{1}\mathbf{1}^\top \overline{C} \right)$ and $\overline{C}$ is defined as in Definition D.1. Recall that $\overline{L}$ is an off-diagonal block matrix. Let $u_1, \ldots, u_k$ be the top $K$ eigenvectors of $L'$. It is easy to see that the top $K$ eigenvectors of $L'$ can be expressed as

$$u_1 = \mathbf{1} = \mathbf{1}_{\Omega_1} + \ldots + \mathbf{1}_{\Omega_K},$$
$$u_2 = \mathbf{1}_{\Omega_1} - \mathbf{1}_{\Omega_2},$$
$$u_3 = \mathbf{1}_{\Omega_2} - \mathbf{1}_{\Omega_3},$$
$$\vdots$$
$$u_K = \mathbf{1}_{\Omega_{K-1}} - \mathbf{1}_{\Omega_K}.$$

Since $\overline{L}$ is an off-diagonal block matrix, its top $2K$ eigenvectors of $\overline{L}$ are $u_1^+, u_1^-, \ldots, u_K^+, u_K^-$. Notice that the eigenvectors of $L'$ lie in the space spanned by the $K$ vectors $\mathbf{1}_{\Omega_1}, \ldots, \mathbf{1}_{\Omega_K}$, and that the eigenvectors of $\overline{L}$ lie in the space spanned by the $2K$ vectors $\mathbf{1}_{\Omega_1}^+, \mathbf{1}_{\Omega_1}^-, \ldots, \mathbf{1}_{\Omega_K}^+, \mathbf{1}_{\Omega_K}^-$.

Suppose that we want to find a vector $x'$ in the space spanned by $\mathbf{1}_{\Omega_1}, \ldots, \mathbf{1}_{\Omega_K}$ that minimizes $\|x - x'\|^2$. Writing $x' = \sum_{k \in [K]} \alpha_k \mathbf{1}_{\Omega_k}$, we see that the vector that minimizes $\|x - x'\|^2$ is

$$x' = \frac{K}{n} \sum_{k \in [K]} \mathbf{1}_{\Omega_k} \mathbf{1}_{\Omega_k}^\top x = \mu_x.$$

Let $\Gamma$ be the projection onto the space spanned by the vectors $\mathbf{1}_{\Omega_1}^+, \mathbf{1}_{\Omega_1}^-, \ldots, \mathbf{1}_{\Omega_K}^+, \mathbf{1}_{\Omega_K}^-$. Then

$$\|x - \mu_x\|^2 \leq \|x - \mu_x\|^2 + \|y - \mu_y\|^2$$

$$\leq \left\| x - \frac{K}{n} \sum_{k \in [K]} \mathbf{1}_{\Omega_k} \mathbf{1}_{\Omega_k}^\top x \right\|^2 + \left\| y - \frac{K}{n} \sum_{k \in [K]} \mathbf{1}_{\Omega_k} \mathbf{1}_{\Omega_k}^\top y \right\|^2$$

$$= \left\| w - \frac{K}{2n} \sum_{k \in [K]} \left( \mathbf{1}_{\Omega_k}^+ \left(\mathbf{1}_{\Omega_k}^+\right)^\top + \mathbf{1}_{\Omega_k}^- \left(\mathbf{1}_{\Omega_k}^-\right)^\top \right) w \right\|^2$$

$$= \|(I - \Gamma)w\|^2.$$

However, the space spanned by $\boldsymbol{u}_1^+, \boldsymbol{u}_1^-, ..., \boldsymbol{u}_K^+, \boldsymbol{u}_K^-$ is a subspace of the space spanned by the vectors $\mathbf{1}_{\Omega_1}^+, \mathbf{1}_{\Omega_1}^-, ..., \mathbf{1}_{\Omega_K}^+, \mathbf{1}_{\Omega_K}^-$. And since the size of the projection onto a subspace is always less than or equal to the size of the projection onto the larger space, we have (noting we are projecting onto the spaces orthogonal to these spaces)

$$\|(\boldsymbol{I} - \Gamma)\boldsymbol{w}\|^2 \leq \|(\boldsymbol{I} - \overline{\Pi})\boldsymbol{w}\|^2,$$

where $\overline{\Pi}$ is the projection onto the space spanned by $\boldsymbol{u}_1^+, \boldsymbol{u}_1^-, ..., \boldsymbol{u}_K^+, \boldsymbol{u}_K^-$. It follows that

$$\|(\boldsymbol{I} - \overline{\Pi})\boldsymbol{w}\|^2 \leq 2\|(\boldsymbol{I} - \Pi)\boldsymbol{w}\|^2 + 2\|(\overline{\Pi} - \Pi)\boldsymbol{w}\|^2$$

$$\leq \frac{2c\|\boldsymbol{w}\|^2}{n^{3/2}} + 2\|\overline{\Pi} - \Pi\|^2 \cdot \|\boldsymbol{w}\|^2$$

where in the second inequality applies the result from Theorem 4.1. From the classic Davis-Kahan Sin-Theta theorem (Stewart & Sun, 1990) applied to the spectral norm, and that $\Delta_{2K} = \Theta(1/n)$ (Lemma C.1) to obtain

$$\|\Pi - \overline{\Pi}\| \leq \frac{\|\boldsymbol{L} - \overline{\boldsymbol{L}}\|}{\Delta_{2K}} \leq n \cdot O\left(\frac{\log^{1/2} n}{n^{1+\rho/2}}\right) = O\left(\frac{\log^{1/2} n}{n^{\rho/2}}\right).$$

Therefore,

$$\|(\boldsymbol{I} - \overline{\Pi})\boldsymbol{w}\|^2 \leq \frac{c\|\boldsymbol{w}\|^2}{n^{\rho}},$$

where $c$ is a constant. This completes the proof. $\qquad\square$

## E.2. Cluster Separation

While Lemma E.1 shows that the vector $\boldsymbol{x}$ has a cluster structure, it does not show that these clusters are separated. In this section, we show that given a solution vector $\boldsymbol{x}$, the distance between cluster means $|\mu_i - \mu_j|$ remains large enough. The main idea of the proof is to show that a sufficient amount of "initial separation" between cluster means must exist because the initialization is random and then prove it persists until the end of the algorithm.

Let $\Pi = \boldsymbol{V}\boldsymbol{V}^\top$ be the projection onto the top $2K$ eigenspace of $\boldsymbol{L}$ and consider a fixed column $\boldsymbol{w}^{(t)}$ of $\boldsymbol{W}^{(t)}$. Throughout this analysis, we focus on the evolution of the projected vector $\boldsymbol{z}^{(t)} = \Pi\boldsymbol{w}^{(t)}$. The analysis will contain three components: first, we study the *expected* update of $\boldsymbol{z}^{(T)}$, where $T$ is the total number of steps in the algorithm, and show that it is close to $(\boldsymbol{I} + \eta\boldsymbol{L})^T\boldsymbol{z}^{(0)}$. Second, we use concentration inequalities for vector-valued martingales to prove that $\boldsymbol{z}^{(T)}$ is close to this "linear" update for small enough initializations. Finally, we argue that this implies the desired separation properties.

The following technical lemma says that when the algorithm ends, the expected value of $\boldsymbol{z}^{(T)}$ is approximately $(\boldsymbol{I} + \eta\boldsymbol{L})^T\boldsymbol{z}^{(0)}$.

*Lemma* E.2. *Suppose that Algorithm 1 is run for a total of $T = \frac{1}{10\eta\Delta_{2K}} \times O(\log n)$ iterations. Then*

$$\mathbb{E}[\boldsymbol{z}^{(T)} \mid \boldsymbol{W}^{(0)}] = (\boldsymbol{I} + \eta\boldsymbol{L})^T\boldsymbol{z}^{(0)} + \eta\sum_{t=0}^{T-1}(\boldsymbol{I} + \eta\boldsymbol{L})^{T-t-1}\Pi \cdot \mathbb{E}[\boldsymbol{M}_t\boldsymbol{w}^{(t)}) \mid \boldsymbol{W}^{(0)}].$$

*Proof.* Using the update for $\boldsymbol{w}^{(t)}$ in Equation (10), multiplying by $\Pi$, and taking the conditional expectation of both sides gives

$$\mathbb{E}[\boldsymbol{z}^{(t+1)} \mid \boldsymbol{w}^{(t)}] = (\boldsymbol{I} + \eta\boldsymbol{L})\boldsymbol{z}^{(t)} + \eta\Pi\boldsymbol{M}_t\boldsymbol{w}^{(t)}, \qquad (27)$$

where we have used the fact that $\Pi$ commutes with $\boldsymbol{L}$. Applying Equation (27) iteratively, we obtain

$$\mathbb{E}[\boldsymbol{z}^{(T)} \mid \boldsymbol{W}^{(0)}] = (\boldsymbol{I} + \eta\boldsymbol{L})^T\boldsymbol{z}^{(0)} + \eta\sum_{t=0}^{T-1}(\boldsymbol{I} + \eta\boldsymbol{L})^{T-t-1}\Pi \cdot \mathbb{E}[\boldsymbol{M}_t\boldsymbol{w}_t \mid \boldsymbol{W}^{(0)}].$$

This completes the proof. $\qquad\square$

In the following lemma, we perform a vector-valued martingale analysis to show that $\boldsymbol{z}^{(T)}$ is close to the expected value expressed in Lemma $E.2$. The proof is quite technical so we defer it to Appendix F.1.

*Lemma* E.3. *Let $\delta \in (0,1)$. Suppose that Algorithm 1 is run for $T = \frac{1}{10\eta\Delta_{2K}} \times O(\log n)$ total iterations. Furthermore, suppose that $\eta$ and $\tau_0^2$ are chosen as in Theorem 4.1. Then with probability at least $1 - \delta$, we have*

$$\left\|(\boldsymbol{I} + \eta\boldsymbol{L})^T\boldsymbol{z}^{(0)} - \boldsymbol{z}^{(T)}\right\| = O\left(\frac{\log n}{n}\right).$$

Using Lemma E.3, we can now show that the cluster means remain separated.

*Lemma* E.4. *Suppose that $\mathcal{G}, \rho, M, T$ be as defined in Theorem 5.1, and let $\boldsymbol{w}, \boldsymbol{x}$ be the resulting embeddings. Then with probability $\geq 0.95$ over the initialization and randomness of the algorithm, we have for all $i, j \in [K]$ and $i \neq j$, that*

$$|\mu_{\Omega_i} - \mu_{\Omega_j}| \geq \frac{1}{60K^2\sqrt{n}}\|\boldsymbol{w}\|.$$

The proof is deferred to Section F.2.

# F. Proofs from Appendix E

We now supply the proofs of some technical lemmas from Appendix E.

## F.1. Proof of Lemma E.3

Recall that the update of the projection vector $\boldsymbol{z}^{(t)}$ is

$$\boldsymbol{z}^{(t)} = \boldsymbol{z}^{(t-1)} + \eta\boldsymbol{V}\boldsymbol{V}^\top\boldsymbol{L}_{(i,j)}\boldsymbol{w}^{(t-1)} + \eta\boldsymbol{V}\boldsymbol{V}^\top\boldsymbol{E}_{(i,j)}\boldsymbol{w}^{(t-1)}.$$

Equation (27) shows that the expected value of $\boldsymbol{z}^{(T)}$ is

$$\mathbb{E}[\boldsymbol{z}^{(t+1)} \mid \boldsymbol{w}^{(t)}] = (\boldsymbol{I} + \eta\boldsymbol{L})\boldsymbol{z}^{(t)} + \eta\Pi\boldsymbol{M}_t\boldsymbol{w}^{(t)}.$$

Our goal is to show that after $T$ iterations, that $\boldsymbol{z}^{(T)} \approx (\boldsymbol{I} + \eta\boldsymbol{L})^T\boldsymbol{z}^{(0)}$.

Let us define:

$$X_t := (\boldsymbol{I} + \eta\boldsymbol{L})^{T-t+1}\boldsymbol{z}^{(t-1)} - (\boldsymbol{I} + \eta\boldsymbol{L})^{T-t}\boldsymbol{z}^{(t)}.$$

Note that $X_t$ is a random vector in $\mathbb{R}^{2n}$ and by construction $\sum_t X_t = \boldsymbol{z}^{(T)} - (\boldsymbol{I} + \eta\boldsymbol{L})^\top\boldsymbol{z}^{(0)}$. We are ultimately interested in bounding the magnitude of this summation. To use vector martingale inequalities directly, we need $\mathbb{E}[X_t \mid \mathcal{F}_{t-1}] = 0$ for all $t$, where $\mathcal{F}_t$ denotes the filtration associated with our random process $\boldsymbol{W}^{(0)}, ..., \boldsymbol{W}^{(t)}$, in addition to independence across $t$. While the latter property holds for our updates, we do not have $\mathbb{E}[X_t \mid \mathcal{F}_{t-1}] = 0$. Thus, we define a new random variable $Y_t$ as

$$Y_t = X_t - \eta(\boldsymbol{I} + \eta\boldsymbol{L})^{T-t}\boldsymbol{V}\boldsymbol{V}^\top\boldsymbol{M}_{t-1}\boldsymbol{w}^{(t-1)}.$$

Notice that $\mathbb{E}[Y_t \mid \mathcal{F}_{t-1}] = 0$ for all $t$.

Now, another problem arises. In order to use vector martingale inequalities directly, $|Y_t|$ needs to be bounded almost surely. While $|Y_t| \leq 2\eta\tau_t$, the upper bound on $\tau_t$ has a non-zero probability of failure.

Therefore, we define a new random sequence. Let $\mathcal{E}_t$ be the event that $|\tau_t| \leq \gamma$, where $\gamma > 0$. We construct the following sequence:

$$Y_t' := \begin{cases} Y_t, & \text{if } \mathcal{E}_{t-1} \text{ occurs}, \\ 0, & \text{otherwise}. \end{cases}$$

Note that $Y'$ is a well-defined random sequence in that is respects the filtration $\mathcal{F}_t$. Also, notice that $\mathbb{E}[Y_t' \mid \mathcal{F}_{t-1}] = 0$ and $|Y_t'| \leq \gamma$ almost surely. Therefore, we can directly apply vector martingale inequalities. Using the vector Azuma's inequality (see Theorem 1.8 in (Hayes, 2003)), we obtain

$$\Pr\left[\left\|\sum_{t=1}^T Y_t'\right\| \geq \gamma\sqrt{2T\log\left(\frac{20}{\delta'}\right)}\right] \leq \delta'.$$

Finally, recall from Lemma C.4 that with probability at least $1 - \delta''$ that

$$\max_{t \in [T]} \| \boldsymbol{W}^{(t)} \|_{2,\infty}^2 \leq 3^M \| \boldsymbol{W}^{(0)} \|_{2,\infty}^2,$$

given our choice of parameters $T$ and $\eta$. It follows that,

$$\Pr \left[ \left\| \sum_{t=1}^{T} Y_t \right\| \geq \gamma \sqrt{2T \log \left( \frac{20}{\delta'} \right)} \right] \leq \Pr \left[ \left\| \sum_{t=1}^{T} Y_t' \right\| \geq \gamma \sqrt{2T \log \left( \frac{20}{\delta'} \right)} \right]$$

$$+ \Pr \left[ \sum_{t=1}^{T} Y_t \neq \sum_{t=1}^{T} Y_t' \right]$$

$$\leq \delta' + \delta''.$$

If we choose $\gamma = 3^{M/2} \tau_0$, then $|Y_t| \leq 2\eta 3^{M/2} \tau_0$. Given our choice of $\eta, \tau_0, T$, and $M$, we have

$$\left\| \sum_{t=1}^{T} Y_t \right\| \leq \gamma \sqrt{2T \log \left( \frac{20}{\delta'} \right)} \leq 2\eta 3^{M/2} \tau_0 \sqrt{2T \log \left( \frac{20}{\delta'} \right)} = O \left( \frac{1}{n^{5/2}} \right).$$

Now, note that,

$$\left\| \sum_{t=1}^{T} Y_t \right\| \geq \left\| (\boldsymbol{I} + \eta \boldsymbol{L})^T \boldsymbol{z}^{(0)} - \boldsymbol{z}^{(T)} \right\| - \left\| \sum_{t=1}^{T} \eta (\boldsymbol{I} + \eta \boldsymbol{L})^{T-t} \boldsymbol{V} \boldsymbol{V}^\top \boldsymbol{M}_{t-1} \boldsymbol{w}^{(t-1)} \right\|.$$

In order to bound the second term on the right-hand side, we utilize Lemma C.4 again, which says that with probability at least $1 - \delta''$ that $\max_{t \in [T]} \tau_t^2 \leq 3^M \tau_0^2$. Let $\mathcal{E}$ be the event that $\max_{t \in [T]} \tau_t^2 \leq 3^M \tau_0^2$. Conditioned on the event $\mathcal{E}$, the second term can be bounded as

$$\left\| \sum_{t=1}^{T} \eta (\boldsymbol{I} + \eta \boldsymbol{L})^{T-t} \boldsymbol{V} \boldsymbol{V}^\top \boldsymbol{M}_{t-1} \boldsymbol{w}^{(t-1)} \right\| \leq \sum_{t=1}^{T} \eta \| \boldsymbol{I} + \eta \boldsymbol{L} \|^{T-t} \cdot \| \boldsymbol{M}_{t-1} \| \cdot \| \boldsymbol{w}^{(t-1)} \|$$

$$\leq \sum_{t=1}^{T} \eta (1 + \eta \sigma_1)^{T-t} \cdot \frac{(1 + s_n) r^{1/2}}{n^{1/2}} \cdot \tau_{t-1}^3$$

$$\leq T \cdot \eta e^{\frac{\eta}{n} \cdot T} \cdot \frac{(1 + s_n) r^{1/2}}{n^{1/2}} \cdot 3^{3M/2} \tau_0^3.$$

Given our choice of parameters $T, M, \eta$, and $\tau_0$, we have

$$\left\| \sum_{t=1}^{T} \eta (\boldsymbol{I} + \eta \boldsymbol{L})^{T-t} \boldsymbol{V} \boldsymbol{V}^\top \boldsymbol{M}_{t-1} \boldsymbol{w}^{(t-1)} \right\| = O \left( \frac{\log n}{n} \right).$$

Set $\delta' = \delta/2$ and $\delta'' = \delta/2$. It follows that with probability at least $1 - \delta$ that

$$\left\| (\boldsymbol{I} + \eta \boldsymbol{L})^T \boldsymbol{z}^{(0)} - \boldsymbol{z}^{(T)} \right\| \leq O \left( \frac{1}{n^{5/2}} \right) + O \left( \frac{\log n}{n} \right) = O \left( \frac{\log n}{n} \right).$$

This concludes the proof.

### F.2. Proof of Lemma E.4

*Proof.* Let us define the vector $\boldsymbol{u} = \sqrt{\frac{K}{2n}} \left( \begin{bmatrix} \mathbf{1}_{\Omega_i} \\ \mathbf{0}_{n \times 1} \end{bmatrix} - \begin{bmatrix} \mathbf{1}_{\Omega_j} \\ \mathbf{0}_{n \times 1} \end{bmatrix} \right)$ for two clusters $i$ and $j$. Let $\{\sigma_i, \boldsymbol{v}_i\}_{i \in [n]}$ be the eigenvalue-eigenvector pairs of $\boldsymbol{L}$. From Lemma C.1 we know that $\| \boldsymbol{L} - \overline{\boldsymbol{L}} \| = \widetilde{O} \left( \frac{1}{n^{1+\rho/2}} \right)$. Also, since $s_n = 1$ by assumption, we have

$$\sigma_i \in \begin{cases} \left[ \frac{1}{n} - \frac{1}{n^{1+\rho/2}}, \frac{1}{n} + \frac{1}{n^{1+\rho/2}} \right], & \text{for } i \leq 2K, \\ \left[ 0, \frac{1}{n^{1+\rho/2}} \right], & \text{for } i > 2K. \end{cases}$$

Let $\{\overline{v_i}\}_{i \in [2K]}$ be the eigenvectors of $\overline{L}$. By definition $u \in \text{span}(\overline{v}_1, ..., \overline{v}_{2K})$. This implies that

$$\|u - \Pi u\| = \|(\overline{\Pi} - \Pi)u\| \leq \frac{1}{n^{\rho/2}}.$$

Let $u = \sum_i \alpha v_i$ be the expansion of $u$ in the eigenbasis of $L$. Because of the above, $u$ mostly lies in the span of the top $2K$ eigenvectors of $L$, and thus,

$$\sum_{i \leq 2K} \alpha_i^2 \geq \frac{1}{2}\|u\|^2 = \frac{1}{2}.$$

Using the decomposition of $u$ above, we also have

$$(I + \eta L)^T u = \sum_{i=1}^n (1 + \eta\sigma_i)^T \alpha_i v_i.$$

By choice $z^{(0)}$ is a Gaussian vector in the span of $\{v_1, v_2, ..., v_{2K}\}$, and because of rotational invariance, $\langle z^{(0)}, (I+\eta L)^T u \rangle$ is distributed as a univariate Gaussian with variance

$$\sigma^2 = \frac{\|z^{(0)}\|^2}{2K} \sum_{i=1}^{2K} (1 + \eta\sigma_i)^{2T} \alpha_i^2 \geq \frac{\|z^{(0)}\|^2}{8K}\left(1 + \frac{\eta}{n}\right)^{2T}.$$

In the last step, we used the fact that $\sum_{i \leq 2K} \alpha_i^2 \geq 1/2$, along with the fact that

$$\left(1 + \frac{\eta}{n} - \frac{\eta}{n^{1+\rho/2}}\right)^{2T} \geq \frac{1}{2}\left(1 + \frac{\eta}{n}\right)^{2T},$$

for our choice of $T$. Thus, with probability at least $1 - \frac{1}{20K^2}$ (over the initialization),

$$\langle z^{(0)}, (I + \eta L)^T u \rangle \geq \frac{1}{20K^2}\left(1 + \frac{\eta}{n}\right)^T \frac{\|z^{(0)}\|}{\sqrt{2K}}.$$

Finally, we note that

$$\|(I + \eta L)^T z^{(0)}\|^2 \leq \left(1 + \frac{\eta}{n} + \frac{\eta}{n^{1+\rho/2}}\right)^{2T} \|z^{(0)}\|^2 \leq 2\left(1 + \frac{\eta}{n}\right)^{2T}\|z^{(0)}\|^2.$$

Together, with Lemma E.3, these prove that

$$\langle u, z^{(T)} \rangle \geq \langle u, (I + \eta L)^T z^{(0)} \rangle - O\left(\frac{1}{n}\right) \geq \frac{1}{40K^2\sqrt{2K}}\|z^{(T)}\|.$$

Since $\langle u, z^{(T)} \rangle = \sqrt{\frac{2n}{K}}|\mu_i - \mu_j|$ by definition, we have with probability at least $1 - \frac{1}{20K^2}$ that

$$|\mu_i - \mu_j| \geq \frac{1}{40K^2\sqrt{n}}\|z^{(T)}\|.$$

Taking a union bound over all the $\binom{K}{2}$ pairs of clusters, and observing that by Theorem 4.1, $\|z^{(T)}\| \geq \frac{2}{3}\|w\|$, the lemma follows. $\qquad \square$

### F.3. Recovery algorithm

Suppose we have $n$ points $x_1, x_2, \ldots, x_n$ on a line that are partitioned into sets $\Omega_1, \Omega_2, \ldots, \Omega_K$ of size $n/K$ each. Let $\mu^{(t)}$ be the mean of the points in $\Omega_t$.

*Theorem F.1. Let points $x_1, \ldots, x_n$ and $\Omega_1, \Omega_2, \ldots, \Omega_K$ be as above. Let $0 < \epsilon < \frac{1}{4K}$ be a parameter (possibly unknown), and let $\Delta > 0$ be given. Suppose we have the following conditions:*

1. *For all $s \neq t \in [K]$, we have $|\mu^{(s)} - \mu^{(t)}| > 6\Delta$.*

2. *The sum of squared distances to the cluster means satisfies*

$$\sum_{t \in [K]} \sum_{j \in \Omega_t} \|x_j - \mu^{(t)}\|^2 \leq \epsilon n \Delta^2. \tag{28}$$

*Then there is an efficient algorithm that outputs disjoint sets $J_1, J_2, \ldots, J_K$ such that for some permutation $\pi$ of $[K]$, we have $\forall t$, $|J_t \cap \Omega_{\pi(t)}| \geq \frac{n}{K} - \epsilon n$.*

Without loss of generality, assume that the points are sorted in increasing order, $x_1 \leq x_2 \leq \cdots \leq x_n$. Additionally, we can assume that the cluster centers are in increasing order (by re-numbering the clusters), so $\mu^{(1)} < \mu^{(2)} < \cdots < \mu^{(K)}$. By hypothesis (the separation condition in the theorem), we have that the intervals $\Gamma_t := [\mu^{(t)} - \Delta, \mu^{(t)} + \Delta]$ are all disjoint. In fact, the distance between two intervals is $> 4\Delta$. Define

$$I_t := \{j \in \Omega_t : |x_j - \mu^{(t)}| \leq \Delta\}.$$

We call these the *inliers* for cluster $i$ and call $O_t := \Omega_t \setminus I_t$ the set of outliers. A first structural observation is the following.

*Observation* F.2. The total number of outliers is $\leq \epsilon n$. I.e., $\sum_t |O_t| \leq \epsilon n$.

*Proof.* This is simply because for any outlier $j$ in $O_t$, we have $|x_j - \mu^{(t)}| > \Delta$ by definition, and thus if we have more than $\epsilon n$ outliers in total, we get a contradiction to Equation (28). $\square$

Consider the following algorithm:

1. Find the smallest $i$ such that $[x_i, x_i + 2\Delta]$ contains at least $\frac{n}{K} - \epsilon n$ points.

2. Let $J$ be the set of all points in the interval $[x_i, x_i + 4\Delta]$. Mark all the points in $J$ into one cluster, remove all the points with coordinate value $\leq (x_i + 4\Delta)$, and repeat.

We note that the choice of $4\Delta$ in the second step (as opposed to $2\Delta$ on the first step) is somewhat important. Let us now analyze the algorithm.

*Lemma* F.3. Let $S \subset [n]$ such that $I_t \cup I_{t+1} \cup \cdots \cup I_K \subseteq S$, for some index $t$. Suppose also that $S \cup I_j = \emptyset$ for $j < t$. Let $i \in S$ be the least index such that $[x_i, x_i + 2\Delta]$ contains $\geq \frac{n}{K} - \epsilon n$ points from $S$. Suppose $J$ is the set of points contained in $[x_i, x_i + 4\Delta]$. Then we have:

1. $I_t \subseteq J$

2. $I_{t+1} \cup \cdots \cup I_K \subseteq S \setminus J$.

In other words, the lemma says that $J$ contains all of the inliers $I_t$ (the leftmost set), and it does not contain any of the subsequent inlier sets.

*Proof.* The key is to observe that any interval $[x_i, x_i + 4\Delta]$ can only intersect one of the intervals $\Gamma_j$ (due to the distance observation from earlier). So if we can show that $[x_i, x_i + 4\Delta]$ intersects $\Gamma_t$, this immediately implies the second part of the lemma. (I.e., the interval cannot contain any of the inliers from later clusters.)

To see the first part, we in fact prove that $[x_i, x_i + 2\Delta]$ intersects $\Gamma_t$, more specifically $x_i \leq \mu^{(t)} - \Delta \leq x_i + 2\Delta$, which implies $[x_i, x_i + 4\Delta]$ *contains* $\Gamma_t$ since the length of $\Gamma_t$ is only $2\Delta$. The former statement holds because $S$ by assumption does not overlap with any of the inliers $I_j$ for $j < t$. The total number of outliers is only $\epsilon n$ from the observation above. Since $\frac{n}{K} - \epsilon n > 0$, the remaining points must come from one of $I_t, I_{t+1}, \ldots$. Since we are choosing the leftmost $x_i$, the claim follows. $\square$

Iteratively using Lemma F.3, Theorem F.1 follows.

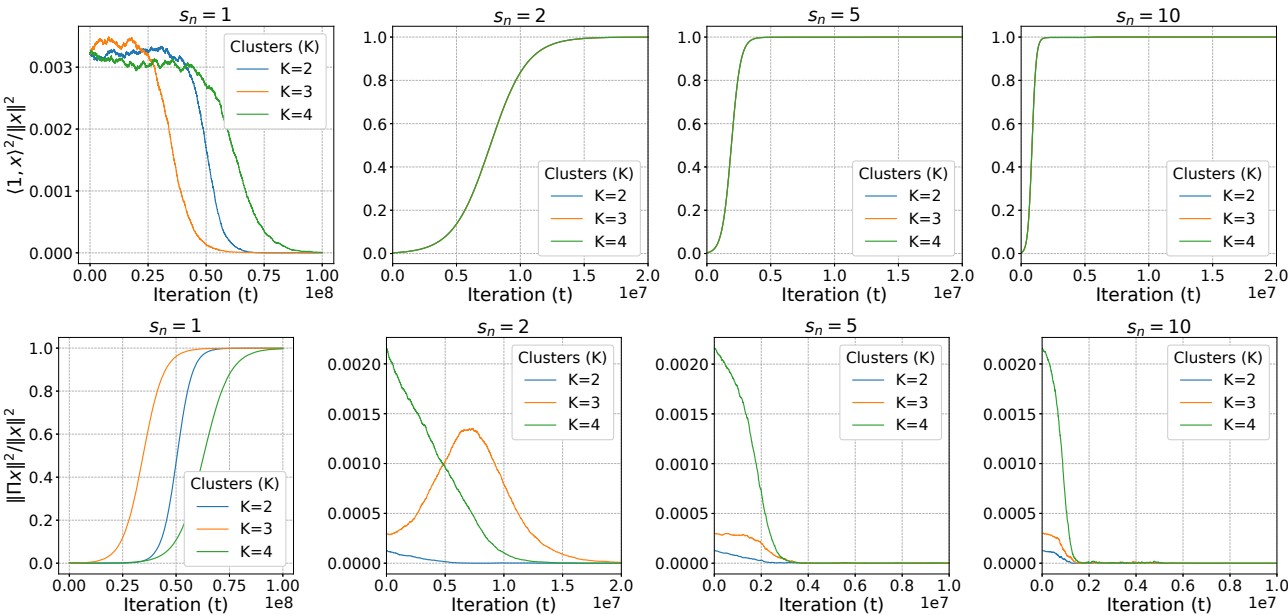

*Figure 2.* Convergence and divergence of a one-dimensional node embedding $x$ throughout training via Skip-gram with negative sampling to (top) the all-ones vector $\boldsymbol{u}_1 = \boldsymbol{1}$, and (bottom) the space spanned by the ground-truth vectors $\boldsymbol{u}_2 = \boldsymbol{1}_{\Omega_1} - \boldsymbol{1}_{\Omega_2}$, $\boldsymbol{u}_3 = \boldsymbol{1}_{\Omega_2} - \boldsymbol{1}_{\Omega_3}$ and $\boldsymbol{u}_4 = \boldsymbol{1}_{\Omega_3} - \boldsymbol{1}_{\Omega_4}$ for an increasing number of negative samples.

# G. Additional Experimental Results

In this section, we provide additional experimental results to supplement those in Section 6. In Section G.1, we provide an additional figure showing the resulting communities learned by the embeddings. In Section G.2, we make an observation about the effect that the initialization radius has on the rate at which the embeddings converge.

## G.1. Effect of the Negative Sampling Parameter on Trajectories

The theoretical analysis of the previous sections shows that the embedding vector $\boldsymbol{w}$ converges to a vector that lies in the span of the top $2K$ singular vectors of the linear portion of the update $\boldsymbol{L} = \mathbb{E}\left[\boldsymbol{L}_{(i,j)}\right]$ (e.g., see Equation (13)). Because the off-diagonal matrices of $\boldsymbol{L}$ are themselves approximately block matrices, the eigenvalue corresponding to $\boldsymbol{u}_1$ has a value of $\lambda_1 = \frac{1}{2n}(1 - s_n)$ while the remaining eigenvalues are $\sim 1/n$ (see the proof of Lemma C.1 for details regarding the spectrum of $\boldsymbol{L}$ in Appendix D). Importantly, $\lambda_1$ depends on the number of negative samples. Therefore, if the negative sampling parameter $s_n = 1$, then the leading eigenvalue is $\lambda_1 = 0$. In this scenario, the $x$-embedding vector is expected to converge to the space orthogonal to $\boldsymbol{u}_1 = \boldsymbol{1}$, which contains the community structure of the graph. This space consists of the ground-truth vectors $\boldsymbol{u}_2 = \boldsymbol{1}_{\Omega_1} - \boldsymbol{1}_{\Omega_2}$, $\boldsymbol{u}_3 = \boldsymbol{1}_{\Omega_2} - \boldsymbol{1}_{\Omega_3}$, and $\boldsymbol{u}_4 = \boldsymbol{1}_{\Omega_3} - \boldsymbol{1}_{\Omega_4}$ (when $K = 4$), and we denote the projection onto this space as $\Pi$. Conversely, as the parameter $s_n$ increases, we would expect the $x$-embedding vector to converge to $\boldsymbol{u}_1$, the all-ones vector, which does not contain any structural information for community detection. This section presents experimental evidence demonstrating the impact of $s_n$ on the quality of the learned embeddings.

We sample a graph from a stochastic block model with $n = 1200$ nodes and $K = 2, 3, 4$ communities using parameters $p = 0.6$ and $q = 0.1$. We construct a co-occurrence matrix as defined by Definition 2.1 and learn one-dimensional embeddings via Skip-gram with negative sampling (e.g., see Algorithm 1) with a step-size of $\eta = 1/n$ and an initial embedding radius of $\tau = 1/n^2$.

Figure 2 shows that the fraction of the mass of $x$ that lies on $\Pi$, as well as cosine between $x$ and the all-ones vector, throughout training for increasing values of the negative sampling parameter $s_n$. As expected, the $x$-embedding aligns with the space orthogonal to $\boldsymbol{u}_1$ when $s_n = 1$, and aligns with $\boldsymbol{u}_1$ vector $\boldsymbol{v}_1$ as $s_n$ increases.

We provide the resulting $x$-embeddings with their labeled ground-truth communities in Figure 3 for graphs with $K = 2, 3$ and $4$ communities, which show that SGNS can recover communities when $s_n = 1$, even with one-dimensional embeddings. This suggests that smaller values of $s_n$ may be more conducive to community recovery problems, and supports prior works

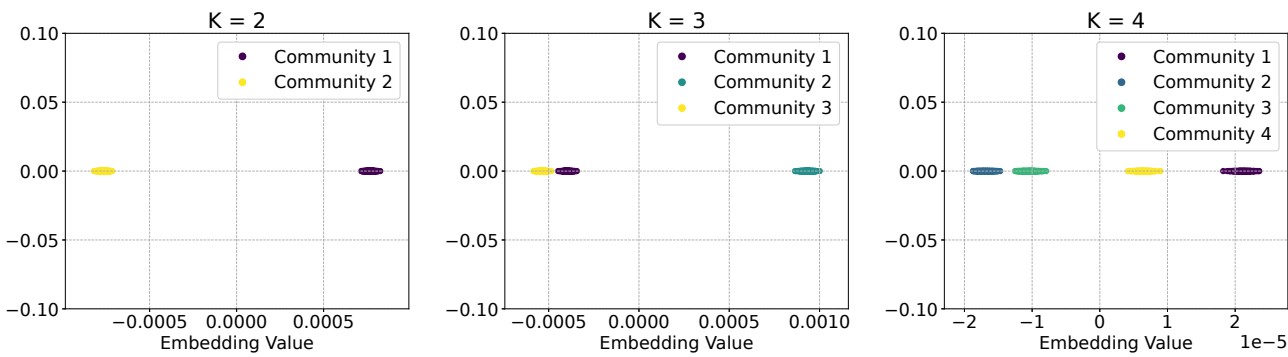

*Figure 3.* The $x$-embeddings learned via SGNS with a parameter $s_n = 1$ and their ground-truth community labeling.

that suggest that an embedding's performance on downstream tasks may be negatively impacted if the number of negative samples is too large (e.g., see Saunshi et al. (2019)).

We also provide the trajectories of two nodes throughout the course of training in Figure 4 on a graph drawn from an SBM with $n = 100$ nodes and $K = 2$ communities using parameters $p = 0.6$ and $q = 0.1$. We ran SGNS for $250,000$ iterations with a step-size of $\eta = 1/n$ and an initial embedding radius of $\tau = 1/n^2$, learning two-dimensional embeddings.

Furthermore, Theorem 4.1 sets the epoch length to $T = \frac{1}{6(1+s_n)^2} \cdot \frac{n}{\eta}$, suggesting that the convergence rate should decrease as the parameter $s_n$ increases. This is supported impirically by Figure 1, but we leave a more in-depth analysis of the effect that $s_n$ has on the rate of convergence to future work.

We observed similar behavior across smaller graphs: Figure 5 shows results for an SBM with n=100 (K=2,4), and we conducted additional trials with $n = 400$ ($K = 4$) and $n = 600$ ($K = 3$). Across all cases, embeddings consistently converge to an informative space when $s_n = 1$ and a non-informative space when $s_n > 1$.

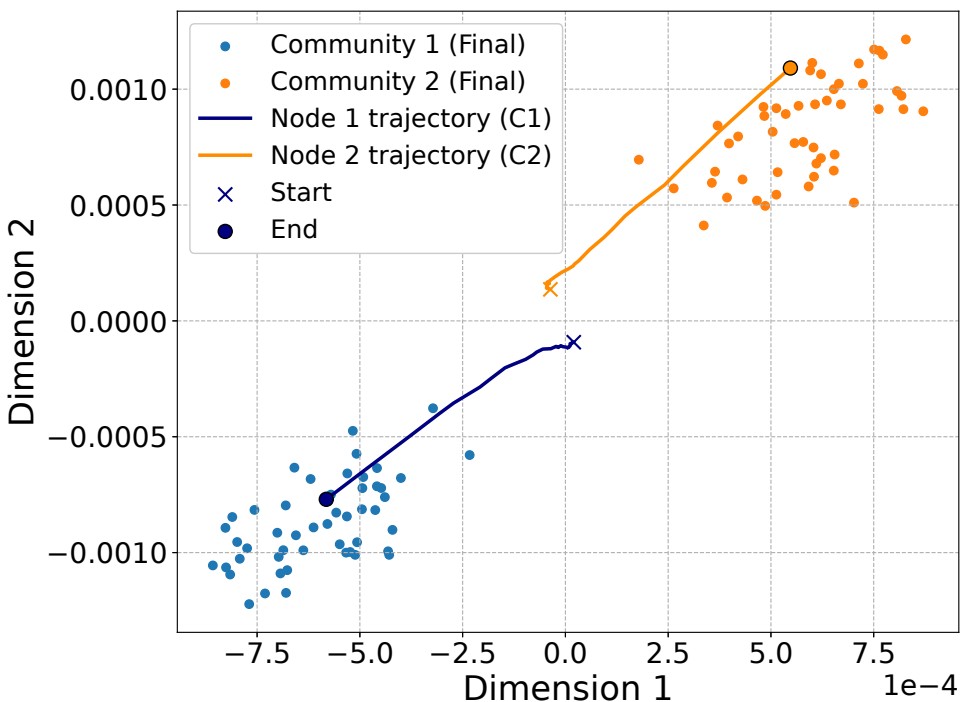

*Figure 4.* The trajectories of two two-dimensional $x$-embeddings learned via SGNS belonging to two nodes, one belonging to each community. For context, the final embeddings for all nodes are shown along with their ground-truth community labeling.

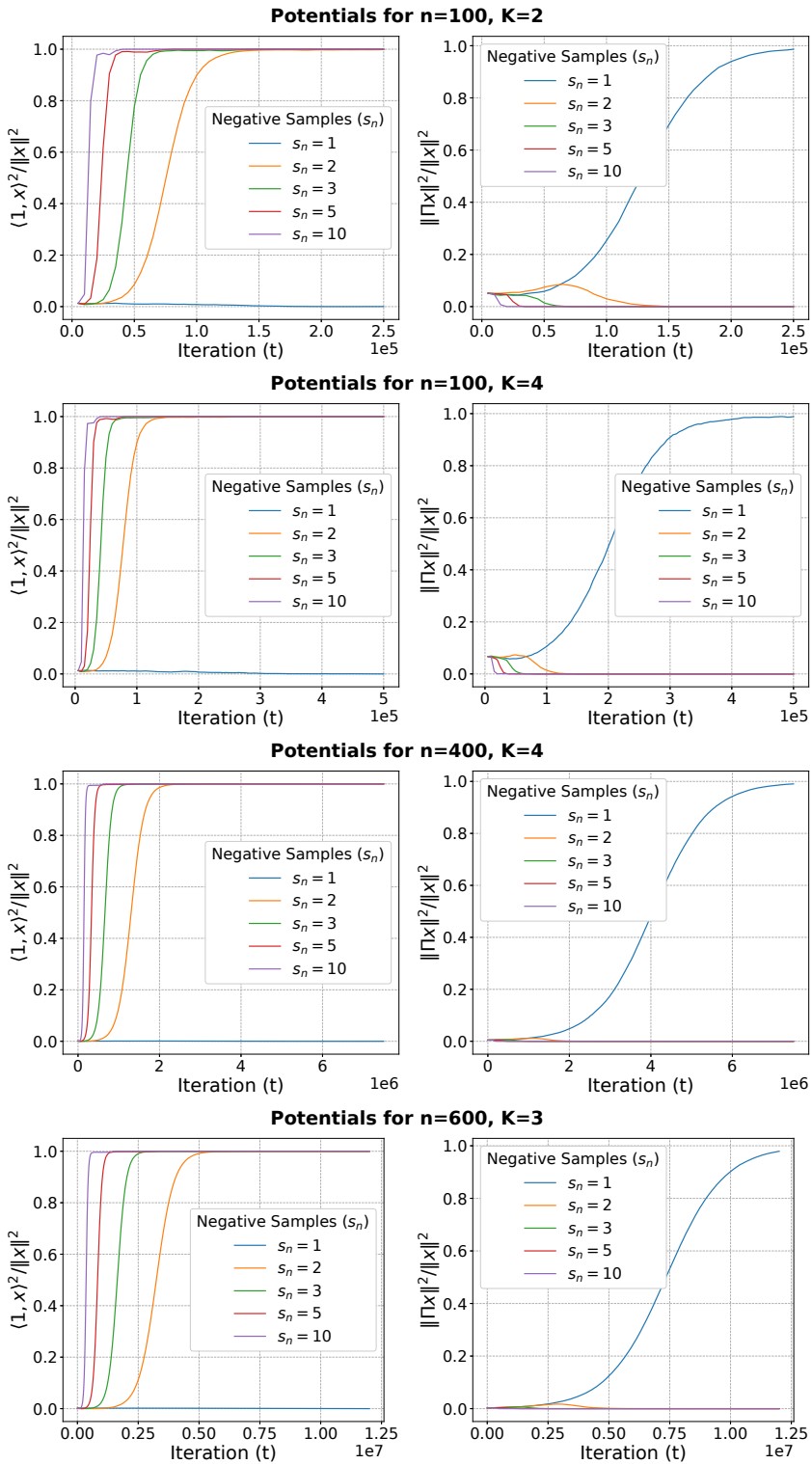

*Figure 5.* Convergence and divergence of a one-dimensional node embedding $x$ throughout training via Skip-gram with negative sampling to (left column) the all-ones vector, and (right column) the space spanned by the ground-truth vectors $\Pi$ for an increasing number of negative samples. We use graphs drawn from an SBM with $n = 100, 400, 600$ nodes, and we show results for $K = 2, 3, 4$ communities.

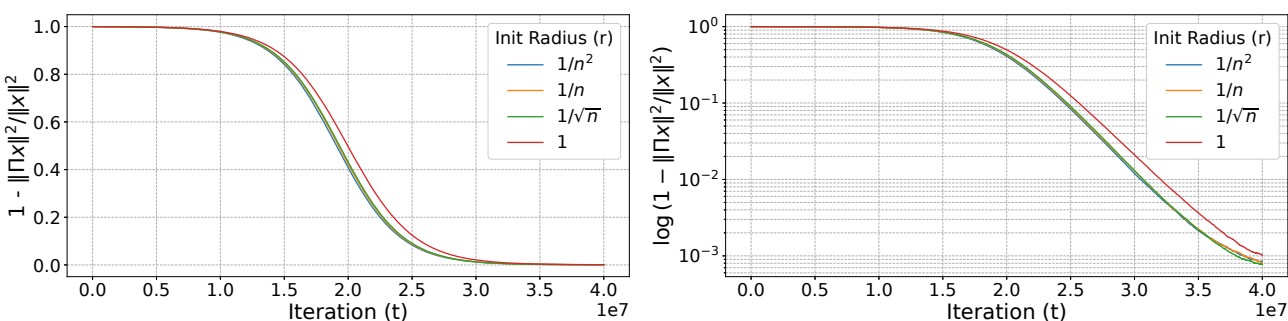

*Figure 6.* (Left) The fraction of the mass of the $x$-embedding that is not aligned with the community structure. (Right) The log of the fraction of mass that is not aligned with the community structure.

### G.2. Effect of Initialization Radius on Convergence

One limitation of the analysis is the requirement that embeddings be initialized within a small ball, with a radius that is inverse polynomial in $n$. This raises a natural question: to what extent can this initialization constraint be relaxed? In this section, we empirically investigate how the initialization scale influences both the rate of convergence and the properties of the resulting subspace.

We consider a graph drawn from a stochastic block model with $n = 1000$ nodes and $K = 2$ communities, fixing the negative sampling parameter $s_n = 1$ and the learning rate $\eta = 1/n$. To assess the sensitivity of the Algorithm to the radius of initialization, we vary the radius of the initial embeddings: $1/n^2, 1/n, 1/\sqrt{n}$, and $1$. As in Section G.1, we define the alignment "error" as

$$error = 1 - \|\Pi x\|^2/\|x\|^2.$$

This represents the fraction of mass that is not yet aligned with the community structure.

Figure 6 displays alignment error (left) and its corresponding logarithm (right). The results indicate that learning begins with an initial "ramp-up" phase, followed by a "fast" alignment phase. During this second stage, the embeddings align with the community-structured subspace at an exponential rate, as evidenced by the log-linear convergence.

Notably, the initialization radius of the embeddings appears to affect neither the asymptotic subspace nor the overall convergence rate. These observations suggest a compelling open question: whether the current analysis can be decoupled from the initialization radius, or perhaps even from the embedding magnitude across all iterations.

