# OpenReview forum: "On the Optimization Trajectory of DeepWalk Embeddings"
_ICML.cc/2026/Conference — ICML 2026 spotlight_

### Official Review · Reviewer_dbB8 · 2026-03-07

**Soundness:** 2
**Presentation:** 2
**Significance:** 2
**Originality:** 3
**Overall Recommendation:** 4
**Confidence:** 4

**Summary:**

This paper is important in theoretically establishing convergence guarantees for the widely used DeepWalk method with skip-gram negative sampling. They show that the learned embeddings converge to a low-rank subspace of the embedding space. As a result, the node embeddings become well clustered. The experimental results show the negative sampling parameters affect both convergence behavior and recovery guarantees and convergence occurs across a relatively wide range of initializations.

**Compliance With Llm Reviewing Policy:**

Affirmed.

**Final Justification:**

After the detailed rebuttal, I think the authors resolve my main concerns, thus I would like to change the final recommendation as "Weak Accept".

**Key Questions For Authors:**

1 The intuition seems a little bit weird to me that you can drop the nonlinear part of the Eq.(10)? Does the analysis suggest that the algorithm could run without the nonlinearity? If so, what role does the softmax component play in the actual algorithm?

2 The conclusion of Theorem 4.1 is interesting. How did you calculate or control the initial embedding radius since the parameters of the model are random initialized?

3 In Theorem 5.1, the lemmas in section E all state a probability bound of >0.95, whereas the theorem reports 0.9. Could the authors clarify the reason for this discrepancy?

4 The paper aims to analyze the trajectory on embedding space, but we don’t have any visualization of embedding space. Could the authors provide visualizations or empirical analyses of the optimization trajectory?

Minor:

1 The normalization term |C| appears to be missing in line 765 in Appendix B.1.

2 The hyper link in the main paper does not jump to the correct theorem.

**Limitations:**

yes

**Strengths And Weaknesses:**

Soundness:

Fair. The theoretical proof is sound, although some steps require further justification. The empirical support seems limited.


Presentation:

Poor. The organization of the paper is somewhat confusing. In particular, in Section 5 the proof of part (2) appears before the proof of part (1), which disrupts the logical flow. Some sections would benefit from clearer structuring and better guidance for the reader. The intuition seems not fully used in the proof of theorem 4.1.


Significance:

Fair. Not sure what are the contributions of the real world that the convergences and recovery could bring for real-world graph representation learning tasks. Besides, the linear approximation introduced in the analysis seems unusual and requires stronger justification.


Originality:

Good. The theoretical results appear to be original contributions, and the paper attempts to provide new theoretical insights into the DeepWalk optimization dynamics.


Weakness:

1 The arrangement of several proofs is confusing, and some derivations are difficult to follow.

2 The visualization and interpretability is missing. The paper discusses convergence to a low-rank subspace and cluster recovery in the embedding space. However, no experimental results visualize the embedding space or illustrate the trajectory of gradient descent. Such analyses could significantly improve interpretability and help validate the theoretical claims.

---

> ### Author Rebuttal · Authors · 2026-03-31
>
> We thank the reviewer for their insightful feedback. We hope our responses below help answer your questions and address your concerns. Please let us know if you have any additional questions.
>
> **Paper organization:** We appreciate this feedback. Of course, we want to give the clearest exposition we can. You are absolutely correct about Section 5, and we have already incorporated your suggestion of discussing the proof of part (1) before the proof of part (2). We also agree that the discussion and sketch proof proceeding Theorem 4.1 could be improved, and we can describe the connection to our intuition more explicitly. We will also outline the proof’s sketch more clearly at the beginning of the sketch to make the order and dependencies between lemmas clearer. We will make these improvements and include them in the final submission.
>
> Again, we want to provide the clearest exposition we can. If there are any other specific sections of the text where you feel the exposition can be improved, we would love to know.
>
> **Linearity:** We did not intend to suggest that we can drop the nonlinear part of Equation (10). But we use the decomposition in Equation (10) to motivate our analysis; if we can manage to keep the nonlinear term small enough, the update will be dominated by the linear term. This is akin to the intuition motivating the analysis of the noisy power method (Hardt & Price, 2015), which shows that if the noise stays small enough, the update is dominated by the linear term rather than the noise. Indeed, controlling the nonlinear portion of the update is what makes the analysis so challenging and requires the novel techniques we introduce, even on simple graph classes like the SBM.
>
> The analysis shows that the stochastic updates converge to a vector that lies in the space spanned by the top eigenvectors of the expected linear term. Does this suggest that one can just use a low-rank SVD of the expected linear term? To recover communities in the SBM, that would require an embedding dimension that is polylogarithmic in the number of communities. This is because each of the top eigenvectors of the expected linear term by themselves only contain partial information about the community structure.  But SGNS embeddings, as we show, only need one-dimensional embeddings. Because they converge to a vector in the space spanned by these eigenvectors, even one-dimensional SGNS embeddings contain all the community information needed to obtain structural results such as Theorem 5.1. There is also a question of practicality: construction of $C$, which is needed to construct $L$ from a large graph is extremely expensive, and $L$’s SVD is expensive, which is why stochastic methods are used in practice and is why we analyze a stochastic version in our paper.
>
> **Embedding Initialization:** We comment on the initialization assumptions in Theorem 4.1 in Appendix C.1. In brief, because Algorithm 1 is assuming that each entry of an embedding is drawn from a normal distribution with small variance, we can use standard concentration inequalities to show that assumption (a) holds with constant probability, and (b) holds with high probability.
>
> **Probability Bounds in Theorem 5.1:** Thank you for catching this discrepancy. We will correct it to $0.95$ in the final version.
>
> **Visualizations of the Embeddings Space and Trajectories:** In Appendix G, we show the final embeddings learned on graphs sampled from an SBM with $K=2,3,$ and $4$ communities. In the final version, we can provide an additional figure in the appendix that maps the trajectories of a few nodes’ embeddings, showing how they start in a small ball and how they move to form communities.
>
> **Minor:**
> * **Missing |C|:** Thank you for catching this typo. We have made the edit in our manuscript.
> * **Broken Hyperlinks:** We apologize for this. We will double check all hyperlinks and make sure they all work before we submit the final version.

---

> > ### Author Rebuttal · Reviewer_dbB8 · 2026-04-01
> >
> > My concern has been fully solved. Therefore, I increase my score.

---

### Official Review · Reviewer_FXka · 2026-03-12

**Soundness:** 4
**Presentation:** 3
**Significance:** 2
**Originality:** 3
**Overall Recommendation:** 4
**Confidence:** 3

**Summary:**

This paper provides a theoretical investigation of the optimization trajectory of deepwalk embeddings generated using negative sampling when they are applied to Stochastic Block Models (SMBs). The key result of the paper (Theorem 5.1) is that the latent blocks in an SBM can be recovered with high probability from the deepwalk embedding matrix learned via stochastic gradient descent. The proof relies on the assumption that node and context embedding norms are small, which leads to gradient updates to approximately linear. This allows for the block structure from the SBM to be recovered from the top 2K eigenvectors of a matrix that is a linear approximation of the gradient update under certain conditions, including a small enough learning rate and initialization radius. The recovery result is based on martingale theory and enables a probabilistic bound on the norm of the embedding matrix. In the experiments, the impact of the number of negative samples on community recovery is analyzed using SBM simulations. The best recovery is found for a small number of negative samples. Moreover, the experiments show that the impact of the initialization radius on the empirical convergence is smaller than required by the theory.

**Compliance With Llm Reviewing Policy:**

Affirmed.

**Final Justification:**

I have updated my recommendation from Weak Reject to Weak Accept after the rebuttal and discussion. I have checked the other reviews and they seem to be consistent regarding a few issues that could be improved if the paper is finally accepted. I recommend the authors to take these comments in consideration to the extension possible in an updated version of the paper.

**Key Questions For Authors:**

- Q1: How do the embeddings generated using the linear approximation of the update compare with DeepWalk embeddings in practice?

- Q2: How are the results presented related with results on autocovariance-based community detection and node embeddings?

- Q3: How does the findings in the paper compare with Harker & Bhaskara (2024)? Can they be used to used to justify negative sampling?

**Limitations:**

Limitations are not explicitly listed and discussed.

**Strengths And Weaknesses:**

**Strengths:**

- S.1. The paper addresses a relevant problem, which is understanding convergence properties of deepwalk embeddings

- S.2. The theoretical results presented in the paper seem novel

- S.3. The paper is well-written and easy to follow to the extent that is possible for a theoretical paper

**Weaknesses:**

- W.1. The theoretical assumptions in the paper are not validated through experiments

- W.2. Relevant related work is not discussed

- W.3. All the plots in the experiments use small fonts and are hard to read

**Detailed discussion:**

This paper is more theoretical than what I consider to be my expertise, so my review is mostly based on my understanding of deepwalk and related embedding methods from a more practical viewpoint. Still, I believe I was able to understand the core ideas of the paper to the level that my review is helpful. I overall enjoyed reading the paper and learned a bit trying to understand its main contributions. Here are some details on the limitations I have identified:

**Theoretical assumptions:**

The paper does not demonstrate whether the theoretical assumptions in the paper cast the resulting deepwalk method useless in practice. For instance, what do embeddings generated using the linear approximation look like? Are they similar to those generated using the original method for the SBM? Even more interesting would be to apply the linear approximation to real graphs and show that the resulting embeddings are still useful. My intuition is that the embeddings will be quite different from each other.

**Related work:**

I believe that the paper fails to make important connections with previous work in the following areas:

Autocovariance-based community detection and graph embeddings: The linearized version of the update seems to be connected with the autocovariance matrix of a random walk process when the number of negative samples is one. There is a lot of work on using autocovariance for community detection and node embedding that is not discussed:

@article{delvenne2010stability,
  title={Stability of graph communities across time scales},
  author={Delvenne, J-C and Yaliraki, Sophia N and Barahona, Mauricio},
  journal={Proceedings of the national academy of sciences},
  volume={107},
  number={29},
  pages={12755--12760},
  year={2010},
  publisher={National Academy of Sciences}
}

@article{schaub2019multiscale,
  title={Multiscale dynamical embeddings of complex networks},
  author={Schaub, Michael T and Delvenne, Jean-Charles and Lambiotte, Renaud and Barahona, Mauricio},
  journal={Physical Review E},
  volume={99},
  number={6},
  pages={062308},
  year={2019},
  publisher={APS}
}

@inproceedings{huang2021broader,
  title={A broader picture of random-walk based graph embedding},
  author={Huang, Zexi and Silva, Arlei and Singh, Ambuj},
  booktitle={Proceedings of the 27th ACM SIGKDD conference on knowledge discovery \& data mining},
  pages={685--695},
  year={2021}
}

There is also work analyzing the optimization trajectory for word2vec embeddings, such as the following:

@article{karkada2025closed,
  title={Closed-Form Training Dynamics Reveal Learned Features and Linear Structure in Word2Vec-like Models},
  author={Karkada, Dhruva and Simon, James B and Bahri, Yasaman and DeWeese, Michael R},
  journal={arXiv preprint arXiv:2502.09863},
  year={2025}
}

Finally, while the paper cites Harker & Bhaskara (2024), which study the softmax case, there is no comparison about the findings in the two papers.


**Plots:**

This is a minor comment, but I found all the plots hard to read due to the use of small fonts and the fact that lines often hide each other.

---

> ### Author Rebuttal · Authors · 2026-03-31
>
> We appreciate your comments and concerns; we address the concerns in the following paragraphs.
>
> **Linear Approximation:** I would like to make a clarification regarding the “linearization” of the update equation that I hope will help answer some of your questions. We do not assume the updates are linear via linear approximations of the update equations. Rather, our analysis relies on the observation that we can express the update equations as the sum of a linear term and a nonlinear term (see Lemma 3.1 and Equation 10 that follows). This helps motivate our analysis; if we can manage to keep the nonlinear term small enough, the linear term will dominate. This is similar to the intuition motivating the analysis of the noisy power method (Hardt & Price, 2015), which shows that if the noise can be proven to be small, the update is dominated by the linear term rather than the noise. We analyze the full update, including the nonlinear term. Controlling the magnitude of the nonlinear term, which changes in a nonlinear way with each iteration as $X_t$ and $Y_t$ update, is what makes the analysis so challenging, even on simple graph classes like the SBM, and is what makes our techniques novel.
>
> The potential function we measure throughout our analysis is the fraction of the embedding’s mass that lies on the space spanned by the top eigenvectors of the (expected) linear term. And since the graph is drawn from an SBM, the top eigenvectors of the linear term contain the community assignments for each node. Therefore, we would expect that the learned embeddings also be “block-like”. In fact, this is what our examples in Section 6 and Appendix G show: that the embeddings are similar to those of the (expected) linear portion because they do in fact converge to the space spanned by the top eigenvectors of the linear portion, as our analysis suggests. In Appendix G, we plot the one-dimensional embeddings, showing how they capture the community structure of the graph.
>
> The literature is full of examples of embeddings generated on real graphs that demonstrate their usefulness on downstream tasks like community recovery, which is why we do not include any examples on real graphs in this paper. Our contributions are theoretical, and we chose to analyze SBMs because they are a standard benchmark in theoretical work analyzing graph clustering algorithms (e.g. McSherry, 2001; Ng, et al., 2001), including prior works studying DeepWalk (see Harker et al. (2024), Zhang et al. (2024), Davison, et al. (2024)). Similar assumptions are common in theoretical works, e.g., in the vast literature on clustering Gaussian mixtures. The SBM case is still a difficult case to analyze (as we show), but we hope that our work will help lay the foundation for future works that may analyze DeepWalk on more realistic graphical models.
>
> **Relevant Work:** Thank you for pointing us to these previous works. We certainly want to include a thorough discussion of all important related works. We will make sure to cite and discuss these in the final version.
>
> The linearized update is related to autocovariance but only at a high level as far as we can tell. Nonetheless, we will cite at least 2-3 recent papers as pointers to the literature. Our focus was on controlling the non-linear error term, so we jumped into methods for analyzing it.
>
> There are some important differences between the analysis done by Karkada, et. al. (2025) and our paper that we believe make our paper important:
> * The most important difference is that Karkada, et. al. (2025) study a taylor approximation of the SGNS objective, ignoring the higher order nonlinear terms. We analyze the full SGNS objective, nonlinearities and all. It is our inclusion of the nonlinearities that makes our analysis, requiring new novel techniques.
> * They analyze the global (non-stochastic) objective, while we analyze the stochastic dynamics.
> * They assume that the node and context embeddings are equal to each other (i.e., $X=Y$), allowing them to take advantage of symmetries. We analyze the asymmetric updates (i.e., $X \ne Y$).
> * They analyze the gradient flow ($\eta \rightarrow 0$). While we require $\eta$ to be small, we do not analyze the update in the limit where $\eta \rightarrow 0$. In fact, we provide an upper bound on how large $\eta$ can be and have our analysis hold (Theorem 4.1).
> We will include these points in the final version.
>
> **Small Font on Plots:** We apologize if the plots were hard to read. Of course, we want the plots to be easy to read and understand. We will make the fonts larger and try to make the plots easier to read in the final version.
>
> We hope this response helps address your questions and concerns. Please let us know if you have any additional questions. We would love to answer them.

---

> > ### Author Rebuttal · Reviewer_FXka · 2026-04-04
> >
> > My intuition for the connection with autocovariance is that L is a weighted sum of autocovariance matrices when the number of negative samples is one. Maybe that is what the authors referred as high-level connection or I misunderstood the definition of L.
> >
> > Regarding my point about the linear approximation, I was asking whether useful embeddings can be generated when we force the nonlinear term to be small enough for the analysis to hold. In general, the paper switches from making general claims about deepwalk embeddings and other that seem to hold only for the SBM (including the experimental results).

---

> > > ### Author Response · Authors · 2026-04-07
> > >
> > > I want to thank the reviewer for their follow up comments and questions.
> > >
> > > **Autocovariance:** You're intuition is correct. The off-diagonal blocks of the expected linear term $L$ can be expressed as a weighted sum of autocovariances when $s_n=1$ (I worked out the math). The update step effectively performs a form of power iteration on a matrix related to the weighted sum of autocovariances. The works of Delvenne et al. (2010), Schaub et al. (2019), and Huang et al. (2021) provide a broader context for why random-walk-based embeddings work by linking them to the dynamical properties of the graph. Our work complements these by showing that the stochastic process of DeepWalk with negative sampling (SGNS) actually tracks these dynamical properties even under high-variance updates when the radius remains small. We appreciate your assistance in making this connection, and we will update the reference section to make this connection explicit.
> > >
> > > **Linear approximation:** You are correct in that the first section of our paper is more general than second; our convergence analysis can be applied more generally, as long as the graph can be shown to have a sufficiently low-rank structure. To apply this analysis to other graph classes, such as mixed membership models, we would need to show that a sufficient spectral gap exists (along with some other minor conditions on $C$). We use the SBM as a concrete, non-trivial class of graphs that allows us to obtain provable structural guarantees, demonstrating the algorithm's usefulness on this class of graphs. We have not explored the performance of small-ball embeddings beyond the context of community recovery, such as node classification or linkage prediction, although this would be an interesting direction for future research.

---

### Official Review · Reviewer_2jr3 · 2026-03-13

**Soundness:** 3
**Presentation:** 4
**Significance:** 3
**Originality:** 3
**Overall Recommendation:** 5
**Confidence:** 3

**Summary:**

The paper studies the optimization dynamics of DeepWalk (with negative sampling), which is one of the foundational approaches in the graph representation learning field. The authors analyze the stochastic gradient updates of the model and discuss that when the embedding norms remain small, the updates can be approximated by linear dynamics. Building on this, it studies the behavior of the method on graphs generated by the Stochastic Block Model and shows that the maximum norm of the learned embeddings can be bounded throughout training under appropriate initialization conditions. Under these assumptions, the analysis shows that the learned embeddings can capture the community structure of the graph and examines how it depends on the magnitude of embedding norms during training.

**Compliance With Llm Reviewing Policy:**

Affirmed.

**Final Justification:**

The authors addressed my concerns, and I decided to keep my positive score (accept).

**Key Questions For Authors:**

**Questions**
- In Defn. 2.1, the co-occurrence matrix $C$ does not explicitly show a dependency on the number of walks $R$. Could you clarify if this is the limiting co-occurrence matrix as $R \rightarrow \infty$, and perhaps state this more explicitly in the text?
- While the theoretical results hold for general embedding dimension $d$, the experiments appear to focus mainly on $d=1$. Have the authors evaluated whether the theoretical insights remain consistent for higher-dimensional embeddings?
- The analysis relies on the assumption that embeddings remain in a small-norm regime during training. Do typical DeepWalk training runs remain in this regime in practice, or would special initialization or regularization be required to ensure this behavior?
- Have you evaluated if the "small norm" constraint impacts the downstream performance compared to standard DeepWalk initializations?
- Your analysis utilizes a co-occurrence matrix $C$. In practice, does your method require the construction of this (potentially dense) matrix?
- The theoretical guarantees are derived mainly for SBM, but how can these insights from this analysis behave on real-world graphs?
- The analysis relies on the SBM graphs, how about beyond SBM?
- The analysis of community recovery depends on the negative sampling parameter being $s_n=1$. Since larger negative sampling values are commonly used in practice, can we expect similar behavior for larger values or are the theoretical guarantees limited to this case?


**Typos and additional notes**
- In Line 68 of the right column, “emebeddings” -> “embeddings”
- In Line 188, “equation” -> “Equation”
- In the text, the notation, $L = \mathbb{E}[L_{(i,j)}]$, makes it somewhat difficult to distinguish between random variable and the expectation terms.

**Limitations:**

yes

**Strengths And Weaknesses:**

The paper is well structured and the authors provide high-level intuition before presenting the technical analysis, which helps the reader to understand the motivation behind the theoretical results. The work provides an interesting perspective on DeepWalk (with negative sampling) by analyzing the optimization trajectory of SGD, rather than focusing only on the objective function. The observation that the updates behave approximately linearly in the small-norm regime provides useful intuition for the connection between DeepWalk and spectral methods. Theoretical results show that embedding norms remain bounded under certain initialization conditions and the embeddings can recover community structure in SBM graphs.

The theoretical guarantees are derived primarily for SBM graphs, but it leaves open the question of how these insights from this analysis behave on real-world graphs. The theoretical analysis for community recovery also appears to rely on the negative sampling parameter being $s_n = 1$ but larger numbers of negative samples are typically used in practice, so this raises questions about how the theoretical guarantees translate to realistic parameter settings.

---

> ### Author Rebuttal · Authors · 2026-03-31
>
> We thank the reviewer for their insightful feedback and questions.
>
> **Co-occurrence Matrix:** Yes, the co-occurrence matrix $C$ defined in Definition 2.1 is the limiting co-occurrence matrix as defined as $R \rightarrow \infty$. We will edit the definition and the surrounding text to highlight this fact for the reader.
> 	In equations (1) and (4), we see that the “negative sampling” term is assumed to be the expected value over the contexts. In order to calculate this sum, all the co-occurrences need to be calculated beforehand. Of course, this is not done in practice. However, the limiting-form of $C$ can be constructed from the adjacency matrix $A$ alone, if one does not want to conduct random walks.
>
> **Higher-Dimensional Embeddings:** The columns of the embedding $W$ has the iteration applied independently of the other columns, and each column converges to the dominant eigenvector (as in power iteration). Thus, we primarily focus on one-dimensional embeddings in the experiments, which also saves on computation. Also, a common objective in community recovery problems is to recover communities using the smallest embedding dimension possible. We wanted to highlight that our results hold for one-dimensional embeddings, while spectral approaches need embeddings of size polylogarithmic in the number of communities.
>
> **Small Norm Regime:** In our experience, the norm remains small as the embeddings “align” with the subspace spanned by the linear term’s eigenvectors. However, we’ve seen that if the algorithm is allowed to continue to run, the norm grows quickly, but the embeddings continue to have this block-like structure that enables recovery. We think that this “alignment” might just be the first “phase” of multiple training phases. But even if there are additional phases of learning, the embeddings are still useful after this first alignment phase. Studying these regimes where the norm is not small is an interesting direction for future work.
>
> Including a regularizer is a very interesting idea, especially if it may help us to control the size of the embeddings more easily. This may be the answer to exploring a larger initialization radius. Nevertheless, we leave this for future researchers.
>
> We have not explored the impact of the initialization radius on downstream tasks outside of community recovery. We leave these as open directions for future research. However, larger initialization radii are still adept at recovering communities, and the literature is full of examples of embeddings generated on real graphs that demonstrate their usefulness on downstream tasks like classification, as well as community recovery.
>
> **Beyond SBMs:** The literature is full of examples of embeddings generated on real graphs that demonstrate their usefulness on downstream tasks like community recovery, which is why we do not include any examples on real graphs in this paper. Our contributions are theoretical, and we chose to analyze SBMs because they are a standard benchmark in theoretical work analyzing graph clustering algorithms (e.g. McSherry, 2001; Ng, et al., 2001), including prior works studying DeepWalk (see Harker et al. (2024), Zhang et al. (2024), Davison, et al. (2024)). Similar assumptions are common in theoretical works, e.g., in the vast literature on clustering Gaussian mixtures. The SBM case is still a difficult case to analyze (as we show), but we hope that our work will help lay the foundation for future works that may analyze DeepWalk on more realistic graphical models.
> 	That said, our first result as stated (about the low rank nature of the solution) applies to any graph with a sufficiently "low rank" structure. So for example, it applies to mixed membership models. Applying these results would require showing that a certain spectral gap condition (and some other minor conditions on $C$) holds (for example, see Lemma C.1 in the appendix).
>
> **Negative Sampling:** Our results hold for sufficiently dense SBMs. The eigenvalues belonging to non all-ones eigenvectors depend on a number of parameters from the SBM: $p$, $q$, the size of the context window $S$, the length of the walk $L$. It may be possible to find a combination of these parameters where the eigenvalue belonging to the all-ones vector isn’t so dominant. If so, we may be able to have values of $s_n \ge 1$ and still be able to implicitly recover communities. Generally, though, our results align with that of prior theoretical works (Saunshi, et al., 2019; Harker and Bhaskara, 2023; Harker and Bhaskara , 2024), all of which suggest that performance on downstream tasks tend to decline as the number of negative samples grows too large.
>
> **Typos and Additional Notes:** Thank you for pointing out these typos. We will correct them in the final version. We will also consider adjusting the notation around expected values.

---

> > ### Author Rebuttal · Reviewer_2jr3 · 2026-04-04
> >
> > Thank you for the additional clarifications, and I'll keep my positive score.

---

### Official Review · Reviewer_HG5x · 2026-03-16

**Soundness:** 3
**Presentation:** 3
**Significance:** 3
**Originality:** 2
**Overall Recommendation:** 4
**Confidence:** 3

**Summary:**

This paper aims to fill the gap in the theoretical analysis of the DeepWalk algorithm. Although DeepWalk is widely used in practice for learning graph node embeddings, existing theoretical results have primarily focused on the global optima of the objective function, leaving a lack of deep understanding regarding the optimization trajectory when running gradient descent from random initialization. The authors prove that for graphs generated from SBM, under the assumptions of "small-norm" initialization and the presence of a spectral gap, the embedding vectors of DeepWalk converge to the column space of a fixed low-rank matrix.

**Compliance With Llm Reviewing Policy:**

Affirmed.

**Key Questions For Authors:**

Can the theoretical results be extended to CSBM?

**Strengths And Weaknesses:**

Strengths：

1. The paper addresses a long-standing theoretical challenge in the field of graph embedding by analyzing the stochastic gradient descent trajectory under non-convex optimization, offering highly valuable guidance.

2. The technical derivations presented in the paper are rigorous and solid.

3. The experimental results align well with the theoretical findings, which is a significant strength.

Weaknesses:

1. The main theoretical results rely heavily on the assumption of "small-norm" initialization, which is a relatively strong precondition.

2. The analysis is limited to Stochastic Block Models (SBM). Extending the framework to Contextual SBMs (CSBM) would be beneficial, as standard SBMs still exhibit a notable gap compared to real-world scenarios.

3. The experiments lack sufficient sensitivity analysis; adding such analysis would strengthen the empirical evaluation.

---

> ### Author Rebuttal · Authors · 2026-03-31
>
> We thank the reviewer for their comments and feedback. To comment on some of the weaknesses raised:
>
> **Small norm initialization:** Indeed, this is a restriction required for our methods, and also for some prior works that handle non-convex objectives (e.g. Stoger and Soltanolkotabi, 2021; Harker and Bhaskara, 2024). This said, even bringing the requirement of small norm to "inverse polynomial" requires a significant effort, as our results show. We state the general problem as an important direction for future research.
>
> **Extensions to other models:** As stated, our first result (about the low rank nature of the solution) applies to any graph with a sufficiently "low rank" structure. So, for example, it applies to mixed membership models. Applying these results would require showing that a certain spectral gap condition (and some other minor conditions on $C$) holds (for example, see Lemma C.1 in the appendix).
> We chose to analyze SBMs specifically because they are a standard benchmark in theoretical work analyzing graph clustering algorithms (e.g. McSherry, 2001; Ng, et al., 2001), including prior works studying DeepWalk (see Harker, et al. (2024), Zhang, et al. (2024), Davison, et al. (2024)). Similar assumptions are common in theoretical works, e.g., in the vast literature on clustering Gaussian mixtures. The SBM case is still a difficult case to analyze (as we show), but we hope that our work will help lay the foundation for future works that may analyze DeepWalk on more realistic graphical models. Exploring other models such as CSBMs is definitely a good direction for future research.
>
> **Sensitivity:** Our experiments primarily show the convergence of the embeddings to the top eigenspace of the expected linear term for varying values of $s_n$. In Appendix G, we provide some additional empirical results for varying values of the number of nodes $n$ and the number of communities $K$. We also provide empirical results on the convergence for increasing radii of initialization, showing that convergence to the top eigenspace of the linear term may occur even in the “not small” radius regime.
> 	One set of parameters where we did not test sensitivity was the sensitivity to the SBM parameters $p$ and $q$. We were not aiming to find, either theoretically or empirically, the ranges of $p$ and $q$ for which our results hold, but rather to find some values of $p$ and $q$ for which we can show convergence to an informational space that could enable community recovery, which was our main objective. That said, determining the ranges for $p$ and $q$ for which our analysis holds, may be an area for future research, especially if studying sparser graphs.

---

> > ### Author Rebuttal · Reviewer_HG5x · 2026-04-05
> >
> > Thank you for the additional clarifications, and I'll keep my positive score.

---

### Decision · Program_Chairs · 2026-04-30

**Decision:**

Accept (spotlight)

**Comment:**

The reviewers unanimously recommend acceptance of the paper with varying degrees of strength. I agree with their assessment and am happy to recommend acceptance of this paper. Several reviewers mentioned that the paper is well-structured and highlighted their appreciation of the theoretical intuition and results. I personally also find this paper and its results very intriguing. The reviewers' concerns have largely been addressed in the discussion period. The reviews and the rebuttal have given rise to several interesting points and results that I encourage the authors to include in their revised manuscript.